Technical Report

# Orthogonal cytokine engineering enables novel synthetic effector states escaping canonical exhaustion in tumor-rejecting CD8+ T cells

Jesus Corria-Osorio [1,2] ✉, Santiago J. Carmona [1,2], Evangelos Stefanidis[1], Massimo Andreatta [1,2], Yaquelin Ortiz-Miranda [1,2], Tania Muller[1,2], Ioanna A. Rota[1,2], Isaac Crespo[1,2], Bili Seijo[1,2], Wilson Castro [1,2], Cristina Jimenez-Luna [1], Leonardo Scarpellino[3], Catherine Ronet[1,2], Aodrenn Spill[1,2], Evripidis Lanitis[1], Pedro Romero [1,2], Sanjiv A. Luther [3], Melita Irving [1] & George Coukos [1,2] ✉

To date, no immunotherapy approaches have managed to fully overcome T-cell exhaustion, which remains a mandatory fate for chronically activated effector cells and a major therapeutic challenge. Understanding how to reprogram CD8+ tumor-infiltrating lymphocytes away from exhausted effector states remains an elusive goal. Our work provides evidence that orthogonal gene engineering of T cells to secrete an interleukin (IL)-2 variant binding the IL-2Rβγ receptor and the alarmin IL-33 reprogrammed adoptively transferred T cells to acquire a novel, synthetic effector state, which deviated from canonical exhaustion and displayed superior effector functions. These cells successfully overcame homeostatic barriers in the host and led—in the absence of lymphodepletion or exogenous cytokine support—to high levels of engraftment and tumor regression. Our work unlocks a new opportunity of rationally engineering synthetic CD8+ T-cell states endowed with the ability to avoid exhaustion and control advanced solid tumors.

Adoptive T-cell therapy (ACT) has emerged as a powerful approach against advanced human tumors[1,2]. Expansion and persistence of adoptively transferred T cells in recipients are required for effective ACT. Host preconditioning with non-myeloablative chemotherapy and post-ACT support with high-dose interleukin (IL)-2 (ref. 3) are commonly used to enable engraftment, expansion and persistence of adoptively transferred T cells, but are associated with important toxicity and clinical costs[4]. Use of T-cell subsets with stem-like potential have been proposed to enhance cell persistence and the efficacy of ACT[5]. The use of synthetic co-stimulatory modules has also enabled remarkable tumor responses upon ACT. However, significant barriers in the tumor microenvironment (TME) still limit T-cell engraftment, persistence and function in solid tumors[6]. Importantly, to date, no immunotherapy approach has managed to effectively overcome T-cell exhaustion, which remains a mandatory fate for chronically activated CD8+ T cells and a major therapeutic challenge for ACT[7]. Indeed, immune checkpoint

[1]Ludwig Institute for Cancer Research, Lausanne Branch, University of Lausanne; and Department of Oncology, Lausanne University Hospital, Epalinges, Switzerland. [2]AGORA Cancer Research Center, Lausanne, Switzerland. [3]Department of Immunobiology, University of Lausanne, Epalinges, Switzerland. ✉e-mail: angeldejesus.corriaosorio@unil.ch; george.coukos@chuv.ch

**Fig. 1 | Orthogonal T-cell engineering enables ACT efficacy without lymphodepletion through in situ expansion of TCF1⁺ precursor and TCF1ⁿᵉᵍ effector CD8⁺ T cells. a**, Experimental design. s.c., subcutaneous. **b**, Waterfall plots showing changes in tumor volumes relative to day 5 after ACT. The best response (smallest tumor volume) observed for each animal after at least 12 d after the first infusion was taken for the calculation (*day 12 post infusion, **day 19 post infusion). ORR includes complete response (CR; 100% reduction in tumor volume) and partial response (PR ≤ −30% tumor change; *n* = 4–14 animals per group). **c,d**, Mice with B16-OVA tumors were treated with either engineered or untransduced OT1 cells as indicated. Tumors were collected on days 5 and 12 after ACT, and cell quantification was performed by flow cytometry. **c**, Total numbers of CD8⁺ TILs at day 12. Data are from four independent experiments (*n* = 4–5 animals per group per experiment). **d**, Total number of CD45.1⁺ OT1 cells on days 5 and 12; day 5: data are from four independent experiments, *n* = 4–5 animals

per group per experiment; day 12: data are from two independent experiments, *n* = 5–6 animals per group per experiment. **e**, Total numbers of TCF1⁺ and TCF1ⁿᵉᵍ OT1 TILs on day 12. Data are from two independent experiments (*n* = 6 animals per group). Data are presented as mean values ± s.d. A Brown–Forsythe and Welch analysis of variance (ANOVA) test combined with Tukey's test to correct for multiple comparisons was used for comparing different groups in **c** and **e**. A two-tailed Student's *t*-test with Welch's correction was used for comparing day 5 and day 12 in **d**. **P* < 0.05, ***P* < 0.01, ****P* < 0.001, *****P* < 0.0001. Representative dot plots show the distribution of CD8⁺ TILs of each indicated treatment based on the expression of TCF1. **f**, Representative immunofluorescence micrographs of tumor sections from each experimental group on day 12. Filled triangle, TCF1⁺OT1; open triangle, TCF1ⁿᵉᵍOT1; white arrows, TCF1⁺ endogenous CD8⁺ TILs. NS, not significant.

blockade primarily attenuates inhibition of exhausted CD8[+] T cells rather than inducing novel, non-exhausted effector states[8].

In this Report, we hypothesized that T cells could be rewired through rational engineering with orthogonal cytokines (that is, perturbations that activate distinct and complementary immune functional axes) to acquire—upon chronic activation—a functional state away from canonical exhaustion, enabling them to reject tumors. We introduced an IL-2 variant (IL-2v), which does not engage the high-affinity IL-2 receptor α-chain (CD25)[9,10], hypothesizing that, unlike canonical IL-2 (ref. [11]), IL-2v would promote CD8[+] T-cell stemness. We also introduced IL-33, a powerful pro-inflammatory alarmin[12], to unleash the cross-priming potential of tumor-associated dendritic cells[13] and complement the stemness-promoting effect of IL-2v. We also tested whether targeting the programmed cell death protein 1 (PD-1)–programmed death ligand 1 (PD-L1) pathway by introducing a secreted PD-1 decoy (PD1d) would improve therapy. We show that orthogonally engineered CD8[+] T cells evade canonical PD-1[+]TOX[+] exhaustion and acquire superior effector functions in vivo, achieving high levels of engraftment and regression of poorly immunogenic tumors, in the absence of preconditioning lymphodepletion or exogenous cytokine support.

## Results

### Orthogonal engineering enables ACT efficacy without lymphodepletion

To begin, we evaluated the PD1d, a fusion molecule comprising the ectodomain of mouse PD-1 linked to the Fc region of human IgG4. Engineered cells efficiently secreted PD1d, which bound PD-L1 specifically in vitro (Extended Data Fig. 1a). ACT using PD1d-engineered OT1 cells in lymphodepleted mice bearing small (~30 mm) tumors, delayed tumor growth (Extended Data Fig. 1b). We ascertained that OT1 cells transduced with the PD1d/IL-2v or PD1d/IL-33 modules could express and secrete the two transgenes simultaneously (PD1d, 50–100 ng ml[−1]; plus IL-2v, 100–300 ng ml[−1]; or IL-33, 10–20 ng ml[−1] per 10[6] cells over 72 h, respectively; Extended Data Fig. 1c,d).

We conducted ACT in mice with more advanced (100 mm[3]) tumors in the absence of preconditioning lymphodepletion (Fig. 1a). Infused OT1 cells were mostly TCF1[pos] central memory-like cells (Extended Data Fig. 1d). Two infusions of $5 \times 10^6$ untransduced OT1 cells had minimal impact on tumor growth (Fig. 1b and Extended Data Fig. 2a). PD1d-transduced OT1 cells had minimal effect, and the combination of anti-(α)PD-L1 with untransduced OT1 cells produced comparable results to PD1d-OT1 cells. IL-2v-engineered or double PD1d/IL-2v-engineered OT1 cells were not more effective. OT1 cells transduced with IL-33 had significant but minimal effect, as did PD1d/IL-33 OT1 cells. Strikingly, PD1d/IL-2v/IL-33 OT1 cells (that is, a 1:1 ratio of PD1d/IL-2v:PD1d/IL-33 cells) induced marked tumor regression and achieved a tumor objective response rate (ORR) of 85.7% (predicted occurrence probability, 83.3%; Extended Data Tables 1 and 2), while the ORR was 0–9% for any other treatment. Orthogonal ACT administered earlier (starting on day 6) led to B16-OVA tumor cures (Extended Data Fig. 2b,c), while it delayed tumor growth and also increased survival in mice bearing advanced MC38-OVA colon tumors (Extended Data Fig. 2d). Similarly, PD1d/IL-2v/IL-33 ACT using Pmel CD8[+] T cells, which

target gp100, impacted B16-F10 tumors (Extended Data Fig. 2e), demonstrating the generalizability of this approach.

### Orthogonal ACT expands TCF1[+] and TCF1[−] CD8[+] tumor-infiltrating lymphocytes in situ

Following PD1d/IL-2v/IL-33 OT1 ACT, we saw marked cell-autonomous expansion of adoptively transferred T cells in tumors. By day 5, we observed >800-fold more CD8[+] tumor-infiltrating lymphocytes (TILs) in tumors treated with PD1d/IL-2v/IL-33 OT1 relative to non-treated B16-OVA tumors (Extended Data Fig. 3a,b). Twelve days after ACT (upon tumor response), we observed significantly more total CD8[+] TILs and marked intratumoral expansion of transferred OT1 cells following PD1d/IL-2v/IL-33 OT1 ACT (Fig. 1c,d and Extended Data Fig. 3c), while PD1d/IL-2v OT1 or PD1d/IL-33 OT1 cells showed modest or minimal levels of tumor expansion, respectively.

In conventional ACT schemes, lymphodepletion is required to mobilize host homeostatic cytokines promoting expansion of the transferred cells[14]. We asked how orthogonally engineered transferred cells achieved such expansion in the absence of lymphodepletion. TCF1 is a transcription factor (TF) essential for early T-cell self-renewal, and TCF1[+] precursor T cells are required for successful checkpoint blockade immunotherapy[15] and ACT[16]. We noticed marked expansion of TCF1[+] OT1 cells when IL-2v was included in the ACT scheme, that is, with PD1d/IL-2v/IL-33 or PD1d/IL-2v (Fig. 1e,f), and the number of total OT1 TILs strongly correlated with the number of TCF1[+] cells (Extended Data Fig. 3d). Conversely, the TCF1[+] compartment failed to expand with PD1d/IL-33-engineered or untransduced OT1 cells, suggesting that IL-2v promoted stemness and expansion of transferred cells, obviating lymphodepletion.

Differentiation to effector T cells is associated with downregulation of TCF1 (ref. [17]). Unsurprisingly, effective tumor control required high numbers of TCF1[neg] effector CD8[+] TILs, a condition met solely following PD1d/IL-2v/IL-33 ACT (Fig. 1e,f), where 75–90% of OT1 TILs were TCF1[neg]. Conversely, there were far fewer TCF1[neg]CD8[+] TILs after PD1d/IL-2v ACT (Fig. 1e,f), suggesting that although IL-2v expanded TCF1[+] precursors, these cells were unlikely to transition to a TCF1[neg] effector-like state in the absence of IL-33. Strikingly, PD1d/IL-33 ACT was associated with a high frequency of TCF1[neg]CD8[+] TILs but overall poor expansion and low TCF1[+]CD8[+] numbers, consistent with the absence of IL-2v from the mix. Thus, orthogonal T-cell engineering enables ACT efficacy without lymphodepletion through in situ expansion of TCF1[+] precursor and TCF1[neg] effector CD8[+] T cells.

### Orthogonal ACT induces a novel effector CD8[+] T-cell state

To understand how orthogonally engineered cells could achieve regression of non-immunogenic B16-OVA tumors, we analyzed total CD8[+] TILs from different ACT settings using single-cell RNA-sequencing (scRNA-seq; Fig. 2a). Unsupervised clustering analysis revealed five distinct states (clusters C1–C5; Fig. 2b), which closely reflected the different engineering strategies. TILs from ACT using untransduced T cells distributed in C1 characterized by naïve/central memory genes (for example, *Sell, Il7r, Tcf7, Lef1*) and C4 characterized by canonical T-cell exhaustion markers (*Tox, Id2, Nfatc1, Pdcd1, Lag3, Havcr2, Entpd1,*

**Fig. 2 | Orthogonal engineering induces a novel effector CD8[+] T-cell state.**
**a**, Experimental design. Briefly, mice with B16-OVA tumors were treated as indicated; then, tumors were collected on days 5 and 12 after ACT, and a cell suspension of CD45[+] enriched in CD8[+] TILs was obtained by FACS sorting and single-cell sequenced using 10x Genomics. **b,c**, Cluster composition per treatment and UMAP plot showing a low-dimensional representation of cell heterogeneity and unsupervised clustering results, where contour plots depict high cell density areas for each treatment. **d**, Violin plots showing expression levels of important T-cell markers. **e**, Projection of each cluster onto the reference TIL map using ProjecTILs. **f,g**, Independent component analysis 26 (ICA26) effector component[21] for cells in the reference TIL atlas (gray) versus

C5 (red) samples, plotted on the z axis of the UMAP plot (**f**) and driver genes for the most significant independent component (ICA26) that separates cluster C5 from the TIL reference map driver genes (**g**). **h**, Volcano plot showing significant DEGs between C4 and C5. Calculations were performed with the function FindMarkers from Seurat using the MAST test for differential expression. **i**, CD8[+] TIL *Tox*-knockout[26] GSEA of Gzmc[+] C5 versus Gzmc[neg] C4 cells. Calculation was performed with the GSEA function from clusterProfiler with default parameters and using the top 200 differentially expressed cluster genes with adjusted *P* value < 0.01 ordered by decreasing FC. False discovery rate (FDR) was calculated with the Benjamin–Hochberg method. FC, fold change; NES, normalized expression score.

*Ccl3*, *Ccl4* and *Ccl5*; Fig. 2c,d). Interestingly, TILs from PD1d/IL-2v ACT clustered mostly in C1 (naïve/central memory), while PD1d/IL-33 ACT TILs clustered mostly in C2, characterized by effector memory genes

(*Gzmk*, *Gzma*, *Ccl5*)[18], highlighting the different effects of the two cytokines when used alone. Strikingly, the co-presence of the two cytokines drove PD1d/IL-2v/IL-33 ACT CD8+ TILs to depart from all

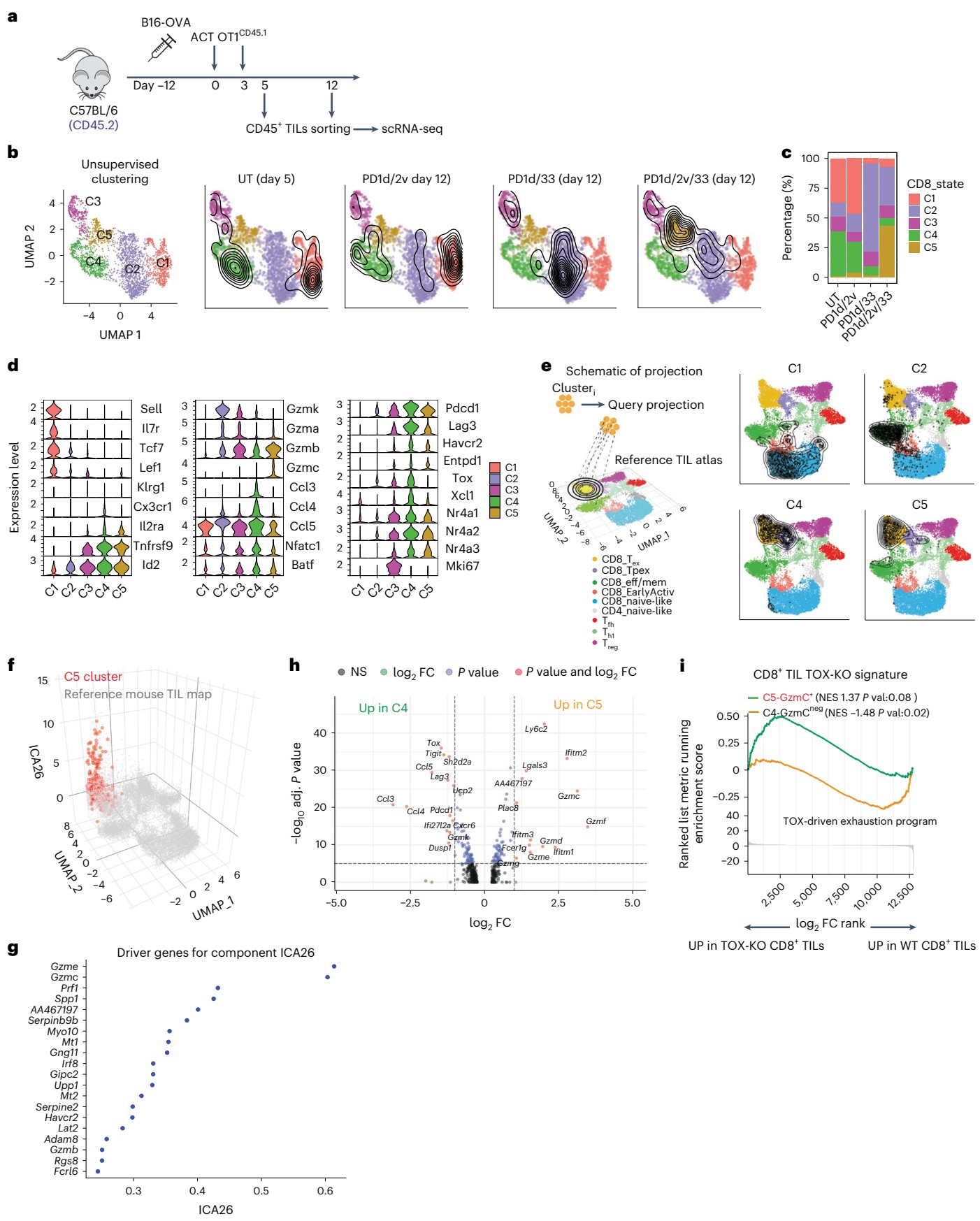

above states and cluster mostly in C5 (Fig. 2c,d), distinguished by low levels of exhaustion and high levels of cytotoxicity genes. The latter included *Gzmb* and notably *Gzmc*, a nonredundant backup granzyme mediating enhanced cytolytic functions upon persistent antigenic stimulation and alloimmunity[19,20]. We validated the gene-based annotation by projecting each cell cluster onto a reference atlas that comprises well-defined TIL states from untreated mouse tumors using ProjecTILs[21]. C1 projected on CD8+ naïve-like cells, C2 on CD8+ effector memory cells and C4 to canonical precursor ($P_{EX}$) and terminal ($T_{EX}$) exhausted states (Fig. 2e). Not surprisingly, C5 effector-like TILs projected largely onto the canonical $T_{EX}$ space, as this is the closest/more similar cell state of the TIL reference map, which only includes previous public data[21]. We next interrogated C5 TILs at a higher dimension using independent component analysis, which significantly separated C5 TILs from the reference $T_{EX}$ TILs (Fig. 2f), attributing to C5 a unique cytotoxic effector-like program, comprising granzymes, notably *Gzmc*, perforin, other effector machinery genes and pro-survival factors (Fig. 2g). Differential gene expression analysis confirmed that C5 is transcriptionally distinct from canonical TOX+ exhaustion, represented by C4 (Fig. 2h and Supplementary Table1). Relative to the latter, C5 TILs overexpressed *Gzmc* and several other granzymes; members of the interferon-inducible transmembrane proteins and *Plac8*, implicated in optimal CD8+ T-cell response against viral infections[22]; *Fcer1g*, a marker of tumor-reactive innate-like high-cytotoxicity TILs[23]; and pro-survival *Bcl2*. Furthermore, C5 TILs overexpressed *Ly6c2*, a marker of precursor CD8+ T cells absent in exhausted cells[18], and downregulated *Pcdc1*, *Tigit* and *Lag3*, exhaustion-associated chemokines *Ccl3*, *Ccl4* and *Ccl5*, as well as exhaustion-associated TFs encoded by *Bhlhe40*, *Nfatc1*, *Nrp2a* and *Nrp3a*[24,25] (Fig. 2d,h). Importantly, *Tox*, encoding a critical TF for the generation and maintenance of exhausted CD8+ T cells[26–30], was nearly absent in C5 cells (Fig. 2d,h). Indeed, C5 cells were enriched for the gene signature of *Tox*-knockout CD8+ TILs[26] (Fig. 2i). Altogether, this analysis demonstrates that PD1d/IL-2v/IL-33 ACT induces a noncanonical PD-1+ TOX^neg effector-like cell state.

## C5 is a synthetic T-cell state, uniquely acquired by engineered CD8+ TILs

The above analyses were performed on total CD8+ TILs. We asked whether the novel C5 T-synthetic effector ($T_{SE}$) state was acquired by transferred or endogenous CD8+ TILs. Separately interrogating OT1 and endogenous CD44+CD8+ TILs that were subjected to fluorescence-activated cell sorting (FACS), and analyzing more cells and at different time points following PD1d/IL-2v/IL-33 ACT (Fig. 3a), revealed, in addition to C1, C2 and C5 $T_{SE}$ states, two new states, C6 and C7 (discussed later), and segregated $P_{EX}$ states from canonical $T_{EX}$ states within C4 (Fig. 3b and Extended Data Fig. 4a–c). Importantly, whereas endogenous CD44+CD8+ TILs mostly distributed across C4 (canonical $T_{EX}$) and C2 (effector memory) at the time of tumor regression following PD1d/IL-2v/IL-33 ACT, the C5 $T_{SE}$ effector state identified above was a unique property of OT1 TILs (Fig. 3c and Supplementary

Table 2). Specifically, OT1 TILs gradually acquired C5 during the tumor regression phase; most OT1 cells were distributed in C2, C6 or C7 states, and a minimal fraction entered C5 $T_{SE}$ by day 5 after ACT, while by day 8, cells were mostly in C5 (54%) and C6 (22%) $T_{SE}$ states, and by day 12, >90% of cells were in C5 $T_{SE}$, coinciding with tumor regression (Fig. 3c,d and Extended Data Fig. 4d). Thus, only adoptively transferred orthogonally engineered cells differentiate to the novel C5 $T_{SE}$ cell state during tumor regression.

We confirmed this state was distinct not only from canonical $T_{EX}$ TILs (Fig. 3c), but also from effector states previously described in acute or chronic viral infections, for example, short-lived effector CD8+ T cells (SLECs) or CX3CR1+ exhausted intermediate effector cells[31] (Extended Data Fig. 4e,f); C5 $T_{SE}$ cells displayed the lowest expression of *Tox* and the highest expression of multiple granzymes relative to these naturally occurring CD8+ T-cell states.

To assess whether this novel effector state may be detectable elsewhere, we built a new TIL reference map (Fig. 3e) and interrogated available scRNA-seq antitumor T-cell datasets with ProjecTILs[21]. We identified no C5 $T_{SE}$ cells in human CD8+ TILs in three tumor datasets from immunotherapy-naïve individuals (Fig. 3f) or during response to anti-PD-1 (Fig. 3g). Furthermore, there were no C5-like cells among circulating CD19 CAR CD8+ T cells during peak expansion in four individuals with durable chimeric antigen receptor (CAR)-T-cell persistence[32], and the majority of CAR-T cells corresponded to $T_{EX}$-like cells (Fig. 3h). In mice, OT1 TILs collected following ACT combined with PD-1 blockade[33] exhibited minimal C5 $T_{SE}$ (Fig. 3i) and were enriched in genes of NFAT-dependent exhaustion[34], consistent with the notion that PD-1 blockade does not reprogram TILs away from exhaustion[8]. Finally, we compared $T_{SE}$ TILs with CD8+ 'better effectors' ($T_{BE}$) reported following anti-PD-1–IL-2v bispecific immunocytokine treatment[35]. Although a subset of PD-1–IL-2v-induced $T_{BE}$ TILs reached a noncanonical state expressing low levels of *Tox* (Extended Data Fig. 4g) and projecting on a C5-like space (Fig. 3j), $T_{BE}$ cells were distinct from C5 $T_{SE}$ TILs, which instead expressed no *Tox*, and exhibited lower *Tbx21* or *Cx3cr1* and markedly higher granzyme levels than $T_{BE}$ (Fig. 3j and Extended Data Fig. 4g). Thus, C5 $T_{SE}$ is a novel and unique synthetic effector state, seemingly never seen naturally to date in mice or humans but massively acquired in vivo by exogenous T cells following orthogonal ACT.

## $T_{SE}$ TILs are polyfunctional effector cells with direct antitumor properties

To further characterize the C5 $T_{SE}$ state, we phenotyped TILs gating on the two distinctive biomarkers, TOX and Gzmc, using FACS. In untreated mice at baseline (12 d after tumor inoculation), potential tumor-reactive (CD44^highPD-1+) endogenous CD8+ TILs were as expected TOX+Gzmc^lo–neg (Fig. 4a), consistent with canonical exhaustion. During tumor regression after PD1d/IL-2v/IL-33 ACT, OT1 TILs were mostly PD-1+ but highly enriched in TOX^negGzmc+ cells (Fig. 4a). In addition, most Gzmc+ OT1

**Fig. 3 | C5 is a synthetic T-cell state, uniquely acquired by engineered CD8+ TILs. a**, Experimental design. **b**, UMAP plot showing a low-dimensional representation of cell heterogeneity and unsupervised clustering results. **c**, UMAP plot showing cluster composition. Contour plots depict the clusters covered by each cell compartment and bar plots show the quantification. Volcano plot showing the DEGs between C5 $T_{SE}$ cells and $T_{EX}$ cells from the reference TIL map. Calculation was performed with the function find. discriminant.genes (ProjecTILs package) using a two-sided non-parametric Wilcoxon rank sum test with Bonferroni correction. **d**, Cluster composition/ time point of OT1 TILs recovered 5, 8 and 12 d after orthogonal ACT. **e**, Schematic of projection of human or mouse CD8+ TILs onto the OT1/endogenous transcriptomic space. **f**, Projection of human CD8+ TILs from three different types of tumors. **g**, Projection of human CD8+ TILs upon PD-1 blockade in responder (on-R) versus non responder (on-NR) individuals with breast cancer.

**h**, Projection of CD19-BBz human CAR-T cells recovered during blood expansion peak upon ACT[32], onto the OT1/endogenous transcriptomic space. Contour plots depict the clusters covered by each cell dataset, and bar plots show the cluster composition calculated by ProjecTILs. **i**, Venn diagram shows the overlap between genes upregulated in C5 (in the comparison in **c**) and genes upregulated upon PD-1 blockade from ref. 33. CA-RIT-NFAT1 (ref. 34) GSEA of canonical $T_{EX}$ TILs (genes downregulated in the comparison in **c**), OT1 cells recovered upon PD-1 blockade[33] and C5 cells. **j**, Projection of mouse CD8+ TILs harvested upon systemic treatment with PD-1–IL-2v bispecific immunocytokine[35] onto the OT1/ endogenous transcriptomic space. Contour plots depict the clusters covered by each cell dataset, and bar plots show the cluster composition. The radar plot shows the relative expression of important gene markers between the PD-1–IL-2v TILs projected on C5 $T_{SE}$ and the reference cell state from the map.

cells were TCF1[neg] (Fig. 4b) and expressed LY6C but low/no KLRG1 or CX3CR1 (Extended Data Fig. 5a), thus distinct from SLECs or CX3CR1[+] exhausted intermediate effector-like cells.

Next, we compared PD-1[+] OT1 TILs on day 12 after ACT across all conditions. Gzmc[+] cells were uniquely enriched in tumors following PD1d/IL-2v/IL-33 ACT (Fig. 4c). Notably, these cells exhibited a unique

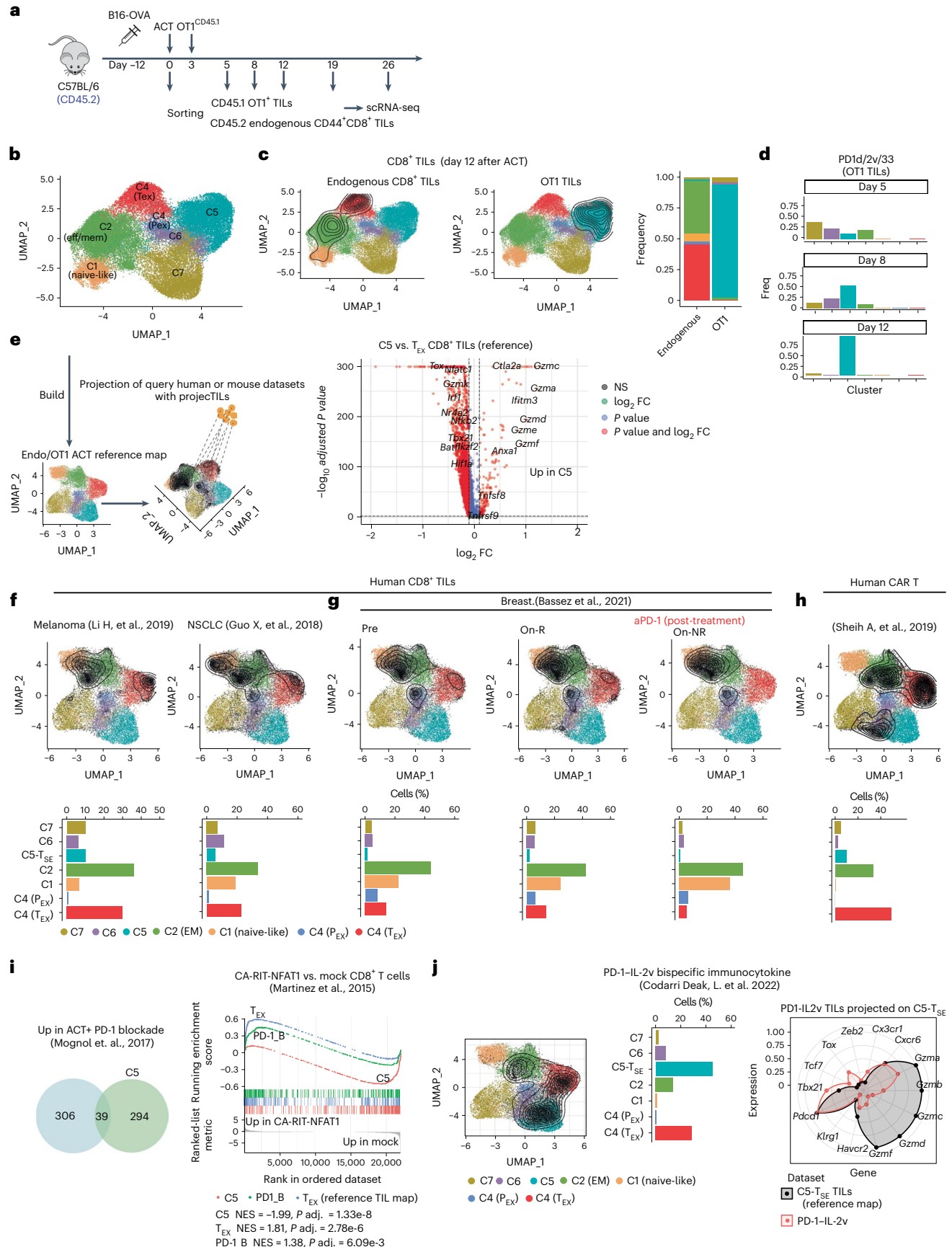

phenotype of polyfunctional effector cells by FACS, with coexpression of Gzmb, Gzma, perforin, tumor necrosis factor (TNF), interferon (IFN)-γ and Ki67 (Fig. 4d), despite also expressing TIM-3 (Fig. 4d,e). Strikingly, while genes upregulated in $T_{EX}$ TILs were involved in pathways associated with sustained T-cell antigen receptor (TCR) signaling, negative regulation of immune system and apoptosis (Extended Data Fig. 5c,d), C5 $T_{SE}$ cells were enriched in energy production, effector molecule synthesis and cell killing (Fig. 4f), revealing improved metabolic and effector fitness. Indeed, relative to canonical $T_{EX}$ TILs, the C5 $T_{SE}$ cell state was enriched in the reported gene signature of CD8+ TILs with high mitochondrial potential[36], as well as in genes associated with better mitochondrial fitness such as cristae formation[37] and mitochondrial translation (Fig. 4g), the latter required for sustained killing by cytotoxic CD8+ T cells[38]. To directly assess this, we collected OT1 TILs from responding tumors following orthogonal ACT, when 80–90% of the OT1 cells are in the $T_{SE}$ state and tested their antitumor activity in vitro and in vivo. Unlike control PD-1+TOX+ exhausted-like OT1 cells generated in vitro (Extended Data Fig. 5e), which exhibited cytolytic activity in vitro, but were unable to control tumor growth in vivo (Fig. 4h), $T_{SE}$ OT1 TILs isolated from mice treated with orthogonal ACT killed B16-OVA cells in vitro and led to tumor rejection in vivo. Thus, the novel Gzmc+ $T_{SE}$ state is endowed with potent cytolytic and effector functions.

## PD-1+ $T_{SE}$ TILs are not functionally restrained by PD-1

We sought to understand whether PD-1 functionally restrains synthetic TOX^neg $T_{SE}$ TILs. Immune checkpoint blockade with αPD-L1 or double αPD-L1/αTIM-3 therapeutic antibody (because TIM-3 was also expressed on OT1 $T_{SE}$ TILs; Fig. 4e) did not improve tumor control by orthogonal ACT (Fig. 5a). Importantly, removing the PD-1 ectodomain from the PD-1–IgG4 decoy neither impacted tumor control by IL-2v/IL-33-transduced OT1 cells (Fig. 5b) nor impaired the accumulation of PD-1+Gzmc+TCF1^neg OT1 TILs (Fig. 5c). OT1 TILs from IL-2v/IL-33 ACT exhibited a similar polyfunctional effector phenotype to TILs from PD1d/IL-2v/IL-33 ACT (Fig. 5d), implying that the novel TOX^neg synthetic cell states were refractory to PD-1-mediated inhibition. Indeed, $T_{SE}$ TILs were not enriched in CD8+ T-cell gene signature of PD-1-mediated inhibition[37] (Fig. 5e). Finally, in a plate-bound PD-L1 assay[37], PD-L1 had no effect on OT1 TILs collected from tumors following IL-2v/IL-33 ACT, while it abrogated the activation of spleen-derived control OT1 T cells upon TCR restimulation ex vivo (Fig. 6). Thus, PD-1 is expressed but remains functionally inconsequential in the novel synthetic TOX^neg differentiation program.

## Orthogonal ACT induces a novel PD-1+ precursor-like cell state

Following orthogonal ACT, Gzmc+ cells were undetectable in tumor-draining lymph nodes (TDLNs) or spleen (Fig. 7a), indicating that transferred cells acquire the $T_{SE}$ cell state specifically in tumors. Importantly, $T_{SE}$ TILs did not express the IL-33 receptor ST2 (Extended Data Fig. 6a), indicating that the contribution of IL-33 is indirect, through reprogramming of the TME[12].

We asked whether $T_{SE}$ could be associated to a precursor-like state, in an analogy to $T_{EX}$ and $P_{EX}$. Following orthogonal ACT, we identified a distinct cluster (C6) of OT1 TILs characterized by precursor features. Much like $P_{EX}$, C6 TILs expressed *Pdcd1* together with stemness-related markers *Ccr7*, *Tcf7* and *Il7r* (Fig. 7b). However, they exhibited no *Tox*, and overexpressed *Gzmc* (Fig. 7c and Supplementary Table 3) and other effector molecules, indicating a novel synthetic state resembling C5 $T_{SE}$ but with precursor potential. Similarly to $T_{SE}$ TILs, these synthetic precursor-like cells ($T_{SP}$) also overexpressed genes associated with the regulation of ATP formation, mitochondrial biogenesis, cristae formation, mitochondrial gene expression and translation (Fig. 7d,e), implying better metabolic and mitochondrial fitness than canonical $P_{EX}$ cells. Moreover, $T_{SP}$ cells were not enriched in CD8+ T-cell gene signatures of PD-1-mediated inhibition[37] (Fig. 7f), indicating that PD-1 expression is also inconsequential in these cells. Trajectory and pseudotime inference analysis suggested that orthogonally engineered OT1 T cells likely transitioned from the original naïve/central memory-like (C1) and effector memory (C2) states of transferred cells to C6 $T_{SP}$ and then C5 $T_{SE}$, bypassing the canonical $P_{EX}$ state and defining a novel PD-1+ TOX^neg synthetic CD8+ T-cell differentiation program (Fig. 7g).

Most PD-1+TCF1+ OT1 TILs during the tumor control phase after PD1d/IL-2v/IL-33 ACT expressed higher Gzmc and lower TOX protein relative to canonical TCF1+PD-1+ $P_{EX}$-like cells from baseline tumors (Fig. 7h). Remarkably, only PD-1+TCF1+ OT1 TILs generated upon orthogonal ACT expressed high levels of Gzmc in contrast to precursor-like cells recovered from PD1d/IL-2v or PD1d/IL-33-treated mice (Fig. 7i). Thus, similar to terminal exhaustion, which is preceded by a PD-1+ precursor state[39], the alternate $T_{SE}$ state achieved by orthogonally engineered cells could be also committed early, and transitions through a PD-1+TCF1+TOX^lo-neg precursor/stem-like synthetic cell state, harnessing a transcriptional program that deviates from canonical $P_{EX}$.

Importantly, C6 $T_{SP}$ cells were also undetectable in TDLNs (Fig. 7j), where most of the TCF1+ precursor-like OT1 cells were PD-1^neg and Gzmc^neg, indicating that they also develop in the TME. Finally, we compared C6 $T_{SP}$ TILs with atypical precursor-like CD8+ T cells reported recently following systemic administration of anti-PD-1 in combination with IL-2 (ref. 40). Although these cells downregulate *Tox*[40], they were mostly classified as effector memory-like cells, thus distinct of C6 $T_{SP}$ TILs, upon projection in the ACT reference map with ProjecTILs (Fig. 7k) and did not express *Gzmc* (Extended Data Fig. 6b). Thus, C6 $T_{SP}$

---

**Fig. 4 | C5 $T_{SE}$ OT1 TILs are polyfunctional effector cells with direct antitumor properties. a**, CD44, PD-1, TOX and Gzmc expression in OT1 TILs from animals treated with PD1d/IL-2v/IL-33 OT1 at day 8 (two independent experiments, $n = 4$ mice per group) and day 12 (five independent experiments, $n = 5$–6 mice per group) after transfer. CD8+ TILs isolated from tumor-bearing mice at baseline were used as the control ($n = 6$ mice per group). Representative dot plots depicting naïve CD8+ T cells from spleen of non-tumor-bearing mice used as negative control. **b**, Gzmc and TCF1 expression in PD-1+ TOX^lo-neg OT1 TILs recovered at day 8 (two independent experiments, $n = 4$ mice per experiment) and day 12 (two independent experiments, $n = 6$ mice per experiment) after transfer. Representative contour plot. **c**, Analysis of Gzmc expression in PD-1+TCF1^neg OT1 TILs recovered at day 12 after transfer. Data from five independent experiments except for PD1d (two independent experiments, $n = 4$ mice per experiment). UT, PD1d/IL-2v and PD1d/IL-33, $n = 4$–5 mice per group per experiment, PD1d/IL-2v/IL-33, $n = 8$ mice per experiment. Data are presented as mean values ± s.d. A Brown–Forsythe and Welch ANOVA test combined with Tukey's test to correct for multiple comparisons was used for comparing different groups; *$P < 0.05$, **$P < 0.01$, ***$P < 0.001$, ****$P < 0.0001$. **d**, Heat map showing normalized protein expression estimated by FACS in PD-1+ TCF1^neg OT1 TILs recovered at day 12. Each column represents an individual mouse ($n = 35$ mice in total). **e**, Representative dot plots showing the expression of TIM-3 and TOX. **f**, Gene Ontology (GO) biological process overrepresentation test of genes upregulated in $T_{SE}$ TILs relative to canonical $T_{EX}$. **g**, GSEA gene set: CD8+ TILs with high mitochondrial potential[36]; ranked list: DEGs between C5 $T_{SE}$ and canonical $T_{EX}$. Reactome pathway GSEA. Ranked list: DEGs between C5 $T_{SE}$ and canonical $T_{EX}$. GSEA was performed with the GSEA function (clusterProfiler package) which uses the Benjamin–Hochberg method for multiple correction. Reactome pathway and GO enrichment were performed with the functions enrichReactome and enrichGO, respectively, from the same package, which uses a one-sided version of the Fisher's exact overrepresentation test to find enriched categories. **h**, In vitro killing assay of sorted OT1 TILs 12 d after orthogonal ACT or in vitro-generated exhausted OT1 cells against labeled B16-OVA tumors using IncuCyte. A two-tailed Student's *t*-test was used for comparing both conditions at 72 h. Antitumor potential of CD44+PD-1+TOX^lo-negGzmC+ OT1 TILs recovered 12 d after transfer in a Winn assay[53] (data from two independent experiments, $n = 5$ mice per experiment for PD1d/IL-2v/IL-33 group and $n = 10$ mice per experiment for the other two groups, $n = 10$ mice per experiment). Data are presented as mean values ± s.d. A log-rank (Mantel–Cox) test was used for comparing the curves. ****$P < 0.0001$.

and C5 T$_{SE}$ represent the two novel states of an alternate, synthetic cell fate, which develops specifically in the TME thanks to the convergence of IL-2v and IL-33.

**IL-2v engineering contributes to key programs to the T$_{SE}$ state**
Because OT1 cells transduced with IL-33 but lacking IL-2v failed to persist (Extended Data Fig. 3c), while cells exposed to both IL-33 and IL-2v

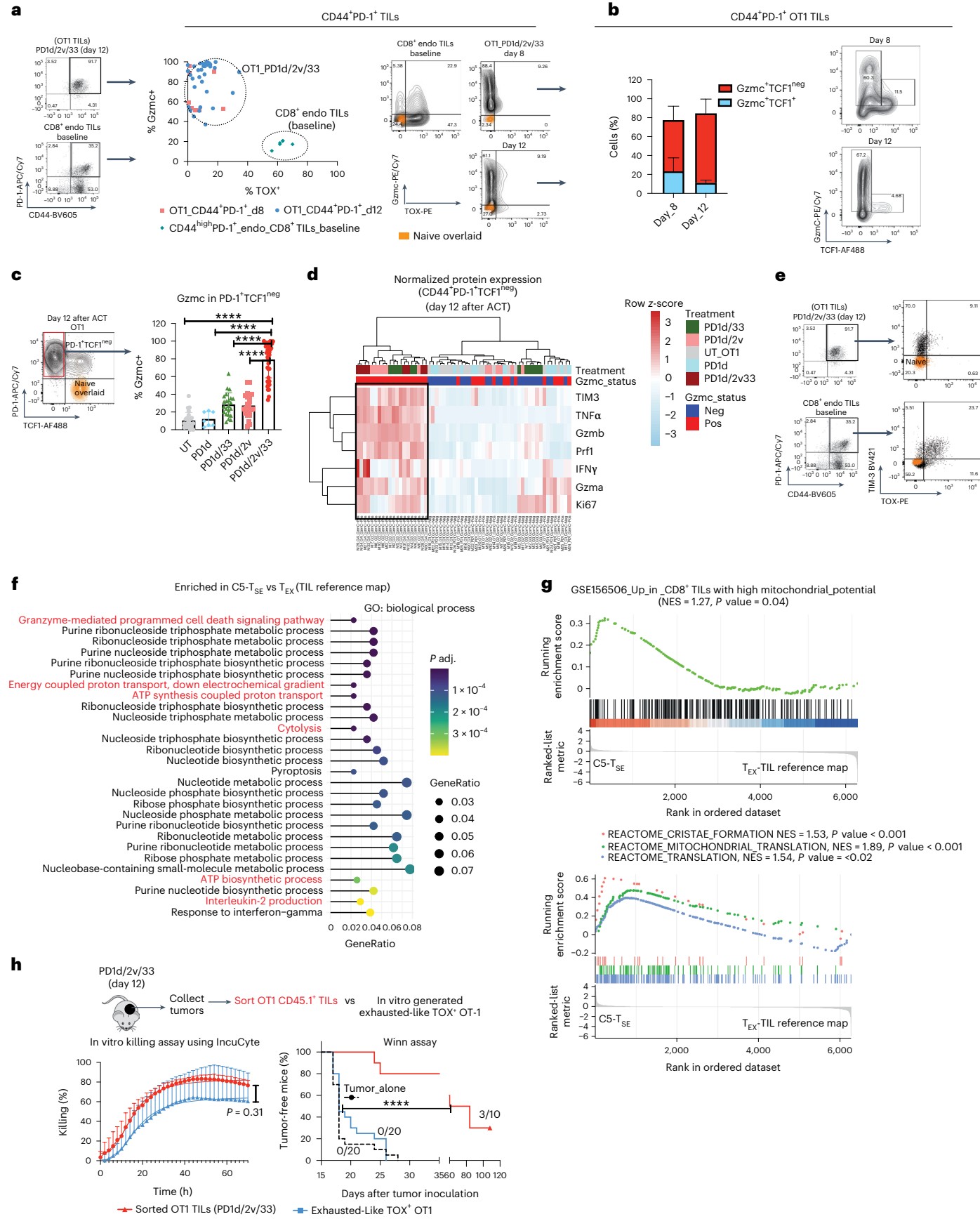

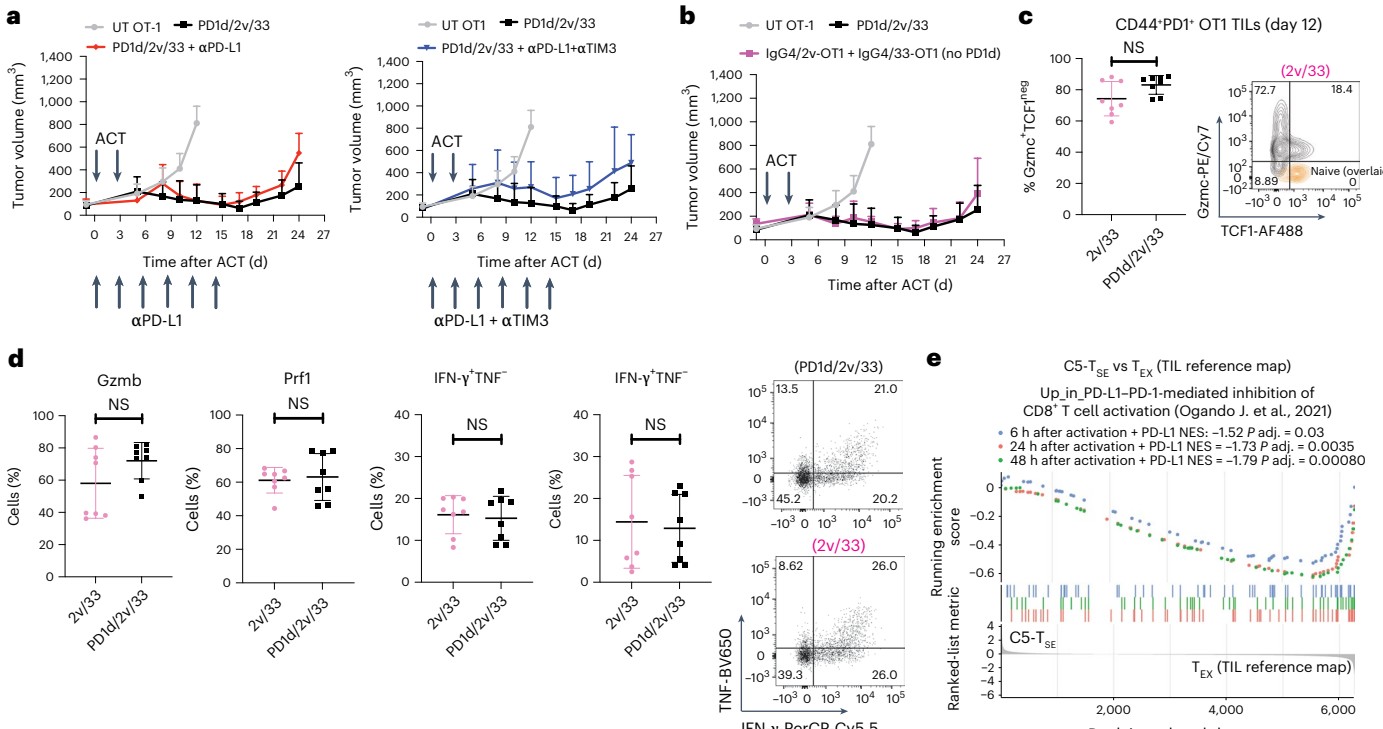

**Fig. 5 | PD-1 decoy is dispensable for orthogonal ACT. a**, Tumor growth control over time of B16-OVA tumor-bearing mice treated with PD1d/IL-2v/IL-33[+] OT1 cells in the presence or absence of 250 µg per mouse of antibodies specific for the indicated immune checkpoints administered intraperitoneally (i.p.) beginning 1 d before the first cell transfer and maintained every 3 d for a maximum of six doses. **b**, Tumor growth control over time of B16-OVA tumor-bearing mice treated with PD1d/IL-2v/IL-33[+] OT1 cells or with OT1 T cells genetically engineered to secrete only IL-2v and IL-33 (no PD-1 ectodomain). **c**, Percentage of Gmzc[+]TCF1[neg] effector-like cells in CD44[+]PD-1[+] OT1 TILs recovered from mice treated with either PD1d/IL-2v/IL-33[+] OT1 cells or IL-2v/ IL-33 OT1 cells 12 d after orthogonal ACT (two independent experiments, $n = 4$ mice per group per experiment). Data are presented as mean values ± s.d. **d**, Expression of effector molecules in CD44[+]PD-1[+]Gzmc[+]TCF1[neg] effector-like cells in OT1 TILs from **b**. A two-tailed Student's $t$-test with Welch's correction was used for comparing both groups in **c** and **d**. NS, $P > 0.05$. A representative experiment of two independent experiments ($n = 6$ animals per group) is shown for experiments in **a** and **b**. **e**, GSEA: CD8[+] T-cell gene signatures of PD-1-mediated inhibition[37]. Ranked list: genes differentially expressed between C5 $T_{SE}$ and canonical $T_{EX}$ from the TIL reference map[21].

exhibited unique fitness, we hypothesized that IL-2v contributed specifically to a TIL-intrinsic differentiation program supporting superior persistence and metabolic fitness. To learn more, we analyzed OT1 TILs 12 d after IL-2v OT1 ACT, that is, transferred in the absence of IL-33 influence (Fig. 8a). These were mostly distributed in a synthetic C7 memory-like cell state (Fig. 8a) characterized by high expression of T-cell activation markers *IL2ra*, *Cd69*, *Tnfrsf9*, the stemness-related *Ccr7*, *Xcl1* (a chemokine involved in DC–T-cell cross-talk) and low expression of *Pdcd1*, *Lag3*, *Havcr2* and *Tigit* and no expression of *Tox*, *Gzmc*, *Cx3cr1* or *Klrg1* (Fig. 8b). Although partly similar to canonical memory-like TIL states (P_EX and effector memory; Extended Data Fig. 7a), C7 memory-like TILs overexpressed genes involved in ribosome biogenesis, glycolysis, ATP generation and nucleotide metabolism (Fig. 8c), suggesting a functional role of IL-2v in preparing cells for superior effector function, fully leveraged in C5 $T_{SE}$ and C6 $T_{SP}$ cells (Figs. 4 and 7).

Using flow cytometry, we validated that most IL-2v OT1 TILs were activated, with high expression of CD25, CD69 and 4-1BB, but arrested in a TOX[neg]TCF1[int]PD-1[lo–neg] C7-like cell state, with no expression of Gzmc (Fig. 8d). Thus, unlike canonical memory cells, which exit stemness and commit to effector (or exhaustion) differentiation upon TCR activation, IL-2v OT1 TILs remain committed to memory/stemness even upon TCR engagement, and only transition to C6 $T_{SP}$ and subsequently C5 $T_{SE}$ upon additional potent signals triggered in vivo by (IL-33 driven) inflammation.

To infer the specific transcriptional wiring induced by IL-2v alone, we identified TFs downstream of IL-2R ($n = 181$) from the ligand–TF matrix of NicheNet[41] and used SCENIC[42] to infer their activity across TIL states. Unbiased clustering analysis revealed 12 groups of differentially active TFs/regulons across the synthetic (C7, C5 $T_{SE}$, C6 $T_{SP}$) or the canonical (C2 effector memory, C4 $P_{EX}$, C4 $T_{EX}$) TIL states (Fig. 8e). Among them, G7 was predominantly active in C7/memory-like cells, and G9 was active in all three synthetic cell states where IL-2v was implicated (Fig. 8g). Orthogonal partial least-squares discriminant analysis (OPLS-DA)[43] revealed TFs from G7 as having the highest statistical power to discriminate between C7 TILs and other states (Extended Data Fig. 7b). Similarly, G9 contained TFs with the highest discriminant power for the three synthetic cell states driven by IL-2v relative to canonical TIL states (Extended Data Fig. 7c). Functional annotation of the target genes of the top 25% most discriminant TFs/regulons of G7 revealed that this set is involved in the regulation of RNA metabolism, mRNA splicing, ribosome biogenesis, and so on, while the target genes of the top 25% most discriminant TFs/regulons of G9 were involved also in the regulation of protein metabolism and translation, in addition to RNA metabolism (Fig. 8f and Extended Data Fig. 7d,e), revealing how IL-2v signaling positions the cell in a memory-like state of transcriptional/translational readiness, while progression to C6 $T_{SP}$ and C5 $T_{SE}$ engages further programs of synthesis to enter the full effector state. Notably, 50% of these TFs (14/28) formed a highly connected subnetwork motif in the SCENIC-inferred gene regulatory network (Fig. 8g), revealing coordinated transcriptional regulation. Thus, IL-2v engineered into CD8[+] T cells, transferred in the absence of lymphodepletion, can regulate two different sets

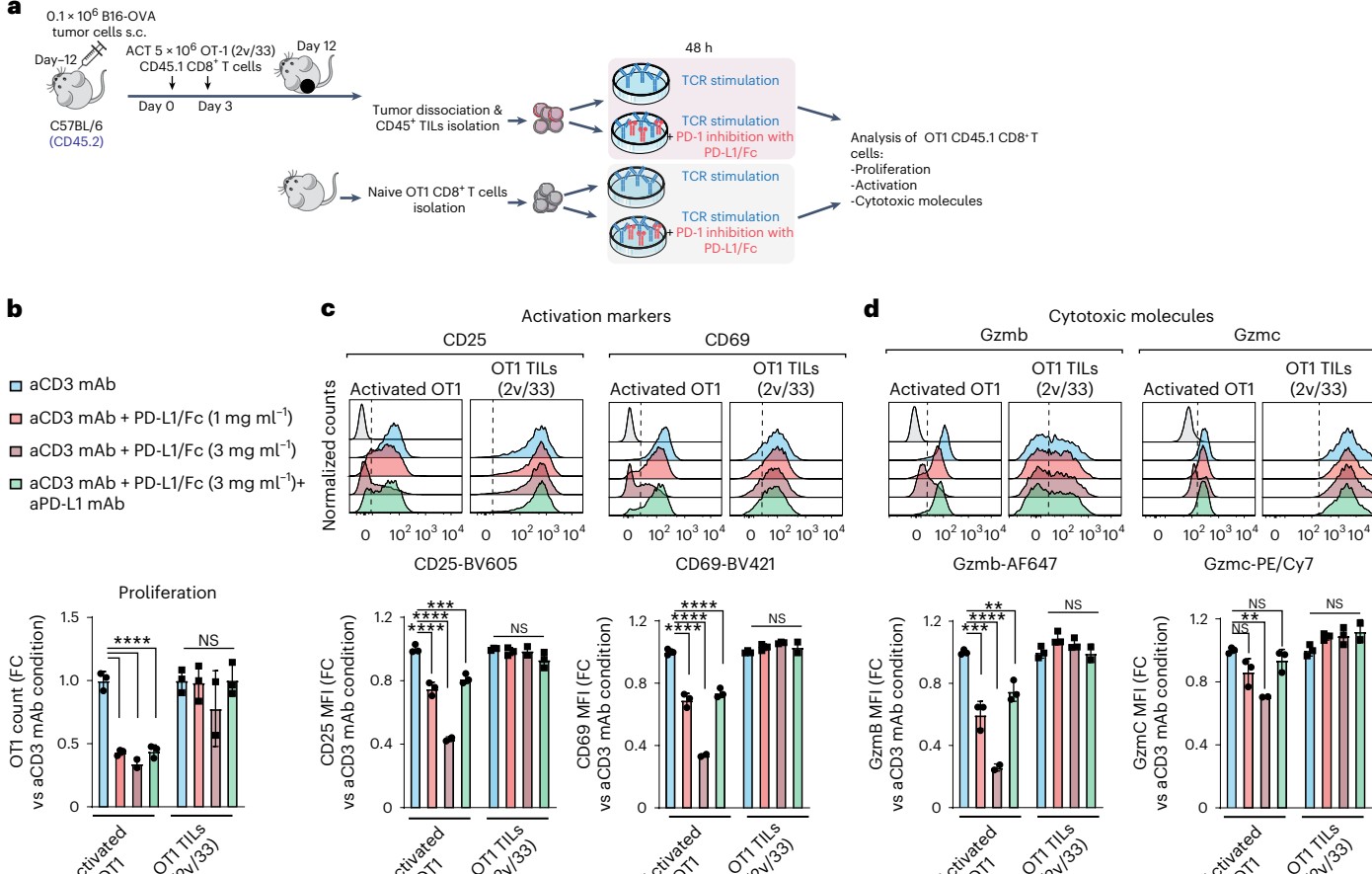

**Fig. 6 | Although PD-1⁺, C5 T_SE cells are not functionally restrained by PD-1.**
**a**, Experimental design to assess the functional restriction of OT1 TILs by plate-bound PD-L1 (ref. 37) upon TCR restimulation ex vivo. Genetically engineered IL-2v/IL-33 OT1 TILs were adoptively transferred into C57BL/6 mice bearing B16-OVA tumors. Total CD45⁺ TILs were then magnetically isolated (Miltenyi Biotech) from tumors 12 d after the first ACT infusion and stimulated for 48 h ex vivo with plate-bound anti-CD3 (3C11, eBiosciences) alone or in the presence of chimeric mouse PD-L1/Fc (1 or 3 µg ml⁻¹). In addition, 20 µg ml⁻¹ of anti-mouse PD-L1 monoclonal antibody (clone 10F.9G2, BioXcell) was added to the cultures as a control.

Moreover, naive OT1 cells were cultured under identical conditions as the control for PD-1 inhibition of T-cell activation. s.c., subcutaneous.
**b**–**d**, Normalized number of cells (**b**), and normalized expression of activation (**c**) and effector molecules (**d**) relative to anti-CD3-mediated stimulation. Data are from three independent experiments (*n* = 3 internal replicates per experiment). A Brown–Forsythe and Welch ANOVA test combined with Dunnet's test were used to correct for multiple comparisons relative to anti-CD3 stimulation alone; *P < 0.05, **P < 0.01, ***P < 0.001, ****P < 0.0001. mAB, monoclonal antibody; MFI, mean fluorescence intensity. FC, fold change.

of IL-2R-related TFs: a first group (encoded by *Fosl1*, *Egr1*, *Tbx21*, *Nfkb1*, *Irf4* and *Myc*) specifically active in C7/memory-like synthetic TILs in the absence of IL-33, where they sustain cell expansion and persistence, stemness and memory/readiness, while a second group (encoded by *Irf8*, *Atf3*, *Atf4*, *Smarca4* and *Batf3*) active across all three synthetic states, seems involved in the metabolic support of sustained gene expression and protein translation, both highly energetically costly cellular processes[44], which are dampened in canonical exhausted CD8⁺ T cells[45]. Strikingly, we detected important target-gene overlap between G7 (*Egr1*, *Tbx21*, *Nfkb1*, *Irf4*, *Myc*) and TFs in G9 (*Irf8*, *Atf3*, *Batf3*; Fig. 8h), explaining how the alternate cell fates C7 memory-like versus C6 T_SP/C5 T_SE can be structured to share important downstream core metabolic programs through alternate TF regulatory networks, which ensures their respective differentiation. Collectively, we conclude that IL-2v alone contributes to reprogram genetically engineered tumor-specific TILs to escape TOX⁺ canonical exhaustion, endows them with superior metabolic transcriptional programs and positions them in an activated, memory-like state after ACT, which can persist in the absence of TOX. This is a new and unexpected finding, as TOX expression is required for persistence under similar conditions[26–28,30].

## Tumor progression is not associated with canonical T-cell exhaustion

Although orthogonally engineered T cells appeared committed to a fate that deviates from canonical exhaustion, we asked whether these cells could eventually capitulate to exhaustion in tumors. We looked for such cells in tumors that resumed growth following regression, expecting these to be the most likely environment for identifying failing TILs. Strikingly, OT1 cells were not exhausted in progressing tumors, but rather remained TOX-negative C7 memory-like cells (Extended Data Fig. 8a). Such results were observed both when we mixed equal proportions of IL-2v and IL-33-transduced OT1 cells, or when we transferred cells transduced with a bi-cistronic IL-2v/IL-33 vector (Extended Data Fig. 8b) or with a dimeric IL-2v–IgG4 (to strengthen IL-2R signaling) and IL-33, conditions in which the secretion of both cytokines was secured in OT1 cells (Extended Data Fig. 8c). We next asked whether lack of differentiation of OT1 cells to T_SE in escaping tumors was caused by lack of antigen. Indeed, tumor escape variants downregulated the surface expression of MHC class I as early as 12 d after ACT (Extended Data Fig. 8d).

Lastly, we theorized that if we strengthened T-cell attack upon ACT, T_SE cells could eradicate tumors before the latter could adapt. Given that

IL-2v also drove an absolute increase in CD4$^+$ T$_{reg}$ cells (Extended Data Fig. 8e), we conducted orthogonal ACT in mice previously depleted of CD4$^+$ cells, in the absence of other lymphodepletion. We observed a 25% cure rate in these mice, illustrating the potential efficacy of C5 T$_{SE}$ cells in eradicating immune-resistant tumors in otherwise immunocompetent hosts (Extended Data Fig. 8f).

## Discussion

We provide evidence that orthogonal combinatorial T-cell engineering reprograms tumor-specific TILs away from canonical TOX$^+$ exhaustion, thereby overcoming homeostatic barriers to engraftment and leading to tumor regression. Although two recent studies have shown the association of novel TOX$^{low}$ cell states with successful combinatorial immunotherapies[35,40], to the best of our knowledge, this is the first report of effector CD8$^+$ cells directed against tumors that, thanks and exclusively to novel T-cell gene engineering principles, fully downregulate TOX and become freed up from the mandatory fate of exhaustion upon chronic activation.

Here, we successfully combined two main secreted components, IL-2v binding to βIL-2R and γIL-2R but not αIL-2R, and IL-33. This combination brought about a unique effector synthetic state in the adoptively transferred genetically engineered T cells, which is distinct from canonical TOX$^+$ terminal exhaustion. This state was associated with marked local CD8$^+$ T-cell intratumoral expansion, potent effector function and tumor control, and was characterized by the expression of multiple granzymes and significant downregulation of several TFs involved in CD8$^+$ TIL dysfunction[25,34]. Notably, this effector state was distinguished by the expression of inhibitory receptors but absent expression of TOX—the pivotal TF for the generation and maintenance of exhausted CD8$^+$ T cells during chronic viral infection and in cancer[26–30]—indicating that PD-1 and other co-inhibitory receptors can be upregulated in the context of sustained antigen stimulation independently of TOX, unlike in canonical exhausted CD8$^+$ T cells[26–30]. Notably, TILs in the synthetic T$_{SE}$ effector state were not constrained functionally by PD-1, effectively showing that cells were liberated from natural exhaustion constraints. Finally, this synthetic effector state diverged from canonical effector states seen during infection[31,46]. In fact, a query of public human and mouse TIL data revealed that the generated synthetic states were never reported to date. Taken together, we speculate that both T$_{SP}$ and T$_{SE}$ CD8$^+$ TILs represent the precursor and effector states, respectively, of a novel PD-1$^+$TOX-indifferent CD8$^+$ T-cell synthetic differentiation program.

The therapeutic manipulation of TOX has been proposed as a promising strategy to abrogate T-cell exhaustion in the context of cancer[47]. This notion is supported by the improved functionality of CAR-T cells upon TOX knockdown or deletion of TOX2 (ref. 30), the strengthening of antitumor T-cell responses reported in heterozygous deletion of TOX[27] and more recently by the combination of PD-1 blockade with either wild-type IL-2 or a different non-alpha IL-2v[35,40]. In this study, we show a new approach to reach synthetic cell states in which TOX is fully transcriptionally suppressed in more than 90% of the adoptively transferred CD8$^+$ TILs without genome editing or knockdown. IL-2v was primarily responsible for promoting downregulation of TOX in the transferred cells. How IL-2 signaling directed through the IL-2Rβγ receptor chains achieves suppression of TOX will require additional investigation. We speculate that lack of TOX expression leads to a major epigenetic reprogramming in chronically stimulated synthetic T cells, since TOX is responsible for the chromatin remodeling necessary for the commitment to canonical exhaustion[26,27], thus allowing chromatin accessibility in regions that would be otherwise inaccessible during chronic TCR stimulation. Notably, a similar IL-2R-driving reprogramming of TOX$^+$ exhausted CD8$^+$ T cells was just recently reported[35,40]; however, this was not described when orthogonal IL-2 or IL-9 cytokine–receptor complexes were used to engineer T cells[48,49].

The fact that IL-2v is associated with a high level of intratumoral persistence of TOX$^{neg}$PD-1$^+$ CD8$^+$ T cells in the context of sustained antigen stimulation is also a new and remarkable finding, because TOX expression is required for persistence under persistent stimulation conditions[26–28,30]. Regulon analysis[42] suggested that IL-2v regulates this process by activating a TF network motif involved in the metabolic support of sustained transcription and protein translation, both highly energetically costly cellular processes[44], which are dampened in canonical exhausted CD8$^+$ T cells[45]. Indeed, the TF *Fosl1* has been associated with long-term persistence of CD8$^+$ T cells in the context of chronic viral infections[50], while *Batf3* encodes a critical TF for CD8$^+$ memory formation and survival[51]. Thus, IL-2v alone contributes to reprogram genetically engineered tumor-specific TILs to escape TOX$^+$ canonical exhaustion, endows them with superior metabolic transcriptional programs and commits them to an activated, memory-like state after ACT, which can persist without TOX. Interestingly, this synthetic memory-like cell state was maintained, unlike canonical memory, even in the presence of TCR engagement, but it was not enough to effectively control tumors, confirming previous observations with *TOX*-knockout TILs[26].

IL-33 has never been used in the context of genetically engineered tumor-specific T cells; however, its simultaneous expression in vivo was required to drive Tcf1 suppression, allowing OT1 cells to exit the memory-like/stemness-associated state and differentiate to polyfunctional PD-1$^+$TOX$^{neg}$ effector cells with direct antitumor potential. Because OT1 TILs did not express the IL-33 receptor ST2, the contribution of secreted IL-33 must be CD8$^+$ T-cell extrinsic, likely due to its ability to reprogram the TME[12] and activate tumor-associated dendritic cells to restore their cross-priming potential[12]. The combination drove a stable synthetic effector state that persisted in association with tumor

**Fig. 7 | Orthogonal engineering induces a novel T-synthetic precursor-like cell state. a**, Analysis of Gzmc expression in CD45.1$^+$ OT1 cells collected at day 12 after transfer from either TDLNs or spleen. **b**, Violin plots showing expression of *Pdcd1* in clusters. Dot plot showing the expression of important stemness-related markers in clusters. **c**, Comparison of C6 T$_{SP}$ OT1 cells from days 8 and 12 (response) with canonical P$_{EX}$-like cells from the TIL reference map[21]. Calculation was performed with the function ProjecTILs find.discriminant. genes using a two-sided non-parametric Wilcoxon rank sum test with Bonferroni correction. FC, fold change. **d**, Reactome pathway overrepresentation test. **e**, GO biological process GSEA. **f**, GSEA gene sets: CD8$^+$ T-cell gene signatures of PD-1-mediated inhibition[37]; ranked list: DEGs between T$_{SP}$ TILs and canonical P$_{EX}$ from the TIL reference map[21]. GSEA was performed with the GSEA function (clusterProfiler), which uses the Benjamin–Hochberg method for multiple correction. Reactome pathway enrichment was performed with the function enrichReactome from the same package, which uses a one-sided version of the Fisher's exact overrepresentation test to find enriched categories. **g**, Trajectory and pseudotime inference analysis of OT1 TILs collected from all studied time points after orthogonal ACT using the Paga-Tree algorithm implemented in

dynverse[54]. The inferred trajectory and the expression of relevant markers were visualized using diffusion maps. **h**, Analysis of TOX and Gzmc expression in PD-1$^+$TCF1$^+$ OT1 TILs from orthogonal ACT relative to canonical P$_{EX}$ cells recovered from baseline tumors ($n = 6$ mice). OT1 TILs from mice treated with orthogonal ACT were recovered at day 8 (two independent experiments, $n = 4$ mice per group) and at day 12 (five independent experiments, $n = 5–6$ mice per group). **i**, Analysis of Gzmc expression in CD44$^+$PD-1$^+$TCF1$^+$ OT1 TILs recovered at day 12 after ACT from mice treated with PD1d/IL-2v, PD1d/IL-33 or PD1d/IL-2v/IL-33. Data are from three independent experiments; $n = 8$ mice per experiment for PD1d/IL-2v/IL-33 and $n = 4–5$ mice per experiment for PD1d/IL-2v and PD1d/IL-33). Data are presented as mean values ± s.d. A representative dot plot is shown. One-way ANOVA in combination with Dunnet's test to correct for multiple comparisons was used; *$P < 0.05$, **$P < 0.01$, ***$P < 0.001$, ****$P < 0.0001$. **j**, Analysis of TCF1 and PD-1 expression in CD45.1$^+$ OT1 cells collected at day 12 after transfer from TDLNs. **k**, Projection of mouse postnatal day 14 (P14) CD8$^+$ T cells collected following systemic administration of anti-PD-1 and IL-2 (ref. 40), onto the OT1/endogenous transcriptomic space. Contour plots depict the clusters covered by each cell dataset and bar plots show the cluster composition.

control. Indeed, tumor progression was not associated with evolution toward T-cell exhaustion or other kind of CD8[+] T-cell hyporesponsiveness states, but rather with loss of antigen presentation, a common

escape mechanism of melanoma tumor cells to immunotherapy[52] as well as with activation of counteracting CD4[+] T-cell-mediated regulatory mechanisms, likely regulatory T cell, in the TME. Overall, our

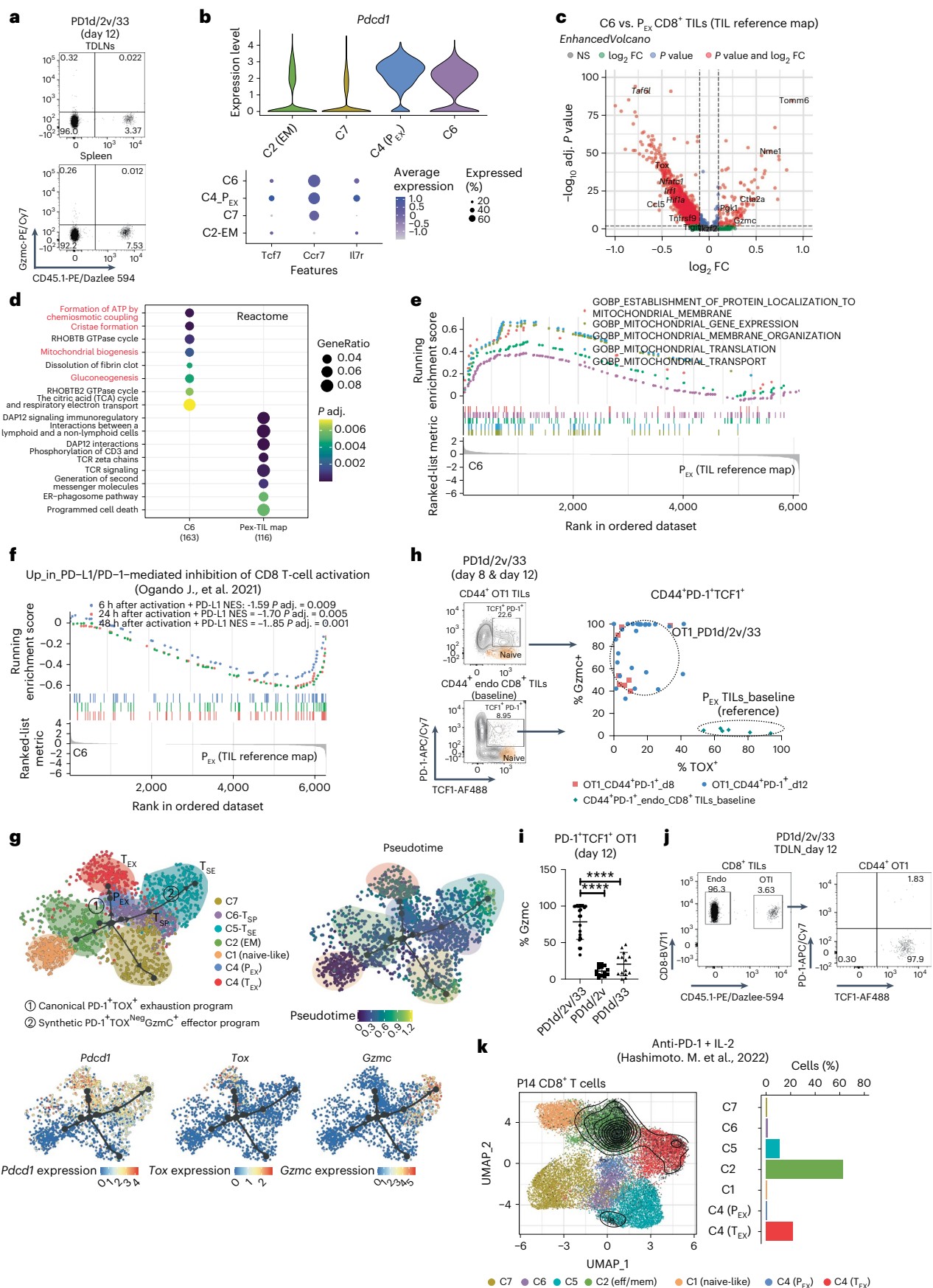

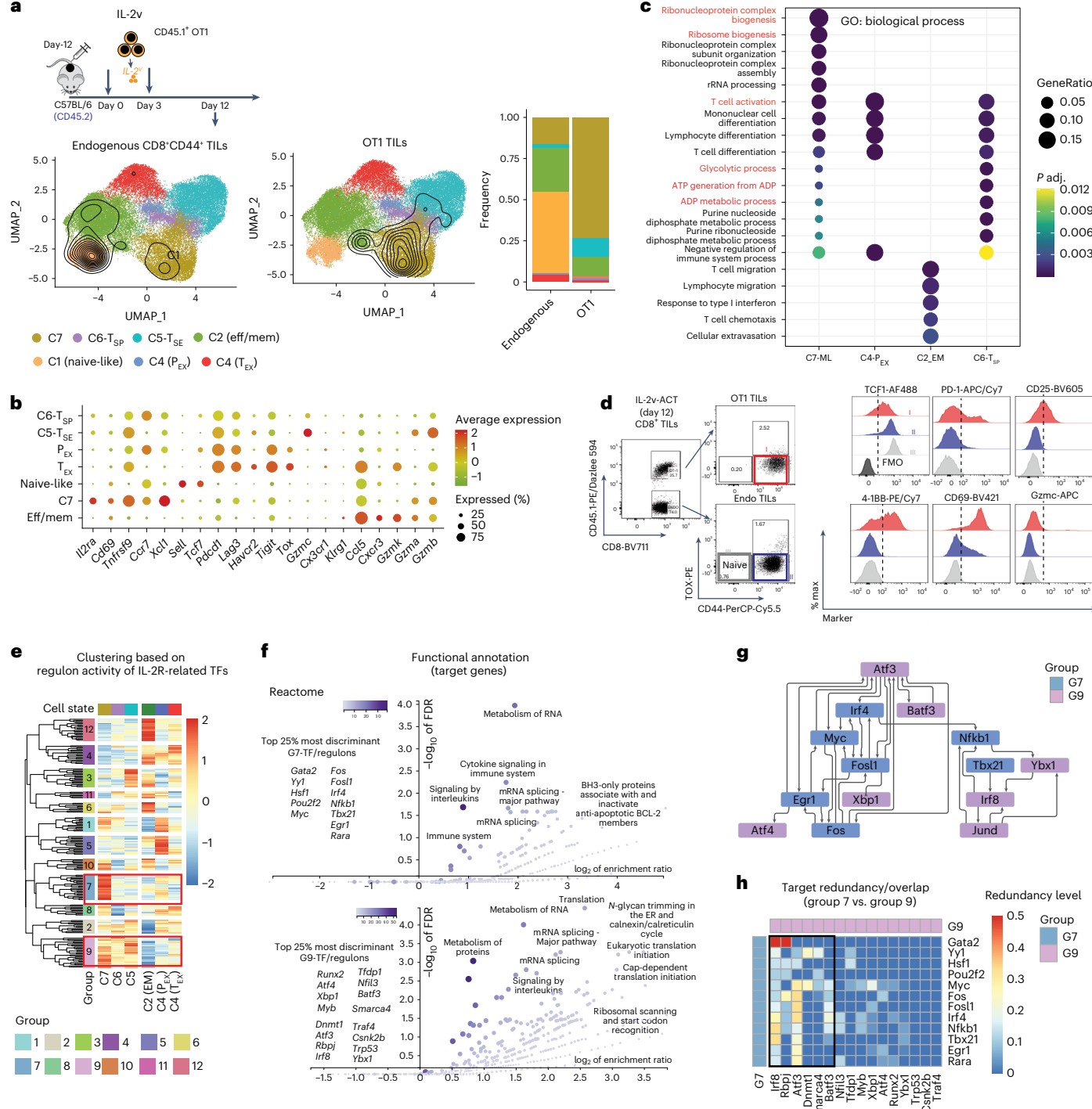

**Fig. 8 | IL-2v engineering contributes to key programs to the T_SE state. a**, UMAP plot showing the cluster distribution of either OT1 TILs or endogenous CD8+ TILs recovered at day 12 after ACT from mice treated with IL-2v-secreting OT1 cells. Contour plots depict the clusters covered by each cell compartment. Bar plots show the cluster composition of each compartment. **b**, Dot plot showing C7-specific markers. **c**, GO biological process overrepresentation test of genes specifically upregulated in each memory-like subset. The calculation was performed with the function enrichGO (ClusterProfiler), which uses a one-sided version of the Fisher's exact overrepresentation test to find enriched categories. **d**, Phenotypic validation at the protein level of some C7-specific markers in OT1 TILs recovered at day 12 after ACT from mice treated with IL-2v OT1 cells. **e**, Unsupervised clustering analysis based on the regulon activity across

canonical and synthetic T-cell states of 181 TFs downstream of the IL-2R. **f**, Reactome pathway overrepresentation test using the targets genes of the top 25% most discriminant TFs of G7 and G9. The target list was analyzed for enrichment GO terms and Reactome pathways using the online tool WebGestalt (http://www.webgestalt.org/). **g**, Network connectivity analysis. To test whether the number of regulatory interactions between a given set of TFs was significantly higher than expected by chance from a random selection of TFs, a connectivity analysis was performed using the SANTA algorithm[55] on the gene regulatory network derived from the regulon analysis. **h**, Target redundancy for two given TFs was calculated as the ratio between the number of common target elements in each regulon and the size of the smallest regulon (Methods).

work demonstrates the potential for clinical translation of innovative combinatorial engineered T cells for reprogramming the TME in the immunocompetent host and inducing highly functional and novel, non-exhausted synthetic CD8[+] states endowed with the ability to control advanced, poorly immunogenic and PD-1-resistant solid tumors.

## Online content

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

## Methods

### Mice and cell lines

Six-week-old female C57BL/6 mice were purchased from Harlan and housed at the animal facility at the University of Lausanne in compliance with guidelines. C57BL/6 OT1 CD45.1[+] and C57BL/6 Pmel were obtained from P.R.'s laboratory (UNIL). All in vivo experiments were conducted in accordance and with approval from the Service of Consumer and Veterinary Affairs (SCAV) of the Canton of Vaud, Switzerland, under the license VD3526.

The MC38-OVA tumor cell line was obtained from P.R.'s laboratory (UNIL). B16-F10 (CRL-6475) tumor cell lines were purchased from the American Type Culture Collection and transduced with a retrovirus to express ovalbumin (B16-OVA). Both were grown as a monolayer in DMEM supplemented with 10% FCS, 100 U ml$^{-1}$ of penicillin and 100 µg ml$^{-1}$ streptomycin sulfate. The Phoenix Eco retroviral ecotropic packaging cell line, derived from immortalized normal human embryonic kidney cells was obtained from the American Type Culture Collection (CRL-3214) and maintained in RPMI 1640-Glutamax medium supplemented with 10% heat-inactivated FBS, 100 U ml$^{-1}$ penicillin and 100 µg ml$^{-1}$ streptomycin sulfate.

### Design of bi-cistronic expression cassettes

The retroviral vector pMSGV1 was used as the backbone for all the constructs. Expression cassettes generically comprised the signal peptide of a murine IgG kappa chain region V-III MOPC 321, followed by the gene string encoding the N-terminal ectodomain of murine PD-1 fused to human IgG4 Fc, referred to here as PD-1 Ig4 decoy (PD1d). The restriction sites AgeI and EcoRI flanked this first part at the 5′ and 3′ ends, respectively. The second part followed the T2A sequence and was composed of the signal peptide of murine IFN-β followed by a gene string encoding one of the following molecules: murine IL-33 and IL-2v (residues: Ala21–Thr153; mutations: p.Arg58Ala, p.Phe62Ala, p.Tyr65Ala, p.Glu82Ala and p.Cys145Ser). The restriction sites MluI and SalI flanked this second part at the 5′ and 3′ ends, respectively. To generate bi-cistronic vector, a gene string encoding IL-2v or IL-2v IgG4 Fc replaced the coding sequence for PD-1 in the PD-1 IgG4–T2A–IL-33 construct, by restriction cloning. All gene strings were murine codon optimized and synthesized by GeneArt AG, and all constructs were fully sequenced by Microsynth AG after cloning in the MSGV vector.

### Murine T-cell transduction and expansion

Replication-defective retrovirus was produced and concentrated as described in ref. 56. Primary murine OT1 cells were isolated from single-cell suspensions of dissociated spleens from CD45.1[+] congenic OT1 C57BL/6 mice aged 6–10 weeks using the Pan T-cell Isolation Kit II for mouse, and cultured in RPMI 1640-Glutamax medium supplemented with 10% heat-inactivated FBS, 100 U ml$^{-1}$ penicillin, 100 µg ml$^{-1}$ streptomycin sulfate, 1 mM pyruvate, 50 µM beta-mercaptoethanol and 10 mM non-essential amino acids (T-cell medium). Retroviral transduction was done following the protocol described in ref. 56. Seven days after transduction, the expression of the molecules was assessed by intracellular flow cytometric analysis and their presence in the supernatant by ELISA. Finally, the cultures were maintained at a cell density of 0.5–1 × 10$^6$ cells per ml, replenished with fresh T-cell medium every other day until day 15 following an in vitro expansion protocol optimized in our laboratory to generate CD44$^+$CD62L$^+$TCF1$^+$ central memory CD8$^+$ T cells[56].

### Flow cytometric analysis to evaluate expression of immunomodulatory factors by genetically engineered T cells

OT1 T cells were incubated with 50 µl of Live/Dead Fixable aqua dead (Thermo Fisher; 1:300 dilution) for 30 min in PBS at 25 °C, washed and then incubated again with 50 µl of FCR blocking reagent (clone 2.4G2, BD Biosciences; 1:50 dilution) for 30 min at 4 °C. Cells were washed and incubated at 4 °C for another 30 min with surface markers directed with PE/Cy5.5 anti-mouse CD3e (clone 2. 145-2C11; 1:100 dilution),

BV711 anti-mouse CD8α (clone 2. 53-6.7, BioLegend; 1:50 dilution) and PE/Dazlee anti-mouse CD45.1 (clone A20, BioLegend; 1:100 dilution). For intracellular staining, the following antibodies were used: PE anti-human hIgG4-Fc (clone HP6025, Abcam; 1:50 dilution) for detecting the PD-1 IgG4 decoy and PE anti-mouse IL-33 (clone 396118, Invitrogen; 1:25 dilution). After surface staining, genetically engineered OT1 cells were washed twice and fixed/permeabilized using the Foxp3 TF staining buffer set (Thermo Fisher Scientific; 1:4 dilution). Cells were washed and resuspended in PBS supplemented with 2% BSA and 0.01% azide (FACS buffer) to be acquired with a BD flow cytometer LSR II cytometer with FACSDiva software v8.0.1 (BD) and analyzed using FlowJo software v10.7. Because Pd1d was a common module, we used it to evaluate the transduction efficiency across conditions. Next, we added UT OT1 cells to get the same number of transduced cells in all the groups.

### ELISA for evaluating the secretion of immunomodulatory factors by genetically engineered T cells

One week after transduction, 10$^6$ genetically engineered OT1 T cells were seeded in a 24-well plate in 1 ml of serum-free RPMI medium at 37 °C for 72 h. Then, the supernatant was harvested and tested for each molecule by ELISA. PD-1 IgG4 homemade-ELISA: coating antibody: anti-mouse PD-1 (R&D, AF1021, final concentration of 2 µg ml$^{-1}$; BioLegend, 421701). Detection antibody: anti-hIgG4-HRP (clone HP6023, Abcam; 1:2,000 dilution). IL-2v homemade ELISA: coating antibody: purified polyclonal anti-human IL-2 antibody (R&D, final concentration of 3 µg ml$^{-1}$ in coating buffer, BioLegend, 421701). Secondary antibody: biotin polyclonal anti-human IL-2 antibody (Invitrogen; 1:500 dilution) and streptavidin-HRP (BioLegend; 1:1,000 dilution). Supernatants from OT1 T cells transduced for expressing either the fusion molecule TIM-3–IgG4 or IL-3 variant were used as negative controls. For detection of IL-33, a commercial ELISA kit was used (LEGEND MAXTM mouse IL-33 ELISA kit, BioLegend).

### Adoptive cell transfer and tumor evaluation

B16-OVA tumor cells were collected with 0.05% trypsin, washed and resuspended in PBS for injection. Then, 1 × 10$^5$ tumor cells were injected subcutaneously in the right flanks of C57BL/6 mice, aged 7 weeks. On day 11 (average tumor volume, 100–200 mm$^3$), mice were randomized to have comparative average tumor volumes between experimental arms, which were adjusted to have between 4 and 10 mice depending on the final number of cells recovered in each experimental group upon 12 d of in vitro expansion. Thus, no statistical methods were used to predetermine sample size, but our sample sizes are similar to those reported in our previous publication[57].

Next, on days 12 and 15, mice were treated with intravenous transfer of 5 × 10$^6$ genetically engineered CD44$^+$CD62L$^+$TCF1$^+$OT1 T cells or control non-transduced OT1. Mice were monitored three times a week, and tumor length ($L$; greatest longitudinal measurement) and width ($W$; greatest transverse measurement) were measured with a caliper by an independent investigator in a blinded manner. Tumor volumes ($V$) were calculated using the formula: $V = (L × W^2)/2$. The average tumor volumes per group are plotted ± s.d. (standard deviation). Mice were euthanized once tumors reached 1,000 mm$^3$, or, according to regulation, if they became distressed, moribund or the tumor became necrotic. Except for the antitumor efficacy experiments, mice were monitored three times a week by an independent investigator in a blinded manner.

### Immune subset depletion and checkpoint blockade

CD4$^+$ T cells were depleted by administering 250 µg per dose of depleting antibody (InVivoMAb anti-mouse CD4, clone GK1.5, BioXcell) i.p. every 3 d beginning 1 d before therapy. For checkpoint blockade, mice were injected i.p. every 3 d with 250 µg per dose of α-mouse PD-L1 (InVivoMAb anti-mouse PD-L1, clone 10F.9G2, BioXcell) and α-mouse TIM-3 (InVivoMAb anti-mouse TIM-3, clone RMT3-23).

## Preparation of single tumor-cell suspensions, antibodies for flow cytometry and ex vivo restimulation for cytokine production

Tumors were excised and dissociated into a single-cell suspension by combining mechanical dissociation with enzymatic degradation of the extracellular matrix using the commercial tumor dissociation kit for mouse (Miltenyi, 130-096-730). Following single-cell suspension, $2.5 \times 10^6$ live cells were seeded in 96-well plates and incubated with 50 µl of Live/Dead Fixable Aqua Dead cell stain kit for 30 min in PBS at 25 °C. Then, Fc receptors were blocked by incubation for 30 min at 4 °C with 50 µl of purified CD16/CD32 monoclonal antibody. Cells were then stained for 30 min at 4 °C with the fluorochrome-conjugated monoclonal antibodies of interest (Reporting Summary) in 50 µl of FACS buffer. Subsequently, the cells were washed twice and fixed/permeabilized using the FoxP3 TF staining buffer set for intracellular staining. Analysis of stained cells was performed using an LSR II cytometer and FlowJo software. Fluorescence-minus-one controls were stained in parallel using the panel of antibodies with sequential omission of one antibody. Isotype control was used for granzyme C staining (BioLegend, 400912 and 400922). Precision count beads (BioLegend, 424902) were used to obtain absolute counts of cells during acquisition on the flow cytometer. A BD FACSAria III instrument or a BD FACSAria II instrument was used for cell sorting. Sorted samples had purity > 95% as confirmed by resampling after sorting.

To stain the ST2 surface receptor, a PE-conjugated anti-ST2 antibody was used (rat anti-mouse anti-ST2; clone DJ8 MD Bioproducts, 101001PE; 1:50 dilution). The PE signal was amplified using the Faser-PE kit (Miltenyi, 130-091-764) following the manufacturer's recommendations. A PE-conjugated rat IgG1, λ isotype control antibody (BioLegend, 401906; 1:50 dilution) was used as a negative control for the staining.

For the detection of cytokine production, a single tumor-cell suspension ($2.5 \times 10^6$ live cells) was in vitro restimulated in 24-well plates with SIINFEKL peptide (EMC Microcollections; final concentration, 10 ng ml$^{-1}$) for 4 h in the presence of brefeldin A (Sigma, SML0700; final concentration, 5 µg ml$^{-1}$) at 37 °C. Cells were surface stained before fixation and permeabilization as described above, which were followed by intracellular staining.

## Immunofluorescence labeling and microscopy

For immunohistochemistry analysis, tumor tissues were isolated and fixed in 1% paraformaldehyde in PBS overnight, infiltrated with 30% sucrose the next day (overnight) and then embedded and frozen in OCT compound. Cryostat sections were collected on Superfrost Plus slides (Fisher Scientific), air-dried and pre-incubated with blocking solution containing BSA, normal mouse serum, normal donkey serum (Sigma) and 0.1% Triton. Then, they were labeled overnight at 4 °C with primary antibodies diluted in PBS with 0.1% Triton. After washing with PBS and 0.1% Triton, the secondary reagents were diluted in PBS with 0.1% Triton and applied for 45 min at room temperature. Finally, after additional washing with PBS and 0.1% Triton, DAPI (Sigma) was used to stain the nuclei followed by a PBS wash and mounting in homemade DABCO. Images were acquired with a Zeiss Axio Imager Z1 microscope and an AxioCam MRc5 camera and treated using Fiji (National Institutes of Health) or Adobe Photoshop. Exposure and image processing were identical for mouse groups, which were directly compared.

## In vitro exhaustion assay

This assay was done as described in ref. 58. Briefly, naïve CD8$^+$ OT1 T cells were purified from the spleens of OT1 transgenic mice by negative selection with magnetic beads. In each well of a 24-well plate, $5 \times 10^5$ of the purified CD8$^+$ T cells per milliliter were cultured at 37 °C in complete mouse T-cell medium with 5 ng ml$^{-1}$ of IL-15 (Miltenyi, 130-095-766) and IL-7 (Miltenyi, 130-095-361) and 10 ng ml$^{-1}$ of OVA$_{(257-264)}$ peptide, which was daily added to the culture for 7 d.

## Winn-type assay

The in vivo antitumor potential of C5-like OT1 TILs was evaluated using a Winn-type assay[53]. First, tumors were excised 12 d after ACT (day 24 after tumor inoculation) and dissociated into a single-cell suspension by combining mechanical dissociation with enzymatic degradation of the extracellular matrix using the commercial tumor dissociation kit for mouse. Next, CD8$^+$CD45.1$^+$ OT1 TILs were purified by FACS. These cells were mixed with fresh B16-OVA tumor cells that were kept in culture for 3 d in a 2:1 (T cell:tumor) ratio. Next, $3 \times 10^4$ total cells ($2 \times 10^4$ OT1 + $10^4$ B16-OVA cells) were subcutaneously injected into the right flank of each mouse. Mice were monitored three times per week, and tumor volumes were calculated as described above. As negative controls, we used in vitro-generated bona fide exhausted CD8$^+$ T cells[58] that were harvested after 7 d of repeated peptide stimulation. These cells were mixed with fresh B16-OVA tumor cells using the same ratio as above, and the mix was subcutaneously injected into the right flank of each mouse.

## In vitro killing assay

In vitro tumor killing assays were performed using the IncuCyte ZOOM imaging platform (Essen Bioscience). Tumor cells were plated in 96-well plates 5 h before the co-culture with T cells. Briefly, $10^4$ B16-OVA NucLight Red cells were co-cultured with either bona fide in vitro-generated exhausted OT1 TILs or purified CD8$^+$OTI TILs isolated at day 24 from mice treated with orthogonal ACT at a 2:1 (effector:target) ratio. Tumor cells alone were used as a negative control of cell death. Images were captured every 2 h for 72 h at 37 °C, and red living target cells were quantified with the IncuCyte ZOOM integrated analysis software.

## PD-1-mediated inhibition of CD8$^+$ T-cell activation

Genetically engineered OT1 T cells (IL-2v/IL-33) were adoptively transferred twice, for a total of $10^7$ cells, into C57BL/6 mice bearing B16-OVA tumors. Total CD45$^+$ TILs were then magnetically isolated (Miltenyi Biotech) from tumors 12 d after ACT infusion and stimulated at 37 °C for 48 h ex vivo with plate-bound anti-mouse CD3 (monoclonal antibody 17A2, eBioscience, 14-0032-82; final concentration, 1 µg ml$^{-1}$) alone or in the presence of chimeric mouse PD-L1/Fc (1 or 3 µg ml$^{-1}$; 1019-B7-100 R&D) inhibition. In addition, 20 µg ml$^{-1}$ of anti-mouse PD-L1 monoclonal antibody (clone 10F.9G2, BioXcell) was added to the cultures as a control. Moreover, naive OT1 cells were cultured under identical conditions as the control for PD-1 inhibition. Subsequently, cells were stained and analyzed by flow cytometry.

## Heat maps

The percentage of OT1 and endogenous CD8$^+$ TILs expressing effector molecules (TNF, IFN-γ, GzmB, GzmA, Prf1, Ki67) was determined by FACS as explained above ('Preparation of single tumor-cell suspensions, antibodies for flow cytometry and ex vivo restimulation for cytokine production'). The values in each row were scaled to the mean. Heat maps and clustering trees were generated in R v4.1 using the pheatmap function from the namesake package. Euclidean distance was used for hierarchical clustering using the 'ward' method.

## 1st single-cell RNA-seq analysis (related to Fig. 2)

CD45$^+$ TILs were isolated from pooled tumors (five mice per condition) and sorted at days 5 and 12 after cell transfer and sequenced using 10x Genomics. Cluster generation was performed with the Illumina HiSeq 4000 PE Cluster Kit reagents. Sequencing was carried out on the Illumina HiSeq 4000 with HiSeq 4000 SBS Kit reagents. The sequencing data were demultiplexed with the bcl2fastq conversion software (v2.20, Illumina). The resulting FASTQ files were processed with the Cell Ranger count using default settings for 5′ RNA gene expression analysis (Cell Ranger v4.0.0, 10x Genomics).

The aggregated unique molecular identifier (UMI) counts matrix generated by Cell Ranger was filtered to select high-quality CD8$^+$ TIL transcriptomes. First, cells were filtered with the following criteria:

500 to 6,000 detected genes; 2,000 to 40,000 UMI counts; ribosomal protein content between 5% and 60%; and mitochondrial content below 10%. Next, CD8[+] T cells were isolated using the scGate package[59], resulting in 3,216 high-quality CD8[+] TIL transcriptomes. For dimensionality reduction, we first identified 1,000 highly variable genes (HVGs) using the vst method from Seurat[60], excluding ribosomal, mitochondrial, heat-shock and TCR genes, as well as genes associated to cell cycling[61]. Standardized HVGs were used for (1) dimensionality reduction using principal component analysis (PCA), and (2) uniform manifold approximation and projection (UMAP) on the first 30 principal components, with other parameters by default. Unsupervised clustering was performed using the FindNeighbors method implemented in Seurat with default parameters and FindClusters with resolution of 0.3. For supervised analysis and comparison with previous annotations, the scRNA-seq data were projected into a reference atlas of TILs using ProjecTILs[21]. Differentially expressed genes (DEGs) between clusters C4 and C5 were assessed using the FindAllMarkers function from Seurat and MAST (v1.10)[62] with parameters min.pct = 0.05 and logfc.threshold = 0.25. DEGs were visualized as volcano plots using the R package EnhancedVolcano (https://github.com/kevinblighe/EnhancedVolcano) with a fold-change cutoff = 1 and minimum $P$ value = $10^{-5}$. Gene-set enrichment analysis (GSEA) of these clusters versus $TOX$-knockout signature (downloaded from Supplementary Table 1 in ref. 26) was calculated using the GSEA function from clusterProfiler (v3.12)[63] with default parameters and using the top 200 differentially expressed cluster genes with $P$ value < 0.01 ordered by decreasing fold change.

### 2nd single-cell RNA-seq analysis (related to Fig. 3)

OT1 and endogenous CD44[+] CD8[+] TILs were independently sorted from pooled tumors (five mice per condition) at baseline (12 d after tumor inoculation) or at days 5, 8, 12, 19 and 26 after cell transfer (PD1d/IL-2v/IL-33 or IL-2v) and sequenced using 10x Genomics. Cluster generation was performed with the Illumina HiSeq 4000 PE cluster kit reagents. Sequencing was carried out on the Illumina HiSeq 4000 with HiSeq 4000 SBS Kit reagents. The sequencing data were demultiplexed with the bcl2fastq conversion software (v2.20, Illumina). Resulting FASTQ files were processed using the Cell Ranger count function with default settings for 5′ RNA gene expression analysis (Cell Ranger v4.0.0, 10x Genomics). The transgene cDNA sequence was included in the *Mus musculus* reference ('refdata-gex-mm10-2020-A' directory), according to 10× Genomics guidelines for building a custom reference genome with Cell Ranger mkref.

An aggregated UMI count matrix generated by Cell Ranger was filtered to high-quality CD8[+] TIL transcriptomes. First, we removed cycling cells and low-quality cells, which were identified using TILPRED[64] and filtered. Next, cells with 400 to 3,000 detected genes, 500 to 12,000 UMI counts, mitochondrial content below 5% and ribosomal protein content below 50% were kept, obtaining 42,866 high-quality CD8[+] TIL transcriptomes. For dimensionality reduction, we first identified HVGs using the vst method in Seurat 4.0.1 with default parameters[60]. Next, mitochondrial, ribosomal protein-coding genes and cell cycle genes (those bearing GO term GO:0007049) were removed from the set of HVGs, and the remaining HVGs (1,626) were scaled to have mean of 0 and variance of 1. Standardized HVGs were used for (1) dimensionality reduction using PCA, and (2) UMAP (as implemented in Seurat v4.0.1) on the first 50 principal components (with other parameters by default). Clustering was performed using the shared nearest-neighbor method of Seurat with parameters using FindNeighbors (k.param = 40) with default parameters and FindClusters with resolution = 0.35. For supervised classification of endogenous CD8[+] TIL states and the C5 state, we used ProjecTILs[21], with default parameters. DEGs between clusters were identified using FindAllMarkers and Wilcoxon's test with parameters min.pct = 0.25, min.diff.pct = 0.1 and logfc.threshold = 0.25. DEGs between $T_{SE}$ and $T_{EX}$ from the TIL reference map were detected upon projection of $T_{SE}$ cells onto the TIL reference map with the function make.

projection and then the function find.discriminant.genes (ProjecTILs, default parameters). A similar approach was followed for the comparisons $T_{SE}/T_{BE}$, $T_{SP}/P_{EX}$, $T_{SE}/SLEC$ and $T_{SE}/Cx3cr1^+$ exhausted intermediate cells from ref. 65. For the last two DEGs, we projected the $T_{SE}$ cells onto the viral reference map, instead. To infer the pseudotime trajectory, we first selected the cells annotated as OT1 and endogenous CD8[+] TILs (baseline). Then, the paga-tree method implemented in dynverse[54] was applied to this subset, downsampling to a maximum of 300 cells per cluster, and rooting the trajectory on *Sell* expression (naïve-like cluster). The inferred trajectory, together with cluster annotation and pseudotime, was visualized using diffusion maps.

### Human–mouse and mouse–mouse projections on the endogenous/OT1 ACT reference space

Mouse and human publicly available datasets ('Data availability') were processed with ProjecTILs[21] to keep only the high-quality CD8[+] T cells, which were next classified by ProjecTILs using default parameters and by setting the parameter human = TRUE for human datasets.

### Gene-set and pathway enrichment analysis

The PD-1 blockade/ACT gene signature was generated as described in ref. 64. RNA-seq data from adoptively transferred (in vitro activated) OT1 cells infiltrating B16-OVA tumors (anti-PD-1 versus control) were obtained from ref. 33. The top 345 significantly upregulated genes ($P < 0.05$, anti-PD-1 versus control) were used in the analysis. The C5 and $T_{EX}$ signatures were generated by differential expression analysis of C5 $T_{SE}$ cells versus canonical $T_{EX}$ cells from the reference TIL map using the function find.discriminant.genes (ProjecTILs) with parameters min.pct = 0.1, logfc.threshold = 0.1, query.assay = RNA and all. genes = T. Next, mitochondrial, ribosomal protein-coding genes, cell cycle genes and TCR-coding genes were removed from the DEGs list. The resultant 310 most upregulated genes in C5 ($P < 0.05$) and the 328 most upregulated genes in $T_{EX}$ were used for the analysis. To generate the CA-RIT-NFAT1 versus mock ranked gene list, RNA-seq data from CD8[+] T cells representing each condition were obtained from the Gene Expression Omnibus (GEO) under accession GSE64409 (ref. 34). The raw data were downloaded and processed with GREIN (http://www.ilincs.org/apps/grein/?gse). Next, we performed differential expression analysis between (mutant NFAT1 and mock CD8[+] T cells) with the package DESeq2 implemented in GENAVI (https://junkdnalab.shinyapps.io/GENAVi/), with default parameters. The gene signature of PD-L1-mediated inhibition in CD8[+] T cells was generated as described in ref. 37. The gene signature of CD8[+] TILs with high mitochondrial potential and RNA-seq data from high mitochondrial potential CD8[+] TILs was obtained from the GEO under accession GSE156506 (ref. 36). GSEA was done with the GSEA functions from clusterProfiler[66] package (v3.12)[63] using default parameters. Reactome pathway and GO enrichment analysis were performed with the functions GSEA, enrichGO and enrichReactome on the biological process and Reactome databases using clusterProfiler with default parameters.

### Regulon analysis

Regulons were inferred using the SCENIC pipeline (https://scenic.aertslab.org)[42]. To capture transcriptional regulatory events that only take place in some subpopulations, we split the dataset and carried out a separate regulon inference for each of the subpopulations, namely C2, C4 ($P_{EX}$), C4 ($T_{EX}$), C5, C6 and C7. The aggregation of targets into raw putative regulons was done using the runSCENIC_1_coexNetwork2modules function with nTopTfs and nTopTargets parameters set to 50 and 5, respectively. Coexpression modules were refined by removing indirect targets by motif discovery analysis using cisTarget algorithm and a *cis*-regulatory motif database[67,68]. At this step, we used mm9-500bp-upstream-7species.mc9nr.feather and mm9-tss-centered-10kb-7species.mc9nr.feather cisTarget databases, and the motifs-v9-nr.mgi-m0.001-o0.0.tbl motif database. The

resulting regulons for the same TF inferred in different subpopulations were aggregated in a merged regulon signature that was evaluated for each individual cell in the dataset. To this end, the AUCell R package was used to quantify the regulon activity in each individual cell (https://github.com/aertslab/AUCell). Regulons with less than five constituents were discarded, as the calculation of the area under the curve for smaller regulons is not reliable.

### Regulon activity clustering

TFs were clustered by regulon activity using the hierarchical clustering function embedded within the pheatmap R package (https://rdrr.io/cran/pheatmap/). The unsupervised hierarchical clustering was performed using Euclidean distances and 'complete linkage' as the clustering method. The cutree_rows argument, which defines the number of clusters, was set to 12. As the input for the clustering, we aggregated the regulon activity matrix taking, for each TF, the mean regulon activity value for each of the six subpopulations, namely, C2, C4 ($P_{EX}$), C4 ($T_{EX}$), C5, C6 and C7.

### OPLS-DA

We used the ropls R package[69] to perform OPLS-DA. The absolute value of the relative contribution of each feature (regulon) to the predictive axis (weightStarMN value) was normalized between 0 and 1, preserving the sign of the original value. The resulting number became the discriminant score of the regulon. The input data for the OPLS-DA was scaled (mean centered and divided by the standard deviation). The comparison of C7 versus background corresponds with an analysis where one class includes the query population (C7) and the other class includes a balanced proportion of the remaining subpopulations. For this purpose, we sampled a subset with a number of cells equivalent to the size of the smallest population. Similarly, we also subset the dataset for the comparison C7/C5/C6 versus C2/C4 ($P_{EX}$)/C4 ($T_{EX}$) so that the number of cells for each contributing population to each of the classes were balanced.

### Network connectivity analysis

To test whether the number of regulatory interactions between a given set of TFs were significantly higher than expected by chance from a random selection of TFs, a connectivity analysis was performed using the SANTA algorithm[55] on the gene regulatory network derived from the regulon analysis. In this work, we used 10,000 permutations for each test and considered P values ≤ 0.05 as significant. The targets of these TFs were analyzed for enrichment GO terms and Reactome pathways using the online tool WebGestalt (http://www.webgestalt.org/).

### Target analysis

The target analysis evaluates the common elements of two given regulons A and B. The redundancy (R) was calculated as follows:

$$R(A,B) = \frac{|A \cap B|}{\min(|A|, |B|)}$$

For unsupervised reclustering of the datasets, see E-MTAB-11773 (ref. 35) and GSE206739 (ref. 40).

For dimensionality reduction, we identified HVGs using Seurat 4.1.1 vst method with default parameters[60]. Next, mitochondrial, ribosomal protein-coding genes and cell cycle genes (those bearing GO term GO:0007049) were removed from the set of HVGs, and remaining HVGs (665) were scaled to have mean = 0 and variance = 1. Standardized HVGs were used for (1) dimensionality reduction using PCA, and (2) UMAP (as implemented in Seurat v4.0.1) on the first 20 principal components (with other parameters by default). Clustering was performed using the shared nearest-neighbor method of Seurat with parameters using FindNeighbors (k.param = 20) with default parameters and FindClusters with resolution = 0.2.

### Statistical analysis

The normal distribution of data was evaluated using the Shapiro–Wilk normality test. A two-tailed Student's *t*-test was used to compare two groups (for a normal distribution and homoscedasticity), or a *t*-test with Welch's correction (for a normal distribution but no homoscedasticity); if data were not normally distributed, the non-parametric Mann–Whitney test was used. For comparing more than two groups, we followed a similar strategy, and a Kruskal–Wallis test was used if there was no normal distribution. A one-way ANOVA was used for a normal distribution and homoscedasticity, or a Brown–Forsythe and Welch ANOVA test was used in case of normal distribution but no homoscedasticity. Correction for multiple comparisons was done using Dunn's test (for Kruskal–Wallis test), Dunnet's test (for one-way ANOVA test) and Tukey's test (for Brown–Forsythe's test). Survival analysis was performed using a log-rank Mantel–Cox model. The Pearson's correlation test was used to calculate the correlation between the number of TCF1$^+$ OT1 intratumoral CD8$^+$ T cells and the total number of tumor-infiltrated OT1 cells. All these statistical analyses were done with GraphPad Prism v9.0. Values of $P < 0.05$ were considered significant and ranked as *$P < 0.05$, **$P < 0.01$, ***$P < 0.001$ and ****$P < 0.0001$.

Statistical analysis of tumor control in Fig. 1b was performed using the percentage change in tumor volume relative to day 17 after tumor inoculation (5 days post 1st cell infusion). The best response (smallest tumor volume) observed for each animal after at least 12 d after the first ACT was taken for the calculation. The ORR and clinical benefit rate by treatment group were calculated over the total number of mice per group as:

- Objective response includes complete response (100% reduction) and partial response (≤ −30% tumor change)
- Clinical benefit includes complete response, partial response and stable disease (−30 < tumor change ≤ +20%)

Predicted probabilities of the variables 'objective response' and 'clinical benefit' were calculated using exact logistic regression. Values of tumor change as a continuous variable were further analyzed using linear regression. P values lower than 0.05 were considered statistically significant.

### Reporting summary

Further information on research design is available in the Nature Portfolio Reporting Summary linked to this article.

## Data availability

Publicly available files corresponding to the GEO accession codes GSE126974, GSE123139, GSE99254 and GSE125881 were obtained from the TISH repository (http://tisch.comp-genomics.org). The dataset EGAS00001004809 was obtained from the European Genome-phenome Archive. Finally, the dataset E-MTAB-11773 was obtained from ArrayExpress and the dataset GSE206739 directly from the GEO.

The scRNA-seq data generated in this study are deposited in the GEO under accession GSE200535. Source data are provided with this paper. All other data are present in the article and Supplementary Information. The processed Seurat R object used for analyses shown in Figs. 3–8 and the TIL_ACT reference map are available in the Figshare repository.

## Code availability

The R script (FirstBatch_comb_TIL_B16_AllSamples.Rmd) to fully reproduce Fig. 2 is available at https://github.com/carmonalab/GEEP_Jesus_Nov2021. The remaining custom R scripts are available upon request.

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

## Acknowledgements

We thank D. Speiser and W. Held for useful discussions and K. Leon (CIM, Havana, Cuba) for providing recombinant IL-2v for in vitro T-cell expansion. We also thank J. Marquis, C. Peter and K. Bojkowska from Lausanne Genomic Technologies Facility at UNIL for their support with scRNA-seq experiments. We also thank F. Sala de Oyanguren, R. Bedel and K. Blackney from the Flow Cytometry Facility at UNIL for the technical support. This work was generously supported by Ludwig Cancer Research, the Biltema and the ISREC Foundations, a generous grant by the prostate cancer foundation and an Advanced European Research Council grant to G.C. (1400206AdG-322875). It was also supported by a Swiss National Science Foundation (SNF) grant to S.A.L. (310030_185226/1) and by a SNF Ambizione grant to S.J.C. (180010). C.J.-L was partially supported by a postdoctoral fellowship from Ramon Areces Foundation. P.R. was supported in part by SNSF grant no. 310030_182735 and Oncosuisse KFS-4404-02-2018.

## Author contributions

G.C., J.C.-O. and M.I. conceived the project and designed the experiments; molecular cloning was performed by J.C.-O. and B.S.; J.C.-O., T.M., E.S., Y.O.-M., W.C., C.J.-L., I.A.R., A.S. and C.R. performed experiments; E.L. optimized the retroviral transduction protocol. Analysis of scRNA-seq datasets was performed by J.C.-O., S.J.C. and M.A.; regulon analysis was performed by I.C.; histological analysis was designed and performed by S.A.L. and L.S.; J.C.-O., S.J.C. and G.C. interpreted data. The manuscript was written by G.C., J.C.-O., P.R and M.I.; S.J.C., M.A. and S.A.L. also contributed to editing the manuscript. The study was supervised by G.C.

## Funding

## Competing interests

G.C. has received grants and research support or is co-investigator in clinical trials by Bristol Myers Squibb, Celgene, Boehringer Ingelheim, Roche, Tigen Pharma, Iovance Biotherapeutics and Kite. Lausanne University Hospital (CHUV) has received honoraria for advisory services. G.C. has also received honoraria for advisory services provided to AstraZeneca, Bristol Myers Squibb, F. Hoffmann-La Roche, MSD Merck and Geneos Therapeutics. G.C. has patents in the domain of antibodies and vaccines targeting the tumor vasculature as well as technologies related to T-cell expansion and engineering for T-cell therapy. G.C. receives royalties from the University of Pennsylvania. J.C.-O. has patents in the domain of immune cytokines (TGF-β mutants) as well as in technologies related to T-cell expansion and engineering for T-cell therapy. The other authors declare no competing interests.

## Additional information

**Extended data** is available for this paper at https://doi.org/10.1038/s41590-023-01477-2.

**Correspondence and requests for materials** should be addressed to Jesus Corria-Osorio or George Coukos.

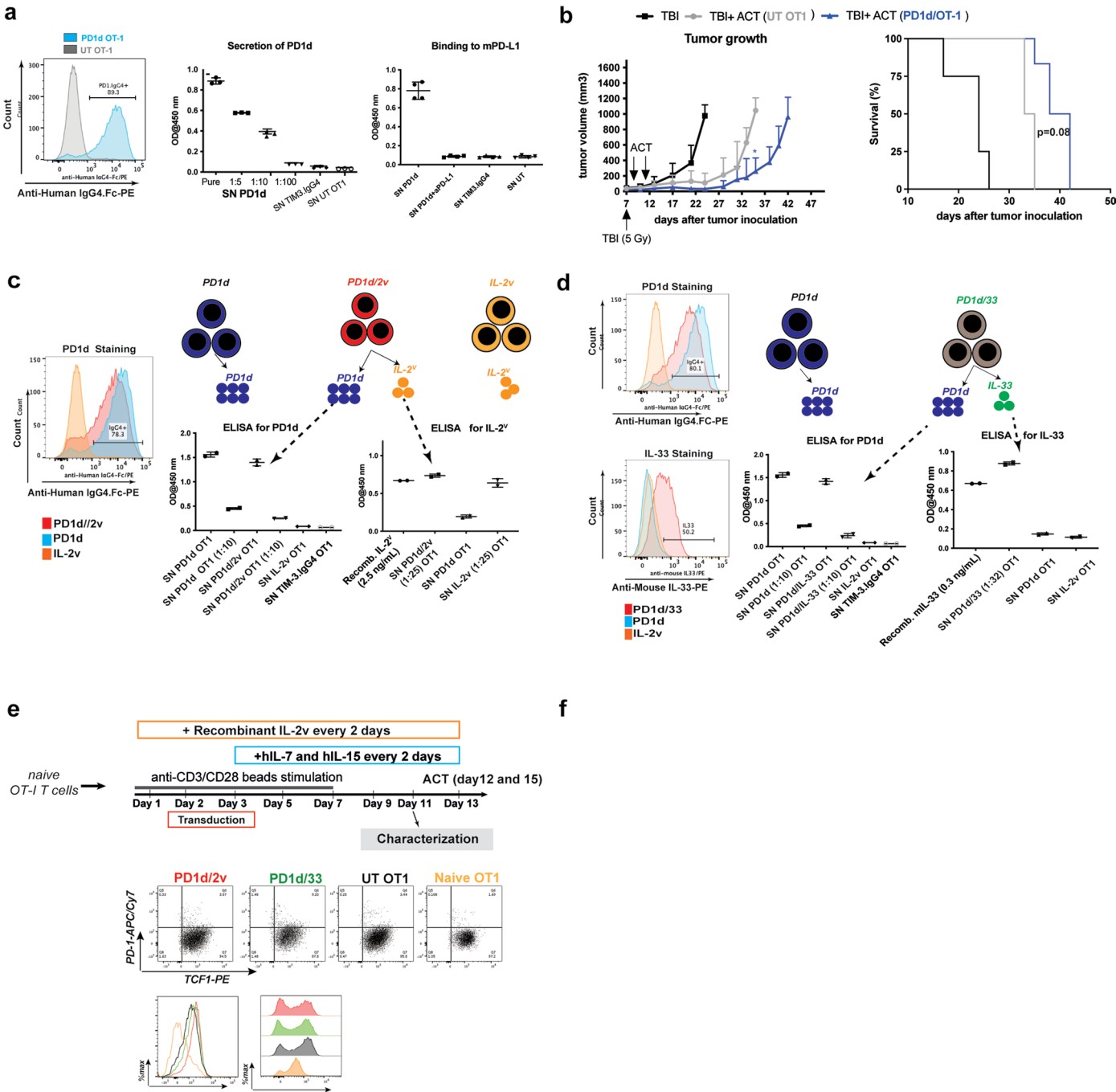

**Extended Data Fig. 1 | OT1 T cells can be effectively transduced to secrete PD1d, IL-2v and IL-33. a left:** Intracellular expression of PD1d by FACS. **Right Top:** Secretion into the SN determined by ELISA ($n = 3$ independent experiments, 2 technical replicate/experiment). **Right Bottom:** Binding to plate-coated mPD-L1. SN from OT-1 T cells gene-engineered to secrete TIM-3.IgG4 decoy was used as negative control. ($n = 4$ independent experiments, 2 technical replicate/experiment). **b)** Anti-tumor activity of OT-1 T cells gene-engineered for secreting the PD-1_IgG4 decoy in small B16-OVA tumors (~30 mm³). Effect on tumor growth (left) and on overall survival (right) of ACT using OT1 cells gene-engineered to secrete the PD-1.IgG4 decoy. A representative experiment out of two independent experiments using 6 mice per group is shown. Survival curves were compared using a log-rank Mantel−Cox test. Tumor growth comparison at day 35 post tumor inoculation was done using a Mann−Whitney Test. ***$P < 0.001$. **c, d** OT-1 CD8 T cells can be gene-engineered to secrete PD1d in combination

with either IL-2ᵛ or IL-33. **c Left:** intracellular expression of PD1d in PD1d/2ᵛ gene-engineered OT1 cells (there are no commercially available antibodies for detecting the IL-2ᵛ using Flow Cytometry). **Right:** secretion of PD1d and IL-2ᵛ into the supernatant by ELISA, ($n = 2$ independent experiments, 2 technical replicate/experiment). **d Left:** Intracellular expression of PD1d and IL-33 in PD1d/33 gene-engineered OT1 cells. **Right:** secretion of PD1d and IL-33 into the supernatant by ELISA. OT1 cells gene-engineered to secrete TIM-3.IgG4 decoy were used as negative control to claim specificity in the PD-1_IgG4 ELISA, ($n = 2$ independent experiments, 2 technical replicate/experiment). Data are presented as mean values ± s.d. **d** Phenotypic characterization of gene-engineered OT1 cells before ACT. Top: schematic showing the transduction and expansion protocol of gene-engineered OT1 cells. Bottom: expression of CD44, CD62L, TCF1 and PD1 in expanded OT1 cells on day 10 post viral transduction. *$P < 0.05$, **$P < 0.01$, ***$P < 0.001$, ****$P < 0.0001$.

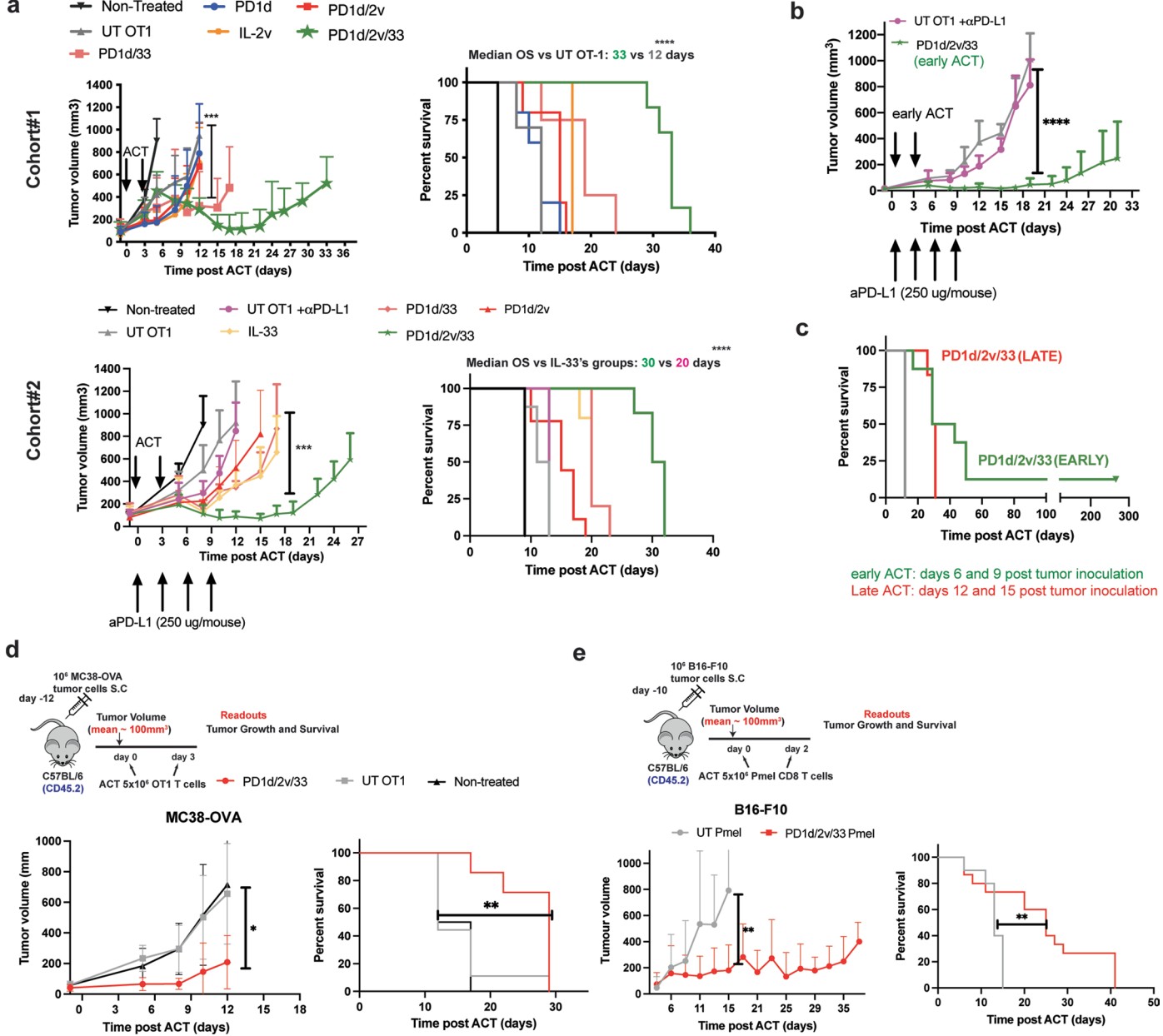

**Extended Data Fig. 2 | Orthogonal T-cell engineering enables ACT efficacy without lymphodepletion. a Top:** a representative first independent experiment comprising non-treated ($n = 10$) or treated mice with UT OT1 ($n = 10$), PD1d ($n = 5$ mice), IL-2$^v$ ($n = 4$ mice/group), PD1d/2$^v$ ($n = 5$ mice/group), PD1d/33 ($n = 4$ mice/group) and PD1d/2$^v$/33 ($n = 6$ mice/group). Bottom: a representative second independent experiment comprising non-treated ($n = 6$) or treated mice with UT OT1 (n = 8), UT OT1 plus αPD-L1 ($n = 5$ mice), PD1d/2$^v$ ($n = 9$ mice/group), PD1d/33 and IL-33 ($n = 5$ mice/group), PD1d/2$^v$/33 ($n = 8$ mice/group). Data are presented as mean values ± s.d. Survival Data were determined using a log-rank Mantel−Cox test. Tumor growth comparison was done using a two-sided Mann−Whitney Test ***$P < 0.001$, ****$P < 0.0001$. **b** A representative experiment showing tumor growth curves of B16.OVA-bearing mice treated with either UT OT1 ($n = 5$ mice), UT OT1 plus αPD-L1 ($n = 5$) or PD1d/2v/33 ($n = 10$) on days 6

and 9 after tumor cell inoculation. **c** Survival curves of B16.OVA-bearing mice treated with either early or late orthogonal ACT from a–b. Data are presented as mean values ± s.d. Survival Data were determined using a log-rank Mantel−Cox test. Tumor growth comparison was done using a two-sided Mann−Whitney Test ****$P < 0.0001$. **d** A representative experiment showing tumor growth and survival curves of non-treated MC38.OVA-bearing mice ($n = 10$) or treated with either UT OT1 ($n = 10$) or PD1d/2$^v$/33$^+$OT1 cells ($n = 10$) on days 12 and 15 after tumor cell inoculation. **e** A representative experiment showing tumor growth and survival curves of B16.F10-bearing mice treated with either UT Pmel ($n = 10$) or PD1d/2$^v$/33$^+$Pmel cells ($n = 15$) on days 12 and 15 after tumor cell inoculation Survival Data were determined using a log-rank Mantel−Cox test. Tumor growth comparison was done using a two-sided Mann−Whitney Test *$P < 0.05$, **$P < 0.01$, ***$P < 0.001$, ****$P < 0.0001$.

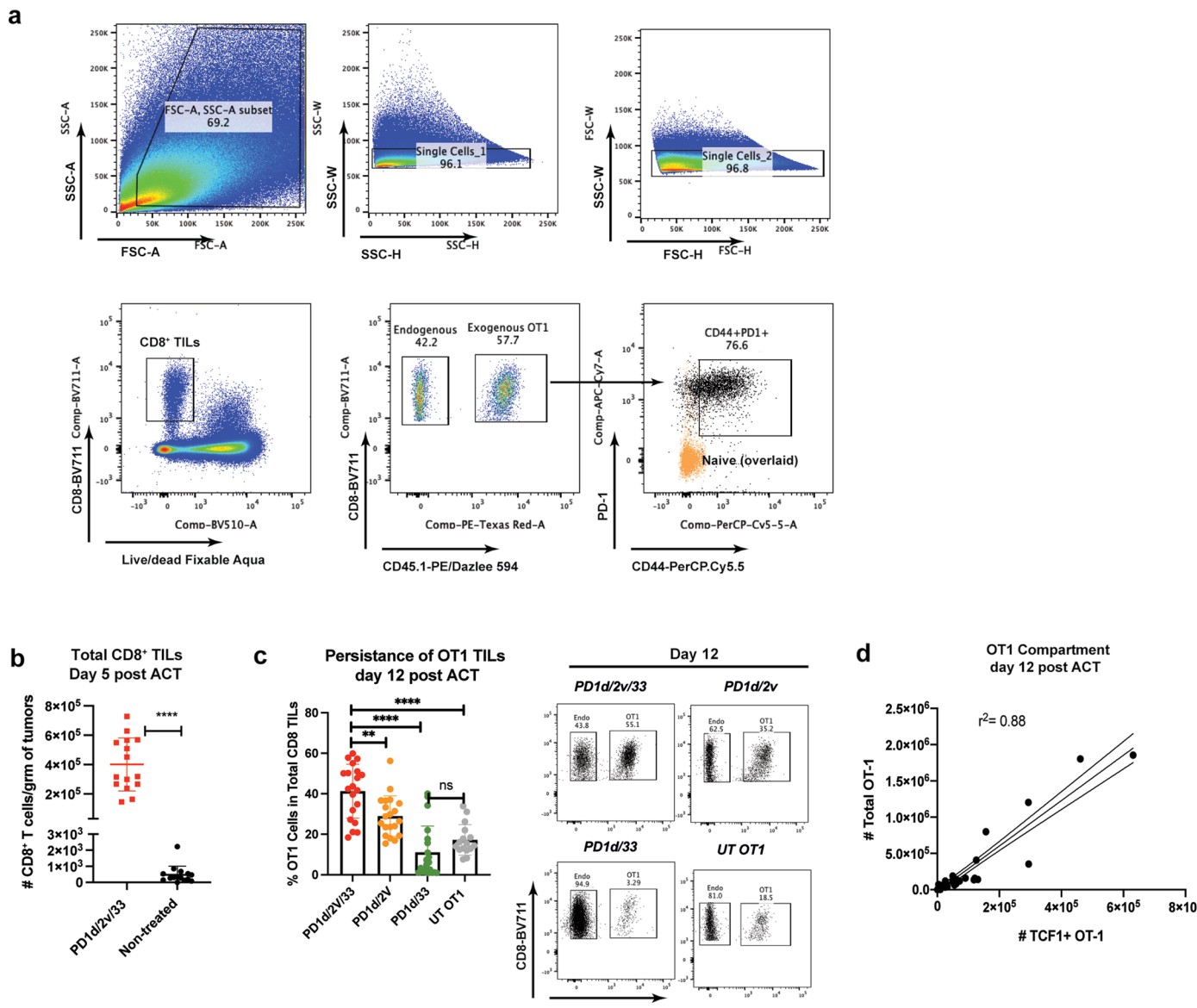

**Extended Data Fig. 3 | Orthogonal ACT expands CD8⁺ TILs *in situ* in a cell autonomous manner. a** General gating strategy to analyze the adoptively transferred OT1 TILs. Mice with B16.OVA tumors were treated with either engineered or untransduced OT1 cells on days 12 and 15 after tumor cell inoculation; then tumors were harvested on days 5 and 12 post ACT and cell quantification was performed by flow cytometry using precision counts beads. **b** Total number of CD8⁺ TILs at day 5 in mice treated with orthogonal ACT versus non-treated tumor bearing mice (17 days post tumor inoculation) evaluated in parallel. Data are from three independent experiments (*n* = 5 mice/group). Data are presented as mean values ± s.d. A two-tailed Student's *t*-test with

Welch's correction was used for comparing both conditions****$P < 0.0001$. **c** Intratumoral persistence of OT1 TILs at day 12 post ACT in each studied group. Right: representative dot plots for each treatment condition. Data are from 4 independent experiments (*n* = 4–5 animals/group/experiment). Data are presented as mean values ± s.d. An ordinary one-way ANOVA with a Tukey's test for multiple comparison was performed for comparing conditions differences, *$P < 0.05$, **$P < 0.01$ ***$P < 0.001$, ****$P < 0.0001$. **d** Pearson's correlation between the numbers of intratumoral TCF1⁺ OT1 T cells and the total number of intratumoral OT1 T cells on day 12 post orthogonal ACT. Data are from two independent experiments (*n* = 6 animals/group/experiment).

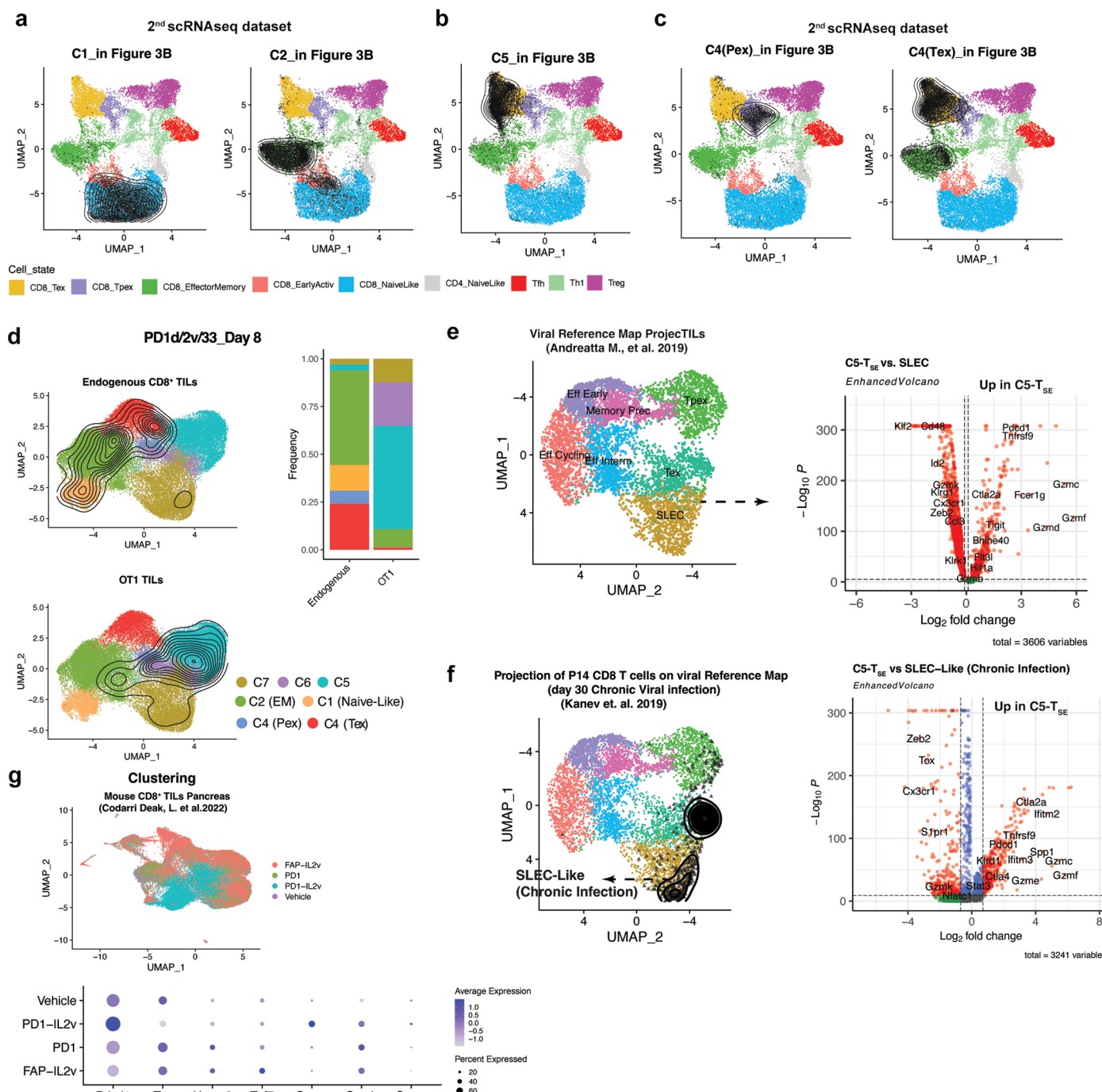

**Extended Data Fig. 4 | C5 is a novel T-synthetic cell effector state (T_SE).**
**a-c** Annotation of clusters C1-Naïve, C2-EffectorMemory, C5-$T_{SE}$ and C4 from the 2nd scRNAseq dataset based on their projection onto the reference mouse TIL map using ProjecTILs, where contour plots depict high cell density areas for each treatment. **d** UMAP plot showing the cluster distribution of either OT1 or endogenous CD8 TILs recovered at day 8 post orthogonal ACT. Contour plots depict the clusters covered by each cell compartment. Bar plots are showing the cluster composition of each compartment. **e** Comparison of C5-$T_{SE}$ TILs and SLEC effector cells induced upon acute viral infection. Volcano plot showing the DEGs between C5-$T_{SE}$ cells and SLEC like cells from the reference viral map[21]. Briefly, C5-TILs were projected onto the viral map using ProjecTILs. Next, the C5-TILs projected on the SLEC space were directly compared with SLEC-cells from the reference map for DEGs. Calculation performed with the function find. discriminant.genes (ProjecTILs) which performs differential expression based

on the non-parameteric Wilcoxon rank sum test. **f** Comparison of C5-TSE TILs and Cx3cr1+ exhausted intermediate effector cells induced upon chronic viral infection. Volcano plot showing DEGs between C5-$T_{SE}$ cells and $Cx_{3}Cr1$ + -SLEC like cells obtained from Kanev. et al[64]. Briefly, scRNAseq profile of P14 CD8 T cells harvested at day 30 post LCMV were projected in the viral reference map using ProjecTILs. Next, the cells classified as SLEC-like were directly compared to C5-TILs for DEGs. Calculation was performed with the function find.discriminant. genes (ProjecTILs) which performs differential expression analysis based on the non-parameteric Wilcoxon rank sum test. **g** Top part: UMAP plot showing the distribution of mouse TILs harvested from mice treated with either antiPD-1, FAP-IL2v, PD1_IL2v bispecific immunocytokine or vehicle, generated from (E-MTAB-11773), Codarri Deak L. el at. 2022[35]. Bottom part: dot plot showing important T-cell gene markers.

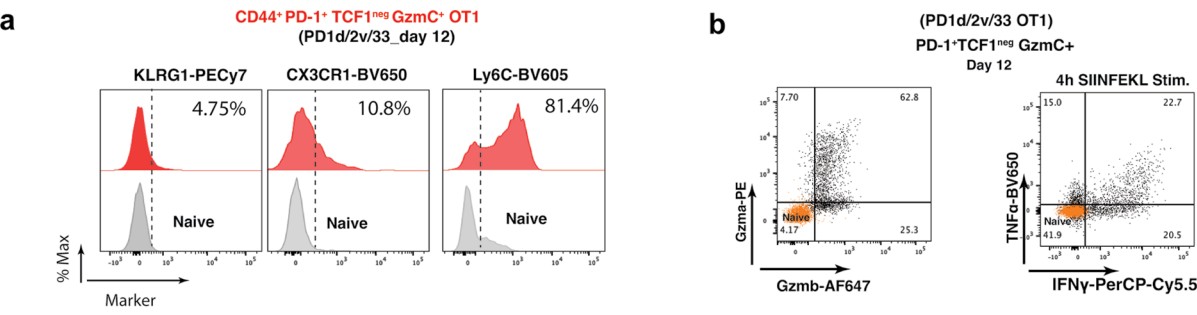

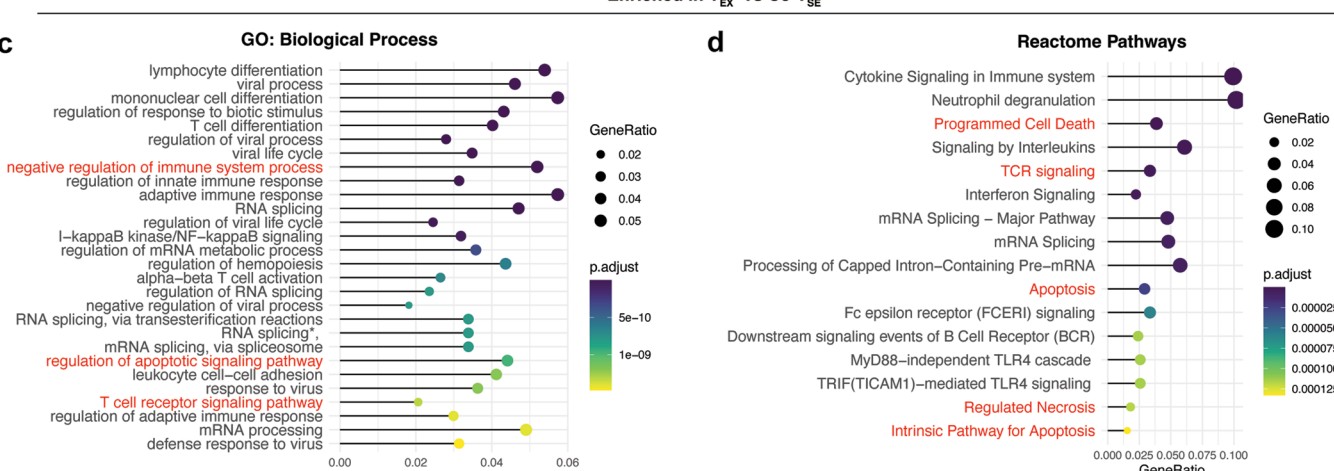

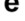

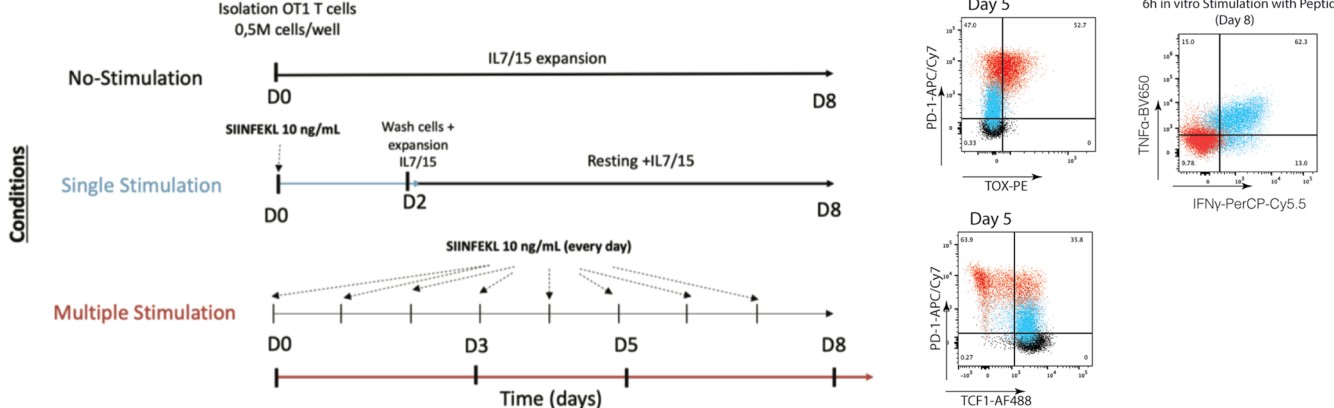

**Extended Data Fig. 5 | C5-T$_{SE}$ OT1 TILs are polyfunctional effector cells.**
**a** Analysis of KLRG1, CX$_3$CR1 and LY6C expression in CD44$^+$ PD-1$^+$ TCF1$^{neg}$ GZMC$^+$ OT1 TILs at day 12 post orthogonal ACT. **b** Representative dot plots showing the expression of effector molecules in CD44$^+$PD-1$^+$TCF1$^{neg}$GZMC$^+$ OT1 TILs harvested at day 12 post orthogonal ACT and 4 h in vitro stimulated with 10 ng/mL of cognate SIINFEKL peptide in the presence of Brefeldin A. GO: Biological Process **(c)** and Reactome pathway **(d)** over-representation tests of genes upregulated in T$_{EX}$ from the reference TIL map[21], relative to C5-T$_{SE}$ TILs. Reactome-Pathway and GO enrichment were performed with the functions, enrichGO and enrichReactome, respectively, from the same package, which use a one-sided version of the Fisher's exact overrepresentation test to find enriched categories. **e** Top: Schematics showing the experimental protocol to *in vitro* generate bona-fide exhausted OT1 T cells. Bottom: Analysis of the expression of PD-1, TOX, TCF1, IFNγ and TNFα in multiple stimulated OT1 cells (red). OT1 cells that received one single stimulation (blue) or none (black), were used as controls.

a

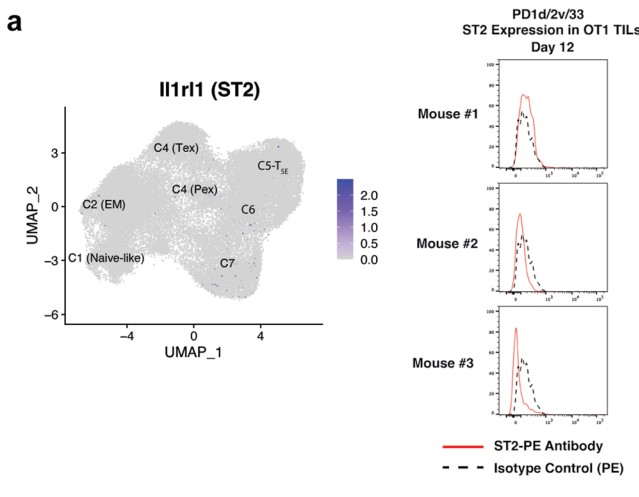

b

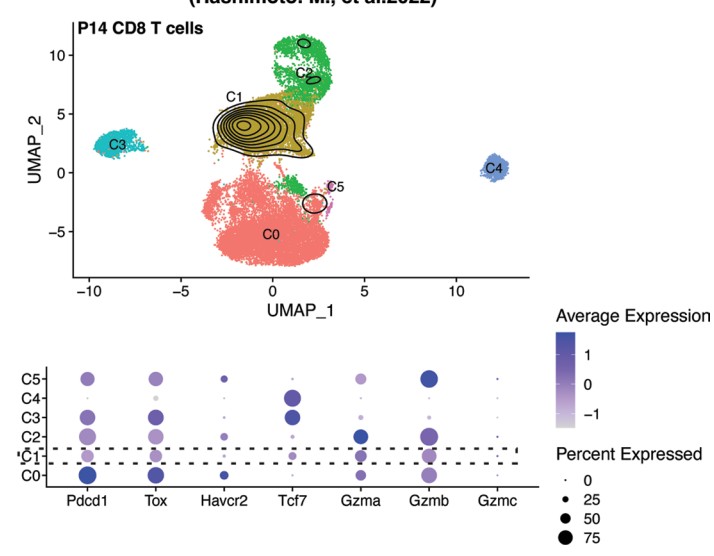

**Extended Data Fig. 6 | Synthetic CD8 T cells do not express ST2 and are distinct of those expanded by anti-PD-1 in combination with IL-2. a** Analysis of the Expression of ST2 at both RNA and protein levels. **b** Top part: UMAP plot showing the distribution of P14 CD8 T cells harvested from LCMV chronically infected mice treated with either antiPD-1, IL2 or their combination generated from (GSE206739), Hashimoto. M., et al.[40]. Contour plots depict the clusters covered by the combination. Bottom part: dot plot showing important T-cell gene markers.

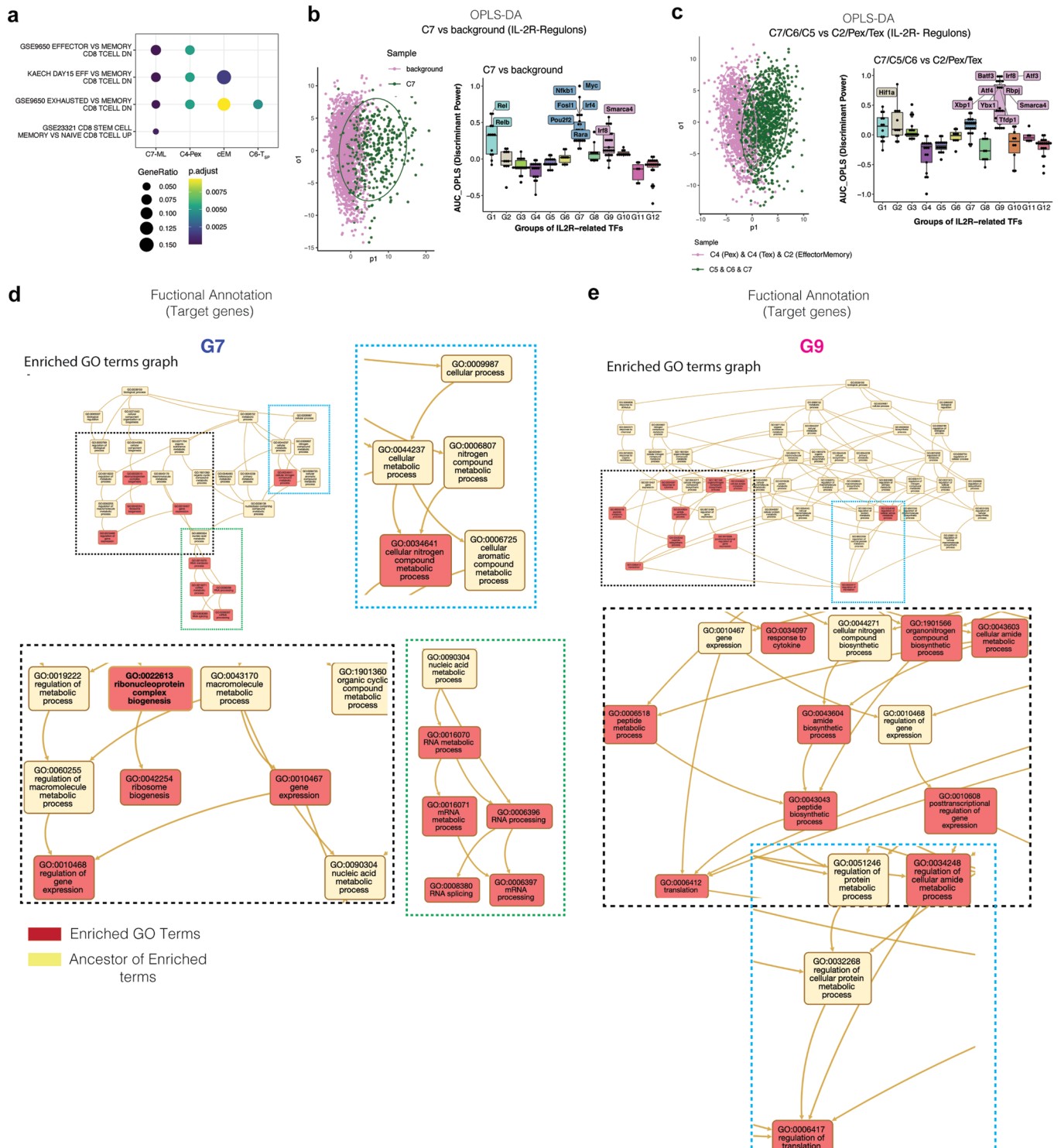

**Extended Data Fig. 7 | Gene sets over-representation test and OPLS-DA to infer the most discriminant IL-2R-related TF/regulons for synthetic CD8 TILs.** **a** Msigdb gene sets over-representation test of genes specifically upregulated in each memory-like subset. Msigdb gene sets enrichment was performed with the function 'enricher' (ClusterProfiler), which use a one-sided version of the Fisher's exact overrepresentation test to find enriched categories. **b**, **c** Orthogonal Partial Least Squares discriminant analysis (OPLS-DA) to identify the most discriminant/important TF/regulons for C7 versus background **(b)** or canonical versus synthetic T-cell states **(c)**. The OPLS-DA is a multivariate analysis that takes advantage of the pre-known classes (cell populations,) to remove the intra- class dispersion and maximize the inter- class dispersion, in order to derive a predictive axis (p1). The resulting predictive axis constitute a projection of the regulon activity matrix that optimizes the split between the elements (cell) of the two given pre-known classes (cell populations). In box plots, the median is indicated by the center line; box limits represent upper and lower quartiles; and whiskers extend to 1.5 times the interquartile range.**d-e** Network Topology-based Analysis (NTA) of the target genes of the top_25% most discriminant TFs of G7 and G9 using the online tool Webgestalt (http://www.webgestalt.org/).

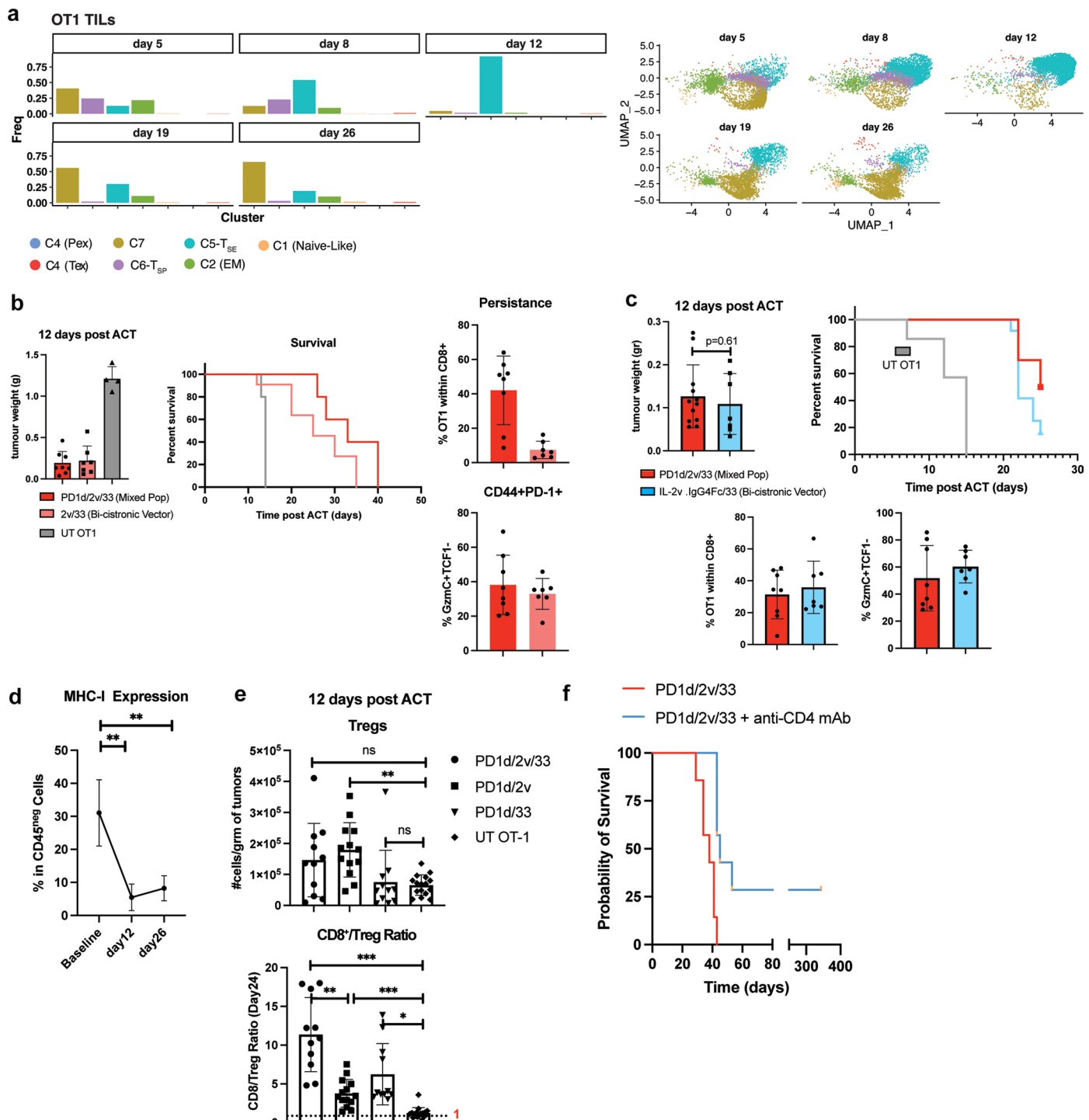

**Extended Data Fig. 8 | See next page for caption.**

**Extended Data Fig. 8 | Tumor progression is not associated with canonical T-cell exhaustion. a** Dynamics of OT1 T-cell states over time. Left: bar plots showing cluster composition (Frequency) in each time point. Right OT1 single cells in each time point. **b** Left: tumor weight comparison and survival curves of B16.OVA-bearing mice treated with either UT OT1 or PD1d/$2^v$/33 + OT1 cells (1:1 mixed PD1d/2v and PD1d/33 cells) or OT1 T cells transduced with a bicistronic vector comprising IL-2v and IL-33. Right part, top: Analysis of OT1 intratumoral persistence at day 12 post ACT and acquisition of the Gzmc$^+$ TCF1$^{neg}$ C5-like cell state (bottom). Data are presented as mean values ± s.d. and are from 2 independent experiments except for UT OT1 ($n$ = 4 mice). PD1d/2v/33 and 2v/33 ($n$ = 4 mice/group/experiment). Data are presented as mean values ± s.d. **c** Similar to **b**, but this time OT1 T cells were transduced to express and secrete a dimeric IL-2v (IL-2v/IgG4) and IL-33. Data are presented as mean values ± s.d. and are from 2 independent experiments $n$ = 6 mice/experiment for PD1d/2v/33 and $n$ = 3–4 mice/experiment for IL2v.IgG4_IL33. Data are presented as mean values ± s.d. A two-tailed Student's $t$-test with Welch's correction was used for comparing both conditions. **d** Analysis of MHC-I expression overtime on CD45$^{neg}$ cells. Data are presented as mean values ± s.d. and are from two independent experiments for samples collected at baseline and day 12 ($n$ = 4–5 mice/time point) and for one independent experiment for samples collected at day 26 ($n$ = 4 mice). A two-tailed Student's $t$-test with Welch's correction was used for pairwise comparisons, $^{**}P$ < 0.01. **e** Top: tumor weight normalized numbers of Treg CD4$^+$ T cells on day 12 post ACT. Bottom: CD8$^+$/Treg ratio at day 12 post ACT. Mice with B16.OVA tumors were treated as indicated then tumors were harvested on day 12 post ACT, and Treg quantification was performed by flow cytometry. Data are presented as mean values ± s.d. and are from two independent experiments ($n$ = 5–7 animals/group). An ordinary one-way ANOVA with a Tukey's test for multiple comparison was performed for comparing conditions. NS, $P$ > 0.05, $^*P$ < 0.05, $^{**}P$ < 0.01, $^{***}P$ < 0.001, $^{****}P$ < 0.0001. **g** A representative survival curve of PD1d/$2^v$/33 + OT1-treated mice in the absence or presence of a depleting antibody specific for CD4 ($n$ = 7 mice/condition) administered i.p. 250 µg/mouse beginning 1 d before initiation of PD1d/$2^v$/33-ACT and maintained until day 43 post tumor inoculation. Survival curves were compared using a log-rank Mantel−Cox test.

**Extended Data Table. 1 | Observed response and predicted probability of objective response and clinical benefit for each treatment group**

| Treatment Group | No of mice | Objective Response (CR/PR) | | Clinical Benefit (CR/PR/SD) | |
|---|---|---|---|---|---|
| | | Observed n (%) | predicted probability (%) | Observed n (%) | predicted probability (%) |
| UT | 14 | 0 | 3.33% | 0 | 3.33% |
| UT + aPDL1 | 7 | 0 | 6.25% | 0 | 6.25% |
| PD1d | 6 | 0 | 7.14% | 0 | 7.14% |
| IL-2V | 4 | 0 | 10% | 0 | 10% |
| IL-33 | 5 | 0 | 8.33% | 1 (20.0) | 25% |
| PD1d + IL-2V | 12 | 0 | 3.85% | 0 | 3.85% |
| PD1d + IL-33 | 12 | 1 (8.33%) | 11.5% | 4 (33.3) | 34.6% |
| **PD1d + IL-2V + IL-33** | **14** | **12 (85.7%)** | **83.3%** | **12 (85.7)** | **83.3%** |

Notes:

Objective Response includes Complete Response (CR; 100% reduction in tumor volume) and Partial Response (PR; ≤-30% tumor change). Clinical Benefit includes CR, PR and Stable Disease (-30%<tumor change≤+20%)

Probability of occurrence was calculated using exact logistic regression.

**Extended Data Table. 2 | Predicted tumor size change from baseline for each group using linear regression**

| Treatment | Predicted (average) (%) change in tumor size | 95% CI of the expected (%) change in tumor size | p-value |
|---|---|---|---|
| UT | *287.1%* | (190% , 384%) | <0.0001 |
| UT + aPDL1 | 274.1% | (136% , 412%) | 0.0002 |
| PD1d | 343.7% | (195% , 493%) | <0.0001 |
| IL-2V | 608.0% | (426% , 790%) | <0.0001 |
| IL-33 | 145.7% | (-17% , 309%) | 0.0377 |
| PD1d + IL-2V | 290.8% | (186% , 396%) | <0.0001 |
| PD1d + IL-33 | 83.17% | (-22% , 188%) | 0.0570 |
| **PD1d + IL-2V + IL-33** | **-56%** | **(-100%\* , 41%)** | - |

\*: -100% is the minimum plausible value for tumor size change.

Notes: p-values compare each treatment effect versus the triple combination (PD1d + IL-2V + IL-33)

Adjusted $R^2$ of the model: 43.8%.

# Reporting Summary

## Statistics

For all statistical analyses, confirm that the following items are present in the figure legend, table legend, main text, or Methods section.

| n/a | Confirmed | |
|---|---|---|
| ☐ | ☒ | The exact sample size (*n*) for each experimental group/condition, given as a discrete number and unit of measurement |
| ☐ | ☒ | A statement on whether measurements were taken from distinct samples or whether the same sample was measured repeatedly |
| ☐ | ☒ | The statistical test(s) used AND whether they are one- or two-sided *Only common tests should be described solely by name; describe more complex techniques in the Methods section.* |
| ☒ | ☐ | A description of all covariates tested |
| ☐ | ☒ | A description of any assumptions or corrections, such as tests of normality and adjustment for multiple comparisons |
| ☐ | ☒ | A full description of the statistical parameters including central tendency (e.g. means) or other basic estimates (e.g. regression coefficient) AND variation (e.g. standard deviation) or associated estimates of uncertainty (e.g. confidence intervals) |
| ☐ | ☒ | For null hypothesis testing, the test statistic (e.g. *F*, *t*, *r*) with confidence intervals, effect sizes, degrees of freedom and *P* value noted *Give P values as exact values whenever suitable.* |
| ☒ | ☐ | For Bayesian analysis, information on the choice of priors and Markov chain Monte Carlo settings |
| ☒ | ☐ | For hierarchical and complex designs, identification of the appropriate level for tests and full reporting of outcomes |
| ☐ | ☒ | Estimates of effect sizes (e.g. Cohen's *d*, Pearson's *r*), indicating how they were calculated |

*Our web collection on statistics for biologists contains articles on many of the points above.*

## Software and code

Policy information about availability of computer code

| | |
|---|---|
| Data collection | Flow cytometry data were collected on LSR II instruments with FACSDiva software v8.0.1 (BD) |
| Data analysis | Flow cytometry data was analyzed on FlowJo v10.7.1 (TreeStar) and statistical analysis performed on Prism v9.0 (GraphPad). scRNA-seq data was processed using the 10x Cell Ranger Pipeline. Genomic analyses were performed using R Studio v4.1 using the following packages : Seurat v4.1, scGate, ProjecTILs v2.0, clusterProfiler v3.12, scRepertoire , dynverse, GREIN(http://www.ilincs.org/apps/grein/?gse), GENAVI (https://junkdnalab.shinyapps.io/GENAVi/), SCENIC, ropls R package, AUCell R package, Webgestalt (http://www.webgestalt.org/) and pheatmap. |
| | Incucyte images were quantified with the IncuCyte ZOOM integrated analysis software. The R script (FirstBatch_comb_TIL_B16_AllSamples.Rmd) to fully reproduce Fig. 2 is available here (https://github.com/carmonalab/GEEP_Jesus_Nov2021). The rest of our custom R scripts are available upon request. |

For manuscripts utilizing custom algorithms or software that are central to the research but not yet described in published literature, software must be made available to editors and reviewers. We strongly encourage code deposition in a community repository (e.g. GitHub). See the Nature Portfolio guidelines for submitting code & software for further information.

## Data

The scRNAseq data generated in this study are deposited in the National Center for Biotechnology Information Gene Expression Omnibus under accession number GSE200535. All other data are present in the article and supplementary files as well as the processed Seurat R object used for the analyses shown in Figs. 3-8 and the TIL_ACT reference map are available in the Figshare repository, as part of the Source data provided with this paper.

Publicly available files corresponding to the GEO accession codes: GSE126974, GSE123139, GSE99254, GSE125881 were obtained from the TISH repository(http://tisch.comp-genomics.org). The dataset EGAS00001004809 was obtained from EGA (European Genome-phenome Archive). Finally, the dataset E-MTAB-11773 was obtained from ArrayExpress and the dataset GSE206739 directly from the GEO repository .

# Field-specific reporting

Please select the one below that is the best fit for your research. If you are not sure, read the appropriate sections before making your selection.

☒ Life sciences  ☐ Behavioural & social sciences  ☐ Ecological, evolutionary & environmental sciences

For a reference copy of the document with all sections, see nature.com/documents/nr-reporting-summary-flat.pdf

# Life sciences study design

All studies must disclose on these points even when the disclosure is negative.

| | |
|---|---|
| Sample size | No statistical methods were used to pre-determine sample size, but our sample sizes (4-10 mice/group) are similar to those reported in our previous publication (https://doi.org/10.1158/2159-8290.CD-21-0003). |
| Data exclusions | No data were excluded from analyses |
| Replication | All experiments presented in this study were performed using at least 2 biological or technical replicates. All presented results were confirmed in at least two independent experiments, unless otherwise noted. Specifically for scRNA-seq experiments were performed one time and combined 5 mice (biological replicates) per sample pool. |
| Randomization | On day 11 post tumor inoculation (average tumor volume 100-200 mm3) mice were randomly regrouped in order to have comparative average tumor volumes between experimental arms. |
| Blinding | Only for the anti tumor efficacy experiments mice were monitored three times/week by an independent investigator in a blinded manner |

# Reporting for specific materials, systems and methods

We require information from authors about some types of materials, experimental systems and methods used in many studies. Here, indicate whether each material, system or method listed is relevant to your study. If you are not sure if a list item applies to your research, read the appropriate section before selecting a response.

## Materials & experimental systems

| n/a | Involved in the study |
|---|---|
| ☐ | ☒ Antibodies |
| ☐ | ☒ Eukaryotic cell lines |
| ☒ | ☐ Palaeontology and archaeology |
| ☐ | ☒ Animals and other organisms |
| ☒ | ☐ Human research participants |
| ☒ | ☐ Clinical data |
| ☒ | ☐ Dual use research of concern |

## Methods

| n/a | Involved in the study |
|---|---|
| ☒ | ☐ ChIP-seq |
| ☐ | ☒ Flow cytometry |
| ☒ | ☐ MRI-based neuroimaging |

## Antibodies

| | |
|---|---|
| Antibodies used | FCR blocking reagent (clone 2.4G2) (1:50) BD Biosciences Cat#553141<br>PE/Cy5.5 anti-mouse CD3e (clone 2. 145-2C11) (1:50) Invitrogen Cat# 14-0031-82<br>Brilliant Violet 711 anti-mouse CD8a (clone 2. 53-6.7) (1:50) Biolegend Cat# 100748 |

PE/Dazlee 594 anti-mouse CD45.1 (clone A20) (1:100) Biolegend Cat# 110748
PE-mouse anti-human IgG4 (clone HP6025) (1:100) abcam Cat#99825
PE-anti-mouse IL-33 (clone 396118) (1:50) Invitrogen Cat#MA5-23640
Purified polyclonal anti-mouse PD-1 (final conc. 2ug/mL) R&D Cat#AF1021
HRP anti-human IgG4 pFc' (clone: HP6023) (1:2000) abcam Cat#ab99817
Purified polyclonal anti-human IL-2 Antibody (final concentration 3 ug/mL) R&D Cat#AF-202-SP
Biotin Polyclonal anti-human IL-2 Antibody (1:500) Invitrogen Cat# 13-7028-81
PE/Dazlee 594 anti-mouse CD4 (clone GK1.5) (1:100) Biolegend Cat# 100456
PE- anti-mouseFOXP3 (clone FJK-16s) (1:50) Invitrogen Cat# 12-5773-82
Brilliant Violet421 - anti-mouse/human CD44 (clone IM7) (1:100) Biolegend Cat# 103040
APC/Cy7-anti-mouse PD-1 (clone 29F.1A12) (1:100) Biolegend Cat# 135224
BV605- anti-mouse Ly-6C (clone HK1.4) (1:800) Biolegend Cat# 128036
APC- anti-mouse Granzyme C (clone SFC1D8) (1:100) Biolegend Cat# 150812
PE/Cy7- anti-mouse Granzyme C (clone SFC1D8) (1:100) Biolegend Cat# 150804
Rabbit anti TCF1 (TCF7) antibody (clone C63D9) (1:200) Cell Signaling Technology Cat# 2203S
PE-anti-rabbit IgG (H+L), F(ab')2 Fragment (1:250) Cell Signaling Technology  Cat# 8885S
Alexa Fluor 488-anti-rabbit IgG (H+L), F(ab')2 Fragment (1:250) Cell Signaling Technology Cat# 4408S
PE/Cy7- anti-mouse/human Granzyme B (clone CLB-GB11) (1:50) Novus Biological Cat# NBP1-50071PECY7
AF647- anti-mouse/human Granzyme B (GB11) (1:50) Biolegend Cat# 515406
PE/Cy7- anti-mouse 4-1BB (clone 17B5) (1:50) Invitrogen Cat# 25-1371-82
Brilliant Violet 605 anti-mouse CD25 (clone PC61) (1:200) Biolegend Cat# 104530
Brilliant Violet 605 anti-mouse CD69 (clone H1.2F3) (1:100) Biolegend Cat# 104530
Brilliant Violet 421 anti-mouse CD69 (clone H1.2F3) (1:100) Biolegend Cat# 104528
Brilliant Violet 605 anti-mouse/human  CD44 (clone IM7) (1:100) Biolegend Cat#  103047
PecCP-Cy5.5 anti-mouse/human CD44 (clone IM7) (1:100) Biolegend Cat#  103032
Brilliant Violet 421 anti-mouse TIM-3(clone RMT3-23) (1:50)  Biolegend Cat# 119723
PE/Cy7 anti-mouse/human KLRG1(clone 2F1/KLRG1) (1:100) Biolegend Cat# 138416
Brilliant Violet 650 anti-mouse CX3CR1 (clone S011F11) (1:300) Biolegend Cat# 149033
Brilliant Violet 605 anti-mouse Ki-67(clone 16A8) (1:100) Biolegend Cat# 652413
PerCP-Cyanine5.5- anti-mouse IFNg (clone XMG1.2) (1:50) Invitrogen Cat# 45-7311-82
PE/Cy7- anti-mouse TNFa (clone MP6-XT22) 1:100  Biolegend Cat# 506324
BV650- anti-mouse TNFa (clone MP6-XT22) 1:100  Bdbioscience Cat# 563943
PE/Cy7- anti-mouse CD62L (clone MEL-14)  (1:100 )Biolegend Cat# 104418
PE- anti-TOX antibodies human and mouse (clone REA473) (1:50) Miltenyi Cat# 130-107-785
APC- anti-TOX antibodies human and mouse (clone REA473) (1:50)  Miltenyi Cat# 130-107-784
APC- Armenian Hamster IgG Isotype Ctrl Antibody (clone HTK888) (1:100) Biolegend Cat# 400912
PE/Cy7- Armenian Hamster IgG Isotype Ctrl Antibody (clone HTK888) (1:100) Biolegend Cat# 400922
PE- GzmA  (1:100) Biolegend Cat# 149704
APC-Prf1 (1:100) Biolegend Cat# 154304
Purified anti-mouse CD8a Antibody  Biolegend Cat#100702
Purified anti-mouse CD105 Antibody  Biolegend Cat#120402
Purified anti-mouse CD45.1 Biotin (clone A20.1) In house -
Cy™3 AffiniPure Donkey Anti-Rabbit IgG (H+L) Jackson ImmunoResearch Cat#711-165-152
Alexa Fluor 488 Donkey anti-Rat IgG (H+L) Highly Cross-Adsorbed Secondary Antibody Invitrogen Cat#A21-208
InVivoMAb anti-mouse CD4 (Clone: GK1.5) BioXcell Cat#BE0003-1
InVivoMAb anti-mouse PD-L1 (Clone: 10F.9G2) BioXcell Cat#BE0101
InVivoMAb anti-mouse TIM-3 (Clone: RMT3-23) BioXcell Cat#BE0115
Rat anti mouse anti-ST2 antibody was used (Clone : DJ8mdbioproducts CatNumber: 101001PE, dilution 1:50)
PE Rat IgG1, λ Isotype Ctrl Antibody Biolegend CAT# 401906
anti mouse CD3 Monoclonal Antibody (17A2), (final conc. 1 ug/mL) eBioscience™ CAT# 14-0032-82

Validation | Antibodies were validated by the manufacturer.

# Eukaryotic cell lines

Policy information about cell lines

| | |
|---|---|
| Cell line source(s) | The MC38-OVA tumor cell line was obtained from Pedro Romero's Lab (UNIL). B16-F10 (CRL-6475) tumor cell lines was purchased from ATCC and retroviral transduced to express ovalbumin (B16-OVA). Both were grown as a monolayer in DMEM supplemented with 10% fetal calf serum (FCS), 100 U/mL of penicillin, and 100 ug/mL streptomycin sulphate. The Phoenix Eco retroviral ecotropic packaging cell line, derived from immortalized normal human embryonic kidney (HEK) cells was obtained from ATCC (CRL-3214) and maintained in RPMI 1640-Glutamax media supplemented with 10% heat-inactivated FBS, 100 U/ml penicillin and 100 g/mL streptomycin sulfate. |
| Authentication | Cell lines were not authenticated after purchase from ATCC |
| Mycoplasma contamination | All cell lines were tested for mycoplasma contamination prior to use in experiments |
| Commonly misidentified lines (See ICLAC register) | NA |

# Animals and other organisms

Policy information about studies involving animals; ARRIVE guidelines recommended for reporting animal research

| | |
|---|---|
| Laboratory animals | 6 weeks-old female C57BL/6 mice aged were purchased from Harlan (Harlan, Netherlands) and housed at the animal facility at the University of Lausanne in compliance with guidelines. C57BL/6 OT-1 CD45.1+ and C57BL/6 Pmel were obtained from Pedro Romero's Lab (UNIL). All in vivo experiments were conducted in accordance and with approval from the Service of Consumer and Veterinary Affairs (SCAV) of the Canton of Vaud, Switzerland under the license VD3526. |
| Wild animals | Study did not involve wild animals |
| Field-collected samples | Study did not involve samples collected in the field |
| Ethics oversight | Service de la consommation et des affaires vétérinaires (SCAV) https://www.ge.ch/organisation/service-consommation-affaires-veterinaires |

Note that full information on the approval of the study protocol must also be provided in the manuscript.

# Flow Cytometry

## Plots

Confirm that:

☒ The axis labels state the marker and fluorochrome used (e.g. CD4-FITC).

☒ The axis scales are clearly visible. Include numbers along axes only for bottom left plot of group (a 'group' is an analysis of identical markers).

☒ All plots are contour plots with outliers or pseudocolor plots.

☒ A numerical value for number of cells or percentage (with statistics) is provided.

## Methodology

| | |
|---|---|
| Sample preparation | Tumors were excised and dissociated into single-cell suspension by combining mechanical dissociation with enzymatic degradation of the extracellular matrix using the commercial Tumor Dissociation kit for mouse (Miltenyi, Cat# 130-096-730). Following single cell suspension, 2.5x106 live cells were seeded in 96-well plates and incubated with 50 ul of Live/Dead Fixable aqua dead for 30' in PBS at 25°C, then Fc receptors were blocked by incubation for 30 min. at 4°C with 50 ul of purified anti-CD16/CD32 mAb. Cells were then stained for 30 min at 4°C with the fluorochrome-conjugated mAbs of interest (see above) in 50 ul of FACS Buffer. Subsequently, the cells were washed twice, and fixed/permeabilized using the FoxP3 transcription factor staining buffer set for intracellular staining. Fluorescence minus one (FMO) controls were stained in parallel using the panel of antibodies with sequential omission of one antibody. Precision Count Beads™ (Biolegend, Cat# 424902) were used to obtain absolute counts of cells during acquisition on the flow cytometer. |
| Instrument | A BD FACSAriaIII instrument or a A BD FACSAriaII instrument were used for cell sorting and a BD LSR II instrument with FACSDiva software v8.0.1 to collect data for analysis. |
| Software | FACSDiva software v8.0.1(BD) was used on LSR II instrument for data collection; data was analyzed on FlowJo v10.7.1 (TreeStar) and statistical analysis performed on Prism v9.0 (GraphPad) |
| Cell population abundance | Sorted samples had purity > 95% as confirmed by re-sampling after sorting. |

Gating strategy

FSC-A/SSC-A was used to gate on cells. Doublets were excluded through SSC-H/SSC-W and FSC-H/FSC-W. Dead cells were excluded with Live/Dead Fixable aqua. CD8 T cells were gated as CD8 positive. OT1 transferred cells were identified using the congenic marker CD45.1. A representative general gating strategy is depicted in Extended Data Figure 3a.

☒ Tick this box to confirm that a figure exemplifying the gating strategy is provided in the Supplementary Information.

