## [Peer Review File · Nature Immunology]

Peer Review Information

Journal: Nature Immunology

Manuscript Title: Orthogonal Cytokine Engineering Enables Novel Synthetic Effector States Escaping Canonical Exhaustion in Tumor-rejecting CD8+ T Cells

Corresponding author name(s): Professor George Coukos, Dr Jesus Corria-Osorio

Editorial Notes:

Redactions – unpublished data Parts of this Peer Review File have been redacted as indicated to maintain the confidentiality of unpublished data.

Reviewer Comments & Decisions:

Decision Letter, initial version:

4th May 2022

Dear George,

Thank you for providing a point-by-point response to the referees' comments on your manuscript entitled "Orthogonal Gene Engineering Enables Novel Synthetic States of Powerful Tumor-rejecting CD8+ T Cells Escaping Canonical Exhaustion". As noted previously, while they find your work of considerable potential interest, they have raised quite substantial concerns that must be addressed. In light of these comments, we cannot accept the current manuscript for publication, but would be very interested in considering a revised version along the lines proposed in your response letter. Thus, we invite you to submit a substantially revised manuscript.

Specifically, the revision should include new experiments to address:

- (1) test anti-tumor responses using a bicistronic vector expressing both IL-2^v and IL-33
- (2) test infusion of soluble cytokines as compared to the engineered CAR-T cells
- (3) further examine the role of the ST2/IL-33 axis on the engineered CAR T cells
- (4) add new bioinformatics analysis of data in-hand, focusing on the metabolic fitness of the cluster 5 CD8+ T cells as compared to exhausted T cells
- (5) address the functional "non-role" played by PD-1 signaling in the cluster 5 T cells
- (6) characterize the molecular effects of IL-2^v on the engineered CD8+ T cells

Regarding the role of ST2, given that ST2 is expressed by multiple cell populations (according to

ImmGen - including ILC2, mast cells and CD4 Treg cells), I think deciphering which population of ST2+ cells is responsible for the in situ anti-tumor effects evoked by the IL-33-expressing CAR-T cells might require additional experiments beyond the scope of the current study. Hence, no need to attempt to delete Il1r1 (St2) in the engineered CAR-T cells.

Please include the additional textual clarifications as indicated in your response letter.

When you revise your manuscript, please take into account all reviewer and editor comments, please highlight all changes in the manuscript text file in Microsoft Word format.

Please bear in mind that we will be reluctant to approach the referees again in the absence of major revisions.

* If you have not done so already please begin to revise your manuscript so that it conforms to our Article format instructions at <http://www.nature.com/ni/authors/index.html>. Refer also to any guidelines provided in this letter.

The Reporting Summary can be found here:

When submitting the revised version of your manuscript, please pay close attention to our [href="https://www.nature.com/nature-portfolio/editorial-policies/image-integrity">Digital Image Integrity Guidelines.](https://www.nature.com/nature-portfolio/editorial-policies/image-integrity) and to the following points below:

[REDACTED]

If you wish to submit a suitably revised manuscript we would hope to receive it within 6 months. If you cannot send it within this time, please let us know. We will be happy to consider your revision so long as nothing similar has been accepted for publication at Nature Immunology or published elsewhere.

Nature Immunology is committed to improving transparency in authorship. As part of our efforts in this direction, we are now requesting that all authors identified as 'corresponding author' on published papers create and link their Open Researcher and Contributor Identifier (ORCID) with their account on the Manuscript Tracking System (MTS), prior to acceptance. ORCID helps the scientific community achieve unambiguous attribution of all scholarly contributions. You can create and link your ORCID from the home page of the MTS by clicking on 'Modify my Springer Nature account'. For more information please visit www.springernature.com/orcid.

Thank you for the opportunity to review your work.

Kind regards,

Laurie

Laurie A. Dempsey, Ph.D.
Senior Editor
Nature Immunology
l.dempsey@us.nature.com
ORCID: 0000-0002-3304-796X

Referee expertise:

Referee #1: T cell engineering

Referee #2: T cells

Referee #3: Tumor immunotherapy

Reviewers' Comments:

Reviewer #1:

Remarks to the Author:

In this study from Coukos and colleagues, the investigators asked if they could overcome tumor-induced CD8 T cell exhaustion using orthogonal gene-engineering. They engineered OTI CD8 T cells to express a PD-1 decoy molecule (PD1d), a previously described IL-2 mutein (IL-2V) that is reported not to bind the high affinity IL-2R (mutated contacts for IL-2Ralpha), or the proinflammatory alarmin IL-33 alone or in combination (PD1d/IL2V, PD1d/IL33, a 1:1 mix of PD1d/IL2V+PD1d/IL33 = PD1d/2V/33, or finally just IL2V/IL33). In the experiments that are central to their conclusion, these cells were tested for their anti-tumor activity against B16-OVA and profiled at days 5, 8, or 12 post adoptive cell transfer (ACT) into recipients by flow cytometry and 10x genomics. The main take home is that CD8 T cells expressing just IL2V and IL-33 (PD1d appears not to matter) differentiate to a novel synthetic effector (Tse) phenotype through a novel synthetic precursor (Tsp) phenotype, neither of which have been reported for mouse CD8 T cell or human TILs. In addition, these orthogonal engineered cells can cause endogenous CD8 T cells to differentiate to the Tse phenotype regardless of if they express a high or low affinity TCR for the tumor antigen. Of note, the Tse are polyfunctional, with hallmark expression of granzyme C, and are Tox low/neg and Tcf1 neg indicating that they do not adopt a classic exhaustion phenotype compared with non-orthogonal ACT control populations. Importantly, the cells infiltrated the tumor and took up residence in the host without any conditioning of the host, suggesting a potential therapeutic strategy for ACT (be it CAR-T or TCR-T or TIL transfer).

Experiments (2 on scale of 1-3):

- The experiments are generally well controlled and with appropriate numbers to allow for statistical analysis for tumor control and flow cytometry analysis. The immune profiling presented here is consistent with the state of the field and uses reference data sets to establish the novelty of the Tse and Tsp populations.

Completeness (2 on scale of 1-3):

- Overall the data are numerous, of reasonably high quality, and make for a reasonably complete interrogation of the question being addressed. Weaknesses include:
 - o Tumor rejection does not appear to be complete and the experiments are terminated at ~d25 just as the tumor appears to grow again (e.g. Fig 4g,h). TIL and dLN profiling of ACT OTI cells at this later timepoint could be useful to determine if the Tse ultimately adopt an exhausted phenotype (as with the endogenous TILs at d26 in Fig 6). In addition, tumor profiling may be useful to determine if tumor outgrowth is due to selection of a variant that is, for example, OVA negative.
 - ♣ In short, the durability of orthogonal ACT treatment is not addressed in the experiments performed here.
 - o For figure 3 it is unclear how many T cells were transferred for ACT.
 - o For 10x genomics data it is unclear if the data represent single cell analysis from one mouse? Were multiple mice per group pooled prior to sorting? Was the analysis repeated on different days or multiple mice on the same day? It is unclear how robust these data are if the analysis stems from a single mouse.
 - o Minor: Mouse T cells typically all express CD44, just at different levels, and therefore classifying cells as CD44+ is uninformative. Mouse T cells typically have a distribution of CD44 low to intermediate (typically antigen-inexperience/naïve cells) as well as a clear CD44 hi population that is antigen-experienced. The data in Figure 4a are curious because the endogenous PD1+ TILs are CD44hi, whereas the PD1+ OTI TILs are CD44int. This is interesting and perhaps worthy of note.

o Minor: The evidence that IL-2V does what they claim at a molecular level is based on indirect evidence from prior studies. It has not been carefully interrogated through direct binding analysis. Doing so is, in my opinion, beyond the scope of this study but it important to note.

Reproducibility (2 on scale of 1-3):

- The tumor control and flow cytometry experiments typically involve ~5mice/grp and are repeated 3x. The presentation of the data suggest that the data from all three experiments are then aggregated for statistical analysis. A clearer statement on this point would aid in the evaluation of the work.

Overall it looks sound.

o The figure legend for Fig 4d indicates 34 mice were analyzed, with each column representing a mouse, however I count 39 columns. This discrepancy should be clarified.

- As noted above, clarity is needed on the nature of the 10x genomics data. Is the data from one mouse?

Scholarship (2 on scale of 1-4):

- There is a lot of data here and it is not clear that all of it is necessary for the bottom line. It becomes hard to follow, particularly when the supplemental figures do not always track in the callouts with the main figures.

o To make the manuscript more accessible, it would benefit from more attention to a clear definition of terms and a more uniform use throughout. The average reader has not thought about this content as much as the authors and cannot easily intuit what is being referred to without clearer definitions and a more consistent use of terms. Rather than using 'orthogonal ACT', consistent and specific notation of the treatment would make the manuscript easier on the reader.

- There is little note in the discussion of how these data relate to other orthogonal engineering studies (e.g. mutated receptor and cytokine pairs, etc...).

- Presumably, the immune system did not evolve to allow CD8 T cells to differentiate to the Tse or Tsp phenotypes for a reason. It may be because there is a fitness cost for the host during chronic infection and evolution made a trade-off that makes immunosurveillance for tumors less advantageous.

Unintended consequences seem like a very real possibility that is not addressed.

o There is no data to suggest that weight, or ruffled fur, or cytokine release syndrome (CRS), or general inflammation were monitored. For translational purpose, these will ultimately have to be considered.

Novelty (2 on scale of 1-4):

- As noted, the Tse and Tsp populations appear to represent novel differentiation states that have not previously been described. This non-native biology is certainly interesting given their enhance tumor control activity. It seems safe to consider this an important first step in exploring the translational applications of this work and the findings are likely to be of reasonably broad interest for T cell biologists and people involved in developing T cell therapies.

Extensibility (2.5 on scale of 1-4):

- Due to questions about durability of the phenotype of the orthogonal ACT cells, and unintended consequences, it is unclear at this juncture how extensible the results reported here may be for translational efforts.

Reviewer #2:

Remarks to the Author:

While adoptive T cell therapy is one of the most powerful current immunotherapies, there are many hurdles including limited persistence and function. To improve the efficacy of adoptive T cell therapy, the authors engineered an IL-2 variant binding to IL-2R $\beta\gamma$ (IL-2V), IL-33 or both. The authors show that ACT with T cells expressing IL-2V or IL-33 (PD1d/2V/33-ACT) reprogram CD8 cells. These CD8 cells acquire a unique phenotype that is distinct from exhaustion and termed T-synthetic cell state (TSE) that is characterized by PD1 and GZMC expression and potent cytolytic and effector function. Interestingly, although CD8 cells with TSE state express PD-1 and TIM3, expression of these exhaustion associated molecules does not have functional consequences. Transcriptome analyses show that TSE cells are polyfunctional non-exhausted effector cells with potent antitumor properties. TSE reprogramming is induced locally within tumor and also has impact on endogenous cells by promoting their acquisition of TSE like phenotype. More important, ACT of engineered CD8 cells during tumor progression allow excellent engraftment and inhibition of tumor growth even without need of lymphodepletion or cytokine provision that are generally required for improved efficacy in conventional ACT approaches.

This study therefore shows that CD8 cells can be engineered to promote in vivo persistence and functional activity that deviate their phenotype from exhaustion to sustained activation. This is an important finding that deserves attention from the field of the cancer immunotherapy where functional maintenance of ACT T cells remains a challenge. While the current ms provides a substantial amount of important information on the new findings, the following points should be addressed before publication.

Specific comments

1. The authors show an expansion of TCF1⁺ and TCF1⁻ cells upon PD1d/2V/33-ACT, which seems to be dLN independent and can be attributed to the intratumoral expansion of ACT T cells. It may be important to show whether the ratio of TCF1⁺ and TCF1⁻ observed within OT-1 cells within tumor is fixed prior to ACT. It may be also important to analyze the baseline TCF1 phenotype by each PD1d/2V and PD1d/33 OT-1 cells.
2. Although the authors used engineered CD8 cells to test the efficacy of PD1d, IL-2v and IL-33, I wonder whether treatment of tumor bearing mice with a-PDL1, IL-2 mutant (binding only to IL2R $\beta\gamma$) and IL-33 injection along with OT-I transfer would induce a similar synthetic effector state by OT-I cells within tumor.
3. To analyze the anti-tumor cytolytic activity and effector function of OT-I cells with TSE state, the authors compared OT-I cells isolated from PD1d/2v/33-ACT mice and OT-I cells induced in vitro for Tse state. This may not be a fair comparison, because multiple variations exist between these two samples in addition to the defined differentiation phenotype. It may be better to compare effector function of OT-I cells isolated from UT, PD1d/2v or PD1d/33-ACT groups.
4. The authors show highly increased expression of GZMC by PD1⁺TCF1⁺TOX1^{low} cells generated by PD1d/2V/33-ACT compared to PD1d/2v and PD1d/33-ACT. Since GZMC is the feature of CD8 cells with TSE state after PD1d/2v/33-ACT and the majority of CD8 cells with TSE state transits to TCF1⁻ cells, it is important to also show GZMC expression by TCF1⁻ cells from their different conditions. The point here is whether the unique feature of CD8 cells with TSE state is mainly contributed by TCF1⁺ cells or both TCF1⁺ and TCF1⁻ subsets.
5. To test the contribution of endogenous CD8 cells to tumor control after PD1d/2V/33-ACT, the

authors compared the ACT efficacy in WT and p14 Tg mice. P14 Tg CD8 cells do not express TCR that can recognize TAA. However, this system also lacks other immune components including CD4 cells that can exert helper function. Therefore, this experiment would not provide the evidence that endogenous CD8 contribution is important for the efficacy of PD1d/2v/33-ACT.

6. The authors show that IL-2v is mainly involved in downregulation of TOX by Tex cells and IL-33 (only together with IL-2v) contributes to the GMZC expression by CD8 cells. IL-2v may have a direct effect on activated CD8 cells that express IL-2R $\beta\gamma$, however, it is not clear how IL-33 would work in this system. Its receptor ST2 is expressed by multiple cell types. Therefore, it is important to show which cell type expresses ST2 and also discuss the potential mechanisms of how combinations of IL-2v and IL33 can have this synergic effect to improve anti-tumor function by CD8 cells.

Reviewer #3:

Remarks to the Author:

In this manuscript, Corria-Osorio et al. 'orthogonally' engineer adoptively transferred CD8+ T cells to coexpress three payloads: a previously described IL-2 variant (hypothesised to promote CD8+ T cell stemness through selective lower-affinity interaction with IL2Rb/g), IL-33 (a proinflammatory alarmin (Villarreal et al. 2014) and a secreted PD-1 decoy, in an attempt to augment anti-tumor immunity and programme 'exhaustion-resistance'. Justification for this selection of genetic payloads was relatively superficial; however using the OT1 adoptive transfer model treating Ova-expressing syngeneic B16 tumours, the co-administration of mixed transfers of orthogonally engineered T-cells expressing A) the PD1-decoy with the IL-2 variant, and B) the PD1-decoy with IL-33, resulted in substantially greater tumour control compared to non-transduced control cells, or cells expressing single or pairwise combinations of these genetic payloads. This was proposed to be a consequence of in-situ reprogramming of exogenous and endogenous CD8+ T cells away from traditional T cell exhaustion and towards a proposed TOX- GMZC+ 'synthetic' state, characterised by exhaustion resistance, greater cytotoxic function dependant on the secretion of both IL-33 and IL-2v from adoptively transferred cells. The limitation on any given transferred cell population expressing only 2 of the 3 payloads appears technical, since bi-cistronic retroviral vectors were used.

Convincing evidence was given for greater expansion of orthologously engineered CD8+ T-cell within the tumor site compared to untransduced controls. Evidence presented through pharmacological blockade of lymph node egress suggests that this response does not require continuous T cell recruitment from lymph nodes. This, in combination with increased effector function could be driving the tumor regression observed. Interestingly, enhanced function of the TOX-GMZC+ 'synthetic' state was observed despite expression of the co-inhibitory/exhaustion molecule PD-1, and the authors claim that PD-1 blockade on these cells did not further augment their function - this is used to support the claim that the cells are 'exhaustion resistant'. Furthermore, the TOX- PD-1+ phenotype of the cell state induced suggests an uncoupling of TOX dependency for expression of PD-1.

This study provides an interesting translational approach that capitalises on pre-existing immunotherapeutic candidates to improve adoptive transfer. However, the paper is overly descriptive, mechanistic insights into how the IL-2 variant and IL-33 synergise are lacking, it is unclear whether PD-1d plays any functional role in the triple-therapy (so why is it included?) and the claim that the cells are 'exhaustion-resistant' are poorly substantiated.

Major points

In Fig 1 it is suggested that all three of the PD1-decoy, IL-2 variant and IL-33 are required to drive the augmented adoptive immunotherapy response. However, it is unclear from the figure whether PD-1d contributes at all to the response. In particular, in Fig 1b, the combination of IL-33 and IL-2v in the absence of PD-1d is not shown - but essential in interpreting the data, and the importance of 'orthogonal' gene delivery. Fig 1b also lacks sufficient mice in some groups (IL-33 and IL2) to reliably demonstrate ineffectiveness of control (single- or double transduced) groups - if these are insufficiently powered, they do not serve as adequate negative controls for the superior functionality of the triple orthogonal therapy.

In Fig 4h, the authors show that PD-1d is not actually required in its decoy mutant form for efficacy, which somewhat undermines the argument of the paper, that indeed all three 'orthogonal' genetic payloads are required for efficacy, as suggested in Fig 1b. If it is not required, why is the effect of all three genetic alterations studied in the paper, rather than the two functional payloads?

Fig 4g is used as evidence that the triple-engineered cells are 'exhaustion resistant' despite expressing PD-1, since anti-PD-L1 treatment is not seen to improve the immunotherapy response. However, a critical positive control for anti-PD-L1 functionality in Fig 4g is not present - does anti-PD-L1 blockade synergise with the transfer of untransduced OT-1 cells? This would be an important control to support the hypothesis.

The claim that PD-1, despite being expressed, is not transducing an inhibitory signal is potentially interesting, but almost completely mechanistically unelucidated. Comparing CD8+ T cells in conventional exhausted and synthetic 'exhaustion-resistant' states *in vitro*, can the authors show that the provision of PD-L1 *in vitro* to exhaustion resistant cells does not result in inhibition of signalling and effector functions, compared with control exhausted cells? The authors should also provide a mechanistic basis (and indeed better experimental evidence) for the proposed exhaustion resistance, since it is hard to understand how PD-L1 would be rendered non-functional as a proximal inhibitory signalling molecule despite its expression on these cells.

The manuscript would benefit from much better mechanistic understanding of the specific contributions of IL-33 and IL-2v to anti-tumour responses. Endogenous TIL reprogramming seems to be largely IL-33 driven, what is the mechanistic basis for this? Do the endogenous TIL directly respond to IL-33, do they express ST-2, or is there an intermediate cell type involved? Do the transferred cells become ST-2 positive? Does ablation of ST-2 on either transferred cell populations or host cells ablate the response?

What proportion of the effects are cell intrinsic vs cell extrinsic - ie can the phenotypes of cells expressing A) the PD1-decoy with the IL-2 variant, and B) the PD1-decoy with IL-33 in the mixed transfer be distinguished and compared by flow cytometry?

Figure 3c is confusing, since the synthetic state of exogenous orthogonally engineered TIL is distinct from that of the endogenous TIL (the latter which overlapped with exhausted and effector memory cells) - yet figure 6 shows reprogramming of endogenous TIL towards the TOXlow Gzmzhigh state following ACT with orthogonal engineered IL-33 IL2v orthogonal cells.

(Figure 5a) GzmC is not detected in the effector memory cluster and only in the 'synthetic state' cluster - is this expected? To what extent is this effector molecule co-regulated with GzmB?

PD-1 is not only a marker of exhausted cells, but an activation marker, upregulated on recently stimulated non-exhausted cells. Given the observed 'synthetic' state is TOX negative, to what extent is this merely activation-driven PD-1 expression? In which case, does this rather represent activation-driven co-inhibitory receptor expression rather than exhaustion-resistance? Evidence should be provided that this is a distinct state from that merely reflecting activation.

Does exogenous (eg intraperitoneal) administration of IL-2v and IL-33 cause a similar anti-tumour response to orthogonal gene-engineered T cell transfer?

To what extent do IL-2v and IL-33 need to be expressed on different co-transferred cells, as is presented here? If they are co-expressed in a bicistronic vector, does this also result in potent tumor immunity. PD-1d may not be required if the cells are indeed 'exhaustion-resistant'

The Methods state, 'Finally, engineered OT-1 T cells were adjusted based on the PD-1. IgG4 decoy expression prior to cell transfer.' What does this mean? Clarity is needed on exactly how transduced cells were prepared and adjusted before transfer

The authors state: On day 12 and 15 mice were treated with i.v transfer of 5×10^6 gene-engineered CD44+ CD62L+ TCF1+ OT-1 T cells or control nontransduced OT-1. It is not possible for live cells to be separated and transferred on the basis of intracellular TCF1 positivity - a more accurate description of how cells were isolated before transfer, and a measurement of their expression of CD44, CD62L, and TCF1 upon transduction of each of the constructs prior to transfer, would enable a more thorough assessment.

Minor point: Typo on Line 81 - (100 m3) tumors

Author Rebuttal to Initial comments

Reviewer #1

(Remarks to the Author)

In this study from Coukos and colleagues, the investigators asked if they could overcome tumor-induced CD8 T cell exhaustion using orthogonal gene-engineering. They engineered OTI CD8 T cells to express a PD-1 decoy molecule (PD1d), a previously described IL-2 mutein (IL-2V) that is reported not to bind the high affinity IL-2R (mutated contacts for IL-2Ralpha), or the proinflammatory alarmin IL-33 alone or in combination (PD1d/IL2V, PD1d/IL33, a 1:1 mix of PD1d/IL2V+PD1d/IL33 = PD1d/2V/33, or finally just IL2V/IL33). In the experiments that are central to their conclusion, these cells were tested for their anti-tumor activity against B16-OVA and profiled at days 5, 8, or 12 post adoptive cell transfer (ACT) into recipients by flow cytometry and 10x genomics. The main take home is that CD8 T cells expressing just IL2V and IL-33 (PD1d appears not to matter) differentiate to a novel synthetic effector (Tse) phenotype through a novel synthetic precursor (Tsp) phenotype, neither of

which have been reported for mouse CD8 T cell or human TILs. In addition, these orthogonal engineered cells can cause endogenous CD8 T cells to differentiate to the Tse phenotype regardless of if they express a high or low affinity TCR for the tumor antigen. Of note, the Tse are polyfunctional, with hallmark expression of granzyme C, and are Tox low/neg and Tcf1 neg indicating that they do not adopt a classic exhaustion phenotype compared with non-orthogonal ACT control populations. Importantly, the cells infiltrated the tumor and took up residence in the host without any conditioning of the host, suggesting a potential therapeutic strategy for ACT (be it CAR-T or TCR-T or TIL transfer).

Experiments (2 on scale of 1-3):

- The experiments are generally well controlled and with appropriate numbers to allow for statistical analysis for tumor control and flow cytometry analysis. The immune profiling presented here is consistent with the state of the field and uses reference data sets to establish the novelty of the Tse and Tsp populations.

Completeness (2 on scale of 1-3):

- Overall the data are numerous, of reasonably high quality, and make for a reasonably complete interrogation of the question being addressed. Weaknesses include:

R1-Q1 :Tumor rejection does not appear to be complete and the experiments are terminated at ~d25 just as the tumor appears to grow again (e.g. Fig 4g,h). TIL and dLN profiling of ACT OTI cells at this later timepoint could be useful to determine if the Tse ultimately adopt an exhausted phenotype (as with the endogenous TILs at d26 in Fig 6). In addition, tumor profiling may be useful to determine if tumor outgrowth is due to selection of a variant that is, for example, OVA negative. In short, the durability of orthogonal ACT treatment is not addressed in the experiments performed here.

Response: We thank the Reviewer for raising this important point. We have performed additional characterization of tumor escape that is now included in the revised manuscript. These data show that tumor progression is not associated with TIL exhaustion (**New Extended Fig. 8**), which is per se an important finding. Below is a detailed account of this data.

Importantly, tumor progression was not linked to evolution toward T-cell exhaustion in the transferred cells (**New Extended Fig. 8a**). Indeed, cells migrated to a synthetic memory-like state which was Tox^{neg} TCF1^{int} PD-1^{neg} GzmC^{neg}. Thus, exhaustion was not the reason for therapeutic failure.

Importantly, tumor escape variants downregulated the surface expression of MHC-I as early as 12 days post ACT (**New Extended Data Fig. 8d**), indicating that they became insensitive to CD8 T cell mediated attack, attesting to the potency of engineered cells and the plasticity of tumor cells. Furthermore, we attribute the therapeutic failure partly to the activation of counteracting regulatory mechanisms in the tumor microenvironment. For example, although the ratio of CD8⁺:Treg cells was significantly increased with orthogonal ACT, IL-2^V did increase the absolute number of Treg cells (**New Extended Data Fig. 8e**),. We thus tested the effect of depleting CD4⁺ T cells prior to PD1d/2v/33-ACT. We found that antibody-mediated

depletion of CD4⁺ T cells significantly improved tumor control and mouse survival (**New Extended Data Fig. 8f**), indicating an additional mechanism contributing to tumor escape.

R1-Q2 : For figure 3 it is unclear how many T cells were transferred for ACT.

Response: We transferred 5 million in each time point. We provide this detail in the material & methods section of the revised manuscript.

R1-Q3 : For 10x genomics data it is unclear if the data represent single cell analysis from one mouse? Were multiple mice per group pooled prior to sorting?

Response: Tumors of five mice per group were pooled prior to sorting, to recover as many T cells as possible and gain in resolution. This has been clarified in the material & methods section of the revised manuscript.

Was the analysis repeated on different days or multiple mice on the same day? It is unclear how robust these data are if the analysis stems from a single mouse.

Response: The analysis was done with multiple mice pooled and analyzed per condition on the same day across the different time points. We and others have previously shown that there is no significant difference in terms of cell state composition of CD8 T cells among individual mice that belong to same experimental group, at a specific time point (1-4). Finally, it is important to note the C5-T_{SE} state was detected in two independent experiments analyzed by scRNAseq, and findings were further validated in several experiments by FACS, which supports the robustness of our observations.

R1-Q4 Minor: Mouse T cells typically all express CD44, just at different levels, and therefore classifying cells as CD44⁺ is uninformative. Mouse T cells typically have a distribution of CD44 low to intermediate (typically antigen-inexperience/naïve cells) as well as a clear CD44^{hi} population that is antigen-experienced. The data in Figure 4a are curious because the endogenous PD1⁺ TILs are CD44^{hi}, whereas the PD1⁺ OTI TILs are CD44^{int}. This is interesting and perhaps worthy of note.

Response: We thank the Reviewer for this observation. It is true that classifying OT1 cells as CD44⁺ per se is uninformative since all OT1 are CD44. However, the point of the figure was not related to CD44 expression but rather to show that during tumor regression PD-1⁺ OT1 cells were mostly TOX^{neg} Gzmc C5-like cells, as predicted in the scRNAseq analysis. We added TILs from untreated baseline tumors as control, to show that these antigen-experienced (CD44^{high} PD-1⁺) endogenous CD8⁺ TILs are largely TOX⁺ and Gzmc^{low/neg} consistent with canonical exhausted cells.

As the Reviewer suggested, we highlight the observation about different levels of expression of CD44 in the revised manuscript (Lines 299-300).

R1-Q5 Minor: The evidence that IL-2V does what they claim at a molecular level is based on indirect evidence from prior studies. It has not been carefully interrogated through direct binding analysis. Doing so is, in my opinion, beyond the scope of this study but it important to note.

Response: The molecule that we used to engineering T cells is the same that have been characterized by Rojas et al (5), where the authors confirmed the lack of binding to CD25 due to the introduced mutations.

We have conducted additional extensive analyses of our *in vivo* scRNAseq data, which we have included in the revised manuscript. These provide novel insights about the cellular and transcriptional reprogramming induced by IL-2v alone on gene-engineered CD8⁺ T cells *in vivo*. Below is a detailed account of this data:

To characterize the T-cell intrinsic effects induced by IL-2v in gene-engineered CD8 T cells we interrogated FACS-sorted OT1 TILs harvested 12 days post IL-2^V-OT1 ACT (**New Fig. 7a top**). These were mostly distributed in a synthetic C7-memory-like (ML) cell state (**New Fig. 7a bottom part**) characterized by high expression of T-cell activation markers *IL2Ra*, *Cd69*, *Tnfrsf9*, the stemness-related *Ccr7* and *Xcl1* chemokine, a molecule involved in migration to dendritic cells areas, low expression of *Pdcd1*, *Lag3*, *Havcr2* and *Tigit*, and no expression of *Tox*, *Gzmc*, *Cx3cr1* or *Klrg1* (**Fig. 7b**). Although similar to some degree to previously described canonical memory-like TIL states (P_{EX} and effector-memory) (**New Extended Data Fig. 7a**), C7-ML TILs overexpressed genes involved in ribosome biogenesis, glycolysis, ATP generation and nucleotide metabolism (**New Fig. 7c**), suggesting a functional role of IL-2v in upregulating these cellular processes. Importantly these processes were also prominent in C5-T_{SE} and C6-T_{SP} cells (**Fig. 4 and Fig. 6**), the other two states achieved when OT1 cells were engineered with IL-2v and with IL-33, thus common to all states driven by the common presence of IL-2v.

We validated by flow cytometry that most IL-2v-OT1 TILs were activated, with high expression of CD25, CD69 and 4-1BB, but arrested in a TOX^{neg}TCF1^{int} PD-1^{low/neg} C7-like cell state, with no transition to C5-T_{SE} cells, as clearly heralded by the absence of expression of *Gzmc* (**New Fig. 7d**). We conclude that IL-2v reprograms T cells into a new memory-like state. Based on our findings, we deduce that unlike canonical memory cells, which exit stemness and commit to effector (or exhaustion) differentiation upon TCR activation, C7 cells remain committed to memory/stemness even upon TCR engagement, and only transition to C6 T_{SP} and subsequently C5-T_{SE} upon additional potent signals triggered *in vivo* by (IL-33 driven) inflammation, the latter likely due to myeloid cues. This point was detailed in the Discussion (see. lines 518-535).

We next sought to infer the specific transcriptional wiring induced by IL-2v alone, we identified 181 transcription factors (TFs) downstream of IL-2R from the ligand-TF matrix of Nichenet (6), a computational method that predicts ligand–target links by combining expression data with a prior knowledge-based mathematical model on signaling and gene regulatory networks, and used SCENIC (7), a computational method for gene-regulatory network (GRN) reconstruction to infer their activity across TIL states. Unbiased clustering analysis revealed 12 groups of differentially active TFs/regulons (i.e. the TFs and their target genes) across synthetic (C7, C5-T_{SE}, C6-T_{SP}) and canonical TIL states (C2-EM, C4-P_{EX}, C4-T_{EX}) (**New Fig.**

7e). Among them, G7 was predominantly active in C7/memory-like cells, and G9 in all three synthetic cell states where IL-2v was implicated (**New Fig. 7g**). Orthogonal partial least squares discriminant analysis (OPLS-DA) (8), a computational approach to discriminate between two or more groups (classes) using multivariate data, indeed revealed TFs from G7 as having the highest statistical power to discriminate between C7-TILs and other states (**New Extended Data Fig. 7b**). Similarly, G9 contained TFs with the highest discriminant power for the three synthetic cell states driven by IL-2v relative to canonical TIL states (**New Extended Data Fig. 7c**). Functional annotation of the target genes of the top 25% most discriminant TF/regulons of G7 revealed that this set is involved in the regulation of RNA metabolism, mRNA splicing, ribosome biogenesis, etc. while the target genes of the top 25% most discriminant TF/regulons of G9 were involved also in the regulation of protein metabolism and translation, in addition to RNA metabolism (**New Fig. 7f, Extended Data Fig. 7d,e**), revealing how IL-2v signaling positions the cell in a memory-like state of transcriptional/translational readiness, while progression to C6-T_{SP} and C5-T_{SE} engages further programs of synthesis to enter the full effector state. Notably, 50% of these TFs (14/28) formed a highly connected sub-network motif in the SCENIC-inferred gene regulatory network (**Fig. 7g**), revealing coordinated transcriptional regulation.

Thus, IL-2v engineered into CD8⁺ T cells, transferred in the absence of lymphodepletion, can regulate two different sets of IL-2R-related TFs: a first group (*Fosf11, Egr1, Tbx21, Nfkb1, Irf4, Myc*) specifically active in C7/memory-like synthetic TILs in the absence of IL-33, where they sustain cell expansion and persistence, stemness and memory/readiness, while a second group (*Irf8, Atf3, Atf4, Smarca4, Batf3*) active across all three synthetic states, seems involved in the metabolic support of sustained gene expression and protein translation, both highly energetically costly cellular processes (9, 10), which are dampened in canonical exhausted CD8 T cells (10). Notably, the AP-1 transcription factors *Fosf11* has been associated with long-term persistence of CD8 T cells in the context of chronic viral infections(11) and *Batf3* has been recently shown as a critical transcription factor for CD8 T-cell memory formation and survival via negative regulation of the proapoptotic factor BIM (12). Indeed, T cells that lacked *Batf3* succumbed to an aggravated contraction and had a diminished memory response(12). Strikingly, we detected important target-gene overlap between G7 (*Egr1, Tbx21, Nfkb1, Irf4, Myc*) and TFs in G9 (*Irf8, Atf3, Batf3*) and (**New Fig. 7h**), explaining how the alternate cell fates C7-ML vs. C6-T_{SP}/C5-T_{SE} can be structured to share important core metabolic programs through alternate TF regulatory networks, which ensures their respective differentiation.

Collectively, this new analysis indicates that IL-2v alone contributes to reprogram gene-engineered tumor specific TILs to escape TOX⁺ canonical exhaustion, endows them with superior transcriptional metabolic programs, and positions them in an novel, activated, memory-like state post-ACT, which can persist in the absence of TOX. This is a novel and unexpected finding, since TOX expression is required for persistence under similar conditions (13-16).

We went on to extensively characterize the effects of IL-2^V on mouse T cells at a molecular level *in vitro*. These data are quite extensive and, as the Reviewer proposes, are beyond the scope of the present manuscript. Therefore, we propose to submit them as a separate manuscript, but we offer them below for the Reviewer.

In summary, we learned that IL-2^V occupies the IL-2R differently than wild-type IL-2, or IL-15, and provides a different type of stimulation to mouse T cells (lower level and tonic activation), thereby driving a unique

signaling in CD8⁺ cells, which leads to decoupling of proliferation from differentiation. This allows CD8⁺ cells to activate, proliferate and simultaneously maintain stemness and powerful effector potential. We have dissected the signaling pathways and metabolic underpinnings of IL-2^V versus IL-2 or IL-15, and have demonstrated that IL-2^V, unlike IL-2, maintains oxidative phosphorylation and mitochondrial biogenesis and fitness

Importantly, in the context of our extensive *in vitro* characterization mentioned above, we confirmed using a well-controlled *in vitro* experimental system and with a new scRNAseq dataset (Fig.R1A) that IL-2v at 50 IU/mL induces activation of 24 out of the 28 regulons predicted above from our *in vivo* OT1 data (Fig.R1B).

Fig. R1: Regulon activity of the 25% top more discriminant IL-2R TFs for in vivo T-cell synthetic states upon in vitro activation/expansion with IL2^V. **A)** Experimental design. **B)** Normalized Regulon Activity inferred applying the SCENIC computational pipeline (7) using a new scRNAseq dataset of CD8 T cells that after three days of activation with aCD3/aCD28 were kept in culture only with IL-2V or wtIL-2 (green: low activity, purple: high activity). In red we show the TFs that were correctly predicted as induced by IL-2V. In blue those that represent false positive hits. In black randomly selected TFs. The Gata2 Regulon was not detected in the second scRNAseq dataset (regulon activity=0).

Reproducibility (2 on scale of 1-3):

R1-Q6: The tumor control and flow cytometry experiments typically involve ~5mice/grp and are repeated 3x. The presentation of the data suggest that the data from all three experiments are then aggregated for statistical analysis. A clearer statement on this point would aid in the evaluation of the work. Overall it looks sound.

Response: We thank the Reviewer for this kind evaluation of our experimental design. We have added this in each figure's legend.

R1-Q7 The figure legend for Fig 4d indicates 34 mice were analyzed, with each column representing a mouse, however I count 39 columns. This discrepancy should be clarified.

Response: We thank the Reviewer for this observation. It was fixed.

R1-Q8 As noted above, clarity is needed on the nature of the 10x genomics data. Is the data from one mouse?

Response: This was answered above in R1-Q3.

Scholarship (2 on scale of 1-4):

- There is a lot of data here and it is not clear that all of it is necessary for the bottom line. It becomes hard to follow, particularly when the supplemental figures do not always track in the callouts with the main figures.

Response: We thank the Reviewer for this remark. Indeed, we fully agree with this comment. For this reason we decided to focus the revised manuscript in describing the new biology within the exogenous OT1 TIL compartment and leave the characterization of the transcriptional reprogramming within the endogenous CD8⁺ TIL compartment for a follow-on manuscript . We think this has improved the clarity of the manuscript and drives the key take-home messages.

R1-Q9: To make the manuscript more accessible, it would benefit from more attention to a clear definition of terms and a more uniform use throughout. The average reader has not thought about this content as much as the authors and cannot easily intuit what is being referred to without clearer definitions and a more consistent use of terms. Rather than using 'orthogonal ACT', consistent and specific notation of the treatment would make the manuscript easier on the reader.

Response: We thank the Reviewer for this suggestion. We have followed this recommendation and this term was replaced by PD1d/2v/33-ACT.

R1-Q10: There is little note in the discussion of how these data relate to other orthogonal engineering studies (e.g. mutated receptor and cytokine pairs, etc...).

Response: We thank the Reviewer for this observation. We have now included this very important point in the revised manuscript. Indeed, it has been recently shown that systemic administration of recombinant anti-PD-1_IL-2v bi-specific immunocytokine differentiated tumor-specific CD8⁺ TILs cells into better

effectors (T_{BE}) with antitumor efficacy (17) and (**New Extended Data Fig. 3g**). Even though PD-1_IL2v TILs expressed *Pdcd1* and low levels of *Tox* (**New Extended Data Fig. 3g**), they were still distinct from C5- T_{SE} CD8⁺ TILs induced by orthogonal cytokine cell engineering in our experiments. Indeed, C5- T_{SE} TILs expressed no *Tox*, lower *Tbx21* and *Cx3cr1*, and much higher granzyme levels than PD-1_IL2v TILs (**Fig.3j**).

Similarly, we compared C6- T_{SP} TILs with atypical precursor-like CD8⁺ T cells reported recently following systemic administration of anti-PD-1 in combination with IL-2 (3). Although these cells downregulate *Tox* (3), they were mostly classified as effector-memory (EM)-like cells, thus distinct of C6- T_{SP} TILs, upon projection in the ACT reference map with ProjecTILs (**New Fig. 6k**), and did not express *Gzmc* (**New Extended Data Fig. 6d**).

Regarding the use of orthogonal IL-2 or IL-9 cytokine-receptor complexes to engineer T cells (18, 19), these approaches were not associated either with generation of synthetic states *in vivo*. Indeed, all the characterization of IL-9R signaling using a chimeric orthogonal cytokine receptor (19) was done *in vitro*, using the infusion product. The authors did not provide data about cell states upon cell transfer *in vivo*. We have added this point in the Discussion (Lines 863-86).

Together, our data indicate that both C5- T_{SE} and C6- T_{SP} are novel, synthetic effector and precursor cell states seemingly never seen at the steady state or in the context of conventional immunotherapies in mice or humans (**Fig. 3i**), but massively acquired *in vivo* by exogenous T cells following orthogonal-engineering ACT with PD1d, IL-2v and IL-33.

R1-Q11: Presumably, the immune system did not evolve to allow CD8 T cells to differentiate to the Tse or Tsp phenotypes for a reason. It may be because there is a fitness cost for the host during chronic infection and evolution made a trade-off that makes immunosurveillance for tumors less advantageous. Unintended consequences seem like a very real possibility that is not addressed. There is no data to suggest that weight, or ruffled fur, or cytokine release syndrome (CRS), or general inflammation were monitored. For translational purpose, these will ultimately have to be considered.

Response: We thank the Reviewer for this observation We have not seen signs of toxicity upon orthogonal ACT in more than 200 treated mice (**Fig.R2**). At a gross level, we have not observed any evidence of mouse suffering (reduced mobility, ruffled fur, increased sleepiness) or short-term weight loss (see Fig. R2). Furthermore, in different experiments we have attained tumor eradication and long-term survival when mice were treated with PD1d/2v/33-ACT early or in mice previously depleted of CD4 cells (see Extended Data Figure. 1g and Extended data Figure. 8f). Additional experiments brought to a large proportion of cured mice, when the endogenous TILs contained a high proportion of high affinity OT1 T cells (experiments not shown since we removed the endogenous T cell data). All these cured mice appeared entirely normal and died of old age beyond 300 days.

As a positive control for these experiments, we note that tumor bearing mice treated with recombinant mouse (rm)IL-33, using the lowest dose with reported antitumor efficacy (20) (0.5 μ g/mouse once a day) and recombinant (r)IL-2v (20 μ g/mouse twice a day) (21), starting on day 0 (**Extended Data Fig. 5a**), with or without untransduced OT1 cells, developed severe adverse side effects and the treatment had to be

stopped within 8 days due to signs of toxicity (ruffled fur, reduced animal mobility) that were apparent after each morning's injection with the cytokine cocktail. In sharp contrast, no systemic toxicity was observed when IL-33 and IL-2v were delivered by transferred OT1.

Fig. R2: Orthogonal ACT does not affect the body weight of treated mice. A representative experiment (n=10 mice/group)

Novelty (2 on scale of 1-4):

- As noted, the Tse and Tsp populations appear to represent novel differentiation states that have not previously been described. This non-native biology is certainly interesting given their enhance tumor control activity. It seems safe to consider this an important first step in exploring the translational applications of this work and the findings are likely to be of reasonably broad interest for T cell biologists and people involved in developing T cell therapies.

Response: We thank the Reviewer for this kind evaluation of our work. We are advancing these concepts to the clinic in our center.

Extensibility (2.5 on scale of 1-4):

- R1-Q12 Due to questions about durability of the phenotype of the orthogonal ACT cells, and unintended consequences, it is unclear at this juncture how extensible the results reported here may be for translational efforts.

Response: We thank the review for this remark. Indeed, these are open questions that will require testing in human in a carefully designed clinical study. We are planning a first in human phase 1 study where the cognate human vectors carrying IL-2v, IL-33 or both, could be tested at different doses, and with different levels of lymphodepletion, to test the persistence of engineered cells in human patients, as well as safety and efficacy. In the current manuscript we demonstrated that CD4-depletion induced durable responses in an otherwise non-lymphodepleted host. Therefore, we plan to combine this therapy without proper lymphodepletion and with low-dose cyclophosphamide pre-conditioning regimens in the clinic, a pharmacological approach that preferentially depletes CD4 regulatory T cells (1). We will test whether this approach this will improve the durability of the cells in T_{SE} state and thus the antitumor efficacy of our strategy.

Reviewer #2

(Remarks to the Author)

While adoptive T cell therapy is one of the most powerful current immunotherapies, there are many hurdles including limited persistence and function. To improve the efficacy of adoptive T cell therapy, the authors engineered an IL-2 variant binding to IL-2R $\beta\gamma$ (IL-2V), IL-33 or both. The authors show that ACT with T cells expressing IL-2V or IL-33 (PD1d/2V/33-ACT) reprogram CD8 cells. These CD8 cells acquire a unique phenotype that is distinct from exhaustion and termed T-synthetic cell state (TSE) that is characterized by PD1 and GZMC expression and potent cytolytic and effector function. Interestingly, although CD8 cells with TSE state express PD-1 and TIM3, expression of these exhaustion associated molecules does not have functional consequences. Transcriptome analyses show that TSE cells are polyfunctional non-exhausted effector cells with potent antitumor properties. TSE reprogramming is induced locally within tumor and also has impact on endogenous cells by promoting their acquisition of TSE like phenotype. More important, ACT of engineered CD8 cells during tumor progression allow excellent engraftment and inhibition of tumor growth even without need of lymphodepletion or cytokine provision that are generally required for improved efficacy in conventional ACT approaches.

This study therefore shows that CD8 cells can be engineered to promote *in vivo* persistence and functional activity that deviate their phenotype from exhaustion to sustained activation. This is an important finding that deserves attention from the field of the cancer immunotherapy where functional maintenance of ACT T cells remains a challenge. While the current ms provides a substantial amount of important information on the new findings, the following points should be addressed before publication.

Specific comments

1. The authors show an expansion of TCF1+ and TCF- cells upon PD1d/2V/33-ACT, which seems to be dLN independent and can be attributed to the intratumoral expansion of ACT T cells. It may be important to show whether the ratio of TCF1+ and TCF1- observed within OT-1 cells within tumor is fixed prior to ACT.

Response: We thank the Reviewer for the evaluation of our work. We respectfully note that there are no OT1 T cells within the tumor prior to ACT, because the host mice are not OT1 transgenic mice.

Regarding the OT1 cells that we transferred, we show that all OT1 cells, independently of the transduced payload, expressed TCF1 and most had a central memory-like (TCF1+ CD62L+) phenotype at the end of expansion and prior to ACT (infusion product, shown in **Extended Data Figure 1d**).

To properly address the issue raised by the Reviewer regarding the endogenous tumor-specific CD8 TIL compartment, we generated OT1 bone marrow chimeric mice bearing B16-OVA tumors. These mice were first transplanted with OT1 bone marrow, to generate hosts where we could efficiently track tumor specific T cells, and then implanted them with B16-OVA tumors (**Fig. R3a**). Successfully, in these mice, 10-20% of endogenous TILs at baseline tumors were OT1 cells.

We looked at these endogenous OT1 TILs recovered from B16-OVA tumors from treatment-naïve OT1 chimeric mice. Twelve days post tumor inoculation, such tumor-specific TILs exhibited Pex or Tex features, with a TCF1⁺ to TCF1^{neg} ratio equal to 2:1 prior to ACT (in the PD-1⁺ cells) (**Fig. R3b**). Because of the congenic set-up, we could still follow these cells and distinguish them from exogenously transferred engineered OT1 cells. Notably, under the conditions of orthogonal (PD1d/2v/33) ACT, these endogenous OT1 cells also reprogrammed to T_{SP}-like and T_{SE}-like precursor and effector cells, respectively (**Fig.R3c**). Thus, this analysis suggests that a TCF1⁺ to TCF1^{neg} ratio >1 within the endogenous tumor-specific compartment could be beneficial for orthogonal ACT. Indeed, the PD1d/2v/33-ACT is curative when performed in OT1 or OT3 chimeric mice (**Fig. R3d**). Please note that we did not include these data in the revised manuscript, as we elect to focus on characterizing in greater depth the synthetic states of transferred cells.

Fig. R3: A) Schematics summarizing the procedure to generate OT1 mixed (50:50 with B6) bone marrow chimeras. B). Phenotypic characterization of endogenous OT1 TILs in B16.OVA tumor-bearing chimeric mice based on CD44, PD-1, TOX and GzmC expression. C) Analysis of TOX, TCF1 and GzmC expression in CD44⁺ PD-1⁺ endogenous OT1 harvested from OT1 chimeric mice treated or not with PD1d/2v/33-ACT. D) Antitumor efficacy of PD1d/2v/33-ACT in chimeric vs. wt-mice.

Whether the endogenous tumor specific TCF1⁺ cells are the ones directly reprogrammed by PD1d/2V/33-OT1 and give rise to the C5-like endogenous TILs is a very interesting question, that will be properly addressed in a follow-on manuscript.

It may be also important to analyze the baseline TCF1 phenotype by each PD1d/2v and PD1d/33 OT-1 cells.

Response: All OT1 cells, independently of the transduced payload, expressed TCF1 and most had a central memory-like (TCF1⁺ CD62L⁺) phenotype at the end of expansion and prior to ACT. This is shown in **Extended Data Figure 1d**.

2. Although the authors used engineered CD8 cells to test the efficacy of PD1d, IL-2v and IL-33, I wonder whether treatment of tumor bearing mice with a-PDL1, IL-2 mutant (binding only to IL2R $\beta\gamma$) and IL-33 injection along with OT-I transfer would induce a similar synthetic effector state by OT-I cells within tumor.

Response: We thank the Reviewer for raising this important point. Indeed, orthogonal ACT is superior to systemic therapy with recombinant IL-2^V and IL-33 and this was added to the revised manuscript. Since we have already shown that anti-PD-L1 was not required for the therapeutic effect, we did not include it in these experiments. Below is a detailed account of this data.

We asked whether systemic treatment of tumor bearing mice with recombinant IL-2^V and IL-33 alone or in combination with untransduced (UT) OT1 would induce similar synthetic states in TILs and similar therapeutic efficacy. To address this, we treated a first group of mice with recombinant mouse (rm)IL-33, using the lowest dose with reported antitumor efficacy (20) (0.5 μ g/mouse once a day) and recombinant (r)IL2^V (20 μ g/mouse twice a day) (21), starting on day 0 (**New Extended Data Fig.5a**). A second group of mice received the same systemic cytokine treatment in combination with UT OT1, and a third group was treated with orthogonal engineered PD1d/2v and PD1d/33 OT1 cells. The treatment had to be stopped on day 8 due to severe adverse side effects in the groups receiving recombinant cytokines. Indeed, signs of toxicity (ruffled fur, reduced animal mobility) were apparent after each morning's injection with the cytokine cocktail. In sharp contrast, no systemic toxicity was observed when IL-33 and IL-2^V were delivered by transferred OT1.

Relative to PBS-treated control mice, systemic treatment with rmIL-33/rIL-2V, did significantly delay tumor growth (**New Extended Data Fig. 5a**). Adoptively transferred UT OT1 did not add any therapeutic benefit relative to the cytokines alone. However, neither group performed as well as did the cohort treated with IL-2^V/33-secreting OT1 T cells. The difference in therapeutic efficacy of rIL-2^V+rmIL-33 was already evident 8 days post ACT, when systemic treatment was discontinued, proving the therapeutic superiority of orthogonal ACT over systemic cytokine treatment. FACS analysis of TILs on day 9 post ACT showed that endogenous CD44⁺PD-1⁺ CD8⁺ TILs recovered from mice treated with rmIL-33/rIL2^V upregulated GzmC in both the TCF1⁺ and the TCF1^{neg} compartments (**New Extended Data Fig. 5b**), thus suggesting that the PD-1⁺GzmC⁺ phenotype on endogenous CD8 TILs can be successfully induced in tumor-specific TILs by the two-cytokine combination *in vivo*.

Thus, even though systemic administration of IL-33/IL-2^V was significantly associated with antitumor activity, and induced similar cell states, IL-33/2^V-secreting OT-1 T cells are needed for safe and potent tumor control. Finally, the fact that addition of UT OT1 to systemic cytokines did not mimic the antitumor effect of orthogonal engineered was surprising. FACS analysis, revealed that persistence of total UT OT1 was markedly shorter (**New Extended Data Fig. 5c**), while the fraction of GzmC⁺ cells was markedly smaller than what we observed following orthogonal engineering (**Extended Data Fig. 5d**). Together, these results suggest that orthogonal engineered T cell therapy is superior to the therapy with the separated components, although the combination of IL-33 and IL-2^V can mediate a C5-like state in endogenous TILs.

3. To analyze the anti-tumor cytolytic activity and effector function of OT-I cells with TSE state, the authors compared OT-I cells isolated from PD1d/2v/33-ACT mice and OT-I cells induced *in vitro* for Tex state. This may not be a fair comparison, because multiple variations exist between these two samples in addition to the defined differentiation phenotype. It may be better to compare effector function of OT-I cells isolated from UT, PD1d/2v or PD1d/33-ACT groups.

Response: Unfortunately, it was not technically feasible to use OT1 TILs from the UT or PD1d/33 groups due to the low intratumoral persistence of these cells (**Extended Data Fig. 2b**). Furthermore, we could not use OT1 TILs from PD1d/IL2^V because they were mostly TCF1⁺ precursor-like cells (**New Fig. 7a-e**), and a comparison with T_{SE} cells would not have been fair either.

We submit that this experiment was really designed to demonstrate that T_{SE} cells are indeed potent effector cells, retain cytolytic activity against tumors, and are thus responsible for mediating the tumor-controlling effects of adoptive T cell therapy. We believe our experiment proved this key point. Although the control arm of *in vitro* exhaustion is not entirely comparable, it is a model that is increasingly used to mimic exhaustion (at a transcriptional, epigenetic and metabolic level) and has been used recently by several groups to model Tex cells (22-24).

4. The authors show highly increased expression of GZMC by PD1+TCF1+TOX1low cells generated by PD1d/2V/33-ACT compared to PD1d/2v and PD1d/33-ACT. Since GZMC is the feature of CD8 cells with TSE state after PD1d/2v/33-ACT and the majority of CD8 cells with TSE state transits to TCF1⁻ cells, it is important to also show GZMC expression by TCF1⁻ cells from their different conditions. The point here is whether the unique feature of CD8 cells with TSE state is mainly contributed by TCF1⁺ cells or both TCF1⁺ and TCF1⁻ subsets.

Response: In **Fig. 4c** we show the expression of Gzmc by PD-1⁺TCF1^{neg} OT1 TILs from all experimental conditions. We found that Gzmc⁺ cells were highly enriched uniquely in TILs harvested from tumors treated with orthogonal ACT, whereas their frequency is low in other conditions. In addition, in **Fig. 6** we demonstrate that orthogonal ACT also induces a TCF1⁺PD-1⁺ Gzmc⁺ synthetic precursor-like state, which would be the precursor of the final TCF1^{neg} T_{SE} cell state based on the inference trajectory analysis. Thus, we speculate that both T_{SP} and T_{SE} CD8⁺ TILs represent the precursor and effector states, respectively, of a novel PD-1⁺TOX-indifferent CD8⁺ Gzmc⁺ synthetic T-cell differentiation program, therefore both compartments might contribute to the unique features of these non-canonical cells.

5. To test the contribution of endogenous CD8 cells to tumor control after PD1d/2V/33-ACT, the authors compared the ACT efficacy in WT and p14 Tg mice. P14 Tg CD8 cells do not express TCR that can recognize TAA. However, this system also lacks other immune components including CD4 cells that can exert helper function. Therefore, this experiment would not provide the evidence that endogenous CD8 contribution is important for the efficacy of PD1d/2v/33-ACT.

Response: We thank the Reviewer for raising this important point. In the manuscript we showed that antibody-mediated depletion of CD4⁺ T cells concomitant to PD1d/2^V/33-ACT did not compromise, but instead significantly improved tumor control and mouse survival, indicating that endogenous CD4⁺ cells did not contribute to tumor control (New **Extended Data Fig. 8f**). Therefore, the absence of CD4⁺ T cells in P14 Tg mice is not expected to have biased our results in a positive way.

Since we decided to focus the revised manuscript in describing the new biology within the exogenous OT1 TIL compartment and leave the characterization of the reprogramming within the endogenous CD8 TIL compartment for a follow-on manuscript, we eliminated this experiment in the revised manuscript. We hope the Reviewer will approve this approach.

6. The authors show that IL-2v is mainly involved in downregulation of TOX by Tex cells and IL-33 (only together with IL-2v) contributes to the GZMC expression by CD8 cells. IL-2v may have a direct effect on activated CD8 cells that express IL-2R $\beta\gamma$, however, it is not clear how IL-33 would work in this system. Its receptor ST2 is expressed by multiple cell types. Therefore, it is important to show which cell type expresses ST2 and also discuss the potential mechanisms of how combinations of IL-2v and IL33 can have this synergic effect to improve anti-tumor function by CD8 cells.

Response: We thank the Reviewer for raising this important point. In fact, we verified that OT1 TILs did not express the IL-33 receptor ST2 (New **Extended Data Fig. 6b**), suggesting that the contribution of secreted IL-33 to the generation of the C5-T_{SE} cell state is CD8 T-cell extrinsic, most likely due to its reported ability to reprogram the TME (20, 25-27).

Indeed, as the Reviewer alludes, ST2 is expressed by CD4⁺ Th2 cells, DCs, mast cells, type 2 innate lymphoid cells (ILC2s), eosinophils, basophils, natural killer (NK) cells, and NK T cells (28). Most recently, activated CD4⁺ Th1 effectors and cytotoxic CD8⁺ T cells as well as CD4⁺ Treg cells were shown to express ST2 (29-32). Recently, IL-33 has been described as a potent stimulator of anti-tumor immune responses (20, 27, 33, 34). Both systemic administration and tumoral expression of IL-33 is sufficient to significantly delay tumor growth in a CD8⁺ T cell and NK cell-dependent manner, through stimulation of the NF- κ B signaling pathway (20, 27, 33). The anti-tumor effect is also dependent on the ability of IL-33 to activate tumor-associated DCs and restore their cross-priming potential. Accordingly, IL-33 has been successfully used as an adjuvant to enhance antigen-specific tumor immunity (20, 25-27, 33, 34), but never in the context of gene-engineered tumor specific T cells. Finally, a recent report indicates that IL-33 can in fact provide direct costimulatory signals to CD4⁺ T cells and drives Th1 polarization, activation, and GVHD exacerbation in the context of allo-transplantation (35).

Based on this evidence, we speculate that IL-33 drives intratumoral inflammation, specifically it activates myeloid cells, triggering mechanisms which are critical to ensure proper T-cell activation. The most likely mechanism is perhaps by unleashing the cross-priming potential of tumor-associated DCs and inducing high level of intratumoral inflammation (20, 25-27, 33, 34). We discussed the above in the Discussion section of the manuscript.

Interestingly, given that both IL-33 and recombinant IL-2^V mobilize NK cells (21), we asked whether triple-engineered ACT activated NK-mediated antitumor immunity. NK-cell depletion did not impact tumor control, although tumor-infiltrating NK cells did increase post-ACT containing IL-2^V, especially with PD1d/2^V/33-ACT, but failed to display an activation phenotype (**Fig.R4**). We elected not to include this data in the current manuscript due to space constraints.

Fig. R4 Intratumoral accumulation of NK Cells induced by orthogonal engineering is not linked to tumor control. (A) Tumor growth over time curve of PD1d/2^V/33+ OT1-treated mice in the absence or presence of a depleting antibody specific for NK1.1 administered i.p. 250 µg/mouse beginning 1 d before initiation of PD1d/2^V/33-ACT and maintained until day 27 post tumor inoculation. (B) Left: Tumor weight normalized numbers of NK cells on days 5 and 12 post ACT. Right: surface expression of NK-related activation markers: CD25 and KLRG1 on days 5 and 12 post ACT.

Reviewer #3

(Remarks to the Author)

In this manuscript, Corria-Osorio et al. 'orthogonally' engineer adoptively transferred CD8+ T cells to coexpress three payloads: a previously described IL-2 variant (hypothesised to promote CD8+ T cell stemness through selective lower-affinity interaction with IL2Rb/g), IL-33 (a proinflammatory alarmin (Villarreal et al. 2014) and a secreted PD-1 decoy, in an attempt to augment anti-tumor immunity and programme 'exhaustion-resistance'. Justification for this selection of genetic payloads was relatively superficial; however using the OT1 adoptive transfer model treating Ova-expressing syngeneic B16 tumours, the co-administration of mixed transfers of orthogonally engineered T-cells expressing A) the PD1-decoy with the IL-2 variant, and B) the PD1-decoy with IL-33, resulted in substantially greater tumour control compared to non-transduced control cells, or cells expressing single or pairwise combinations of these genetic payloads. This was proposed to be a consequence of in-situ reprogramming of exogenous and endogenous CD8+ T cells away from traditional T cell exhaustion and towards a proposed TOX- GMZC+ 'synthetic' state, characterised by exhaustion resistance, greater cytotoxic function dependant on the secretion of both IL-33 and IL-2v from adoptively transferred cells. The limitation on any given transferred cell population expressing only 2 of the 3 payloads appears technical, since bi-cistronic retroviral vectors were used.

Convincing evidence was given for greater expansion of orthologously engineered CD8+ T-cell within the tumor site compared to untransduced controls. Evidence presented through pharmacological blockade of lymph node egress suggests that this response does not require continuous T cell recruitment from lymph nodes. This, in combination with increased effector function could be driving the tumor regression observed. Interestingly, enhanced function of the TOX-GMZC+ 'synthetic' state was observed despite expression of the co-inhibitory/exhaustion molecule PD-1, and the authors claim that PD-1 blockade on these cells did not further augment their function - this is used to support the claim that the cells are 'exhaustion resistant'. Furthermore, the TOX- PD-1+ phenotype of the cell state induced suggests an uncoupling of TOX dependency for expression of PD-1. This study provides an interesting translational approach that capitalises on pre-existing immunotherapeutic candidates to improve adoptive transfer. However, the paper is overly descriptive, mechanistic insights into how the IL-2 variant and IL-33 synergise are lacking, it is unclear whether PD-1d plays any functional role in the triple-therapy (so why is it included?) and the claim that the cells are 'exhaustion-resistant' are poorly substantiated.

Response: We thank the Reviewer for the evaluation of our work. We took this Reviewer's critique under very careful consideration as we designed the manuscript revisions. In response, we have conducted much deeper analyses to elucidate the molecular mechanisms underpinning the effects of IL2^v and IL-33, as well as their synergy. These include computational analyses using scRNAseq data to infer key molecular pathways that underlie the different states, which pointed to metabolic rewiring, and further transcription factor (TF)/regulon analyses that revealed the putative transcriptional networks characterizing these states.

These are now presented in (**New Fig. 7 and Extended Data Fig. 7**). Furthermore, we provide new experimental evidence that PD1d does not play any functional role in restraining the synthetic effector cells generated by the orthogonal cytokine engineering, and that the cells are insensitive to PD-L1 suppression. Below we address each point by the Reviewer, including answering how the IL-2^V and IL-33 synergize, provide justification for using PD-1d in our manuscript strategy, explain how PD1d inhibition plays no functional role in the triple therapy, and substantiate the claim that the synthetic cells are PD1- and exhaustion-resistant.

Major points

In Fig 1 it is suggested that all three of the PD1-decoy, IL-2 variant and IL-33 are required to drive the augmented adoptive immunotherapy response. However, it is unclear from the figure whether PD-1d contributes at all to the response.

Response: We thank the Reviewer for pointing out that our message was not clear enough. In fact, the key argument of the paper was not to show that all three engineering modules were required. The key conclusion of the manuscript is that only the two cytokine modules (IL-2^v and IL-33) are required to induce the novel, non-exhausted differentiation program with direct antitumor cytotoxic potential, and that PD1d is finally not required (**New Fig.5**), as the cells are not TOX⁺ canonical exhausted. We have clarified this key point in the revised manuscript.

In particular, in Fig 1b, the combination of IL-33 and IL-2^v in the absence of PD-1d is not shown - but essential in interpreting the data, and the importance of 'orthogonal' gene delivery.

Response: The Reviewer is right. This combination was not shown in the first group of experiments in figure 1, but it was included later in the paper (**New Fig.5**), where we used it to demonstrate that PD1d was functionally dispensable, and that IL-2^V/33 was similar to PD1d/2^V/33 ACT. This demonstrated that the orthogonal property of our therapy is stemming from the two cytokines, IL-2^V (which drives stemness) and IL-33 (which drives effector differentiation), and not by PD1d.

Fig 1b also lacks sufficient mice in some groups (IL-33 and IL2) to reliably demonstrate ineffectiveness of control (single- or double transduced) groups - if these are insufficiently powered, they do not serve as adequate negative controls for the superior functionality of the triple orthogonal therapy.

Response: We thank the Reviewer for helping us clarify these findings. In general, the type II error of not recognizing a difference when it exists suffers from small power and small sample size, but it is of no interest when a statistically significant difference is found. When concluding that a statistically significant difference exists, the only error of concern is the type I error. Indeed, what these experiments show is that the triple combination was better than the double, as well as better than the single-transduced groups, even in the presence of such small sample size as pointed out by the Reviewer especially in the IL-33 and IL-2^V groups (Fig 1b). If we were to perform the exercise of comparing the ORR between the smallest of the groups (IL-2^V; n=4) and the triple-transduced group (n=14), the p-value for the corresponding Fisher's exact test is p=0.0049. Similarly, for IL-33 (n=5), the p-value is 0.0018. Performing such repeated tests would of course

increase the type I error, but taking multiple testing into account by using the method of Hochberg (36), all 7 of the 7 possible comparisons to the triple-transduced group would still be significant, due to the dramatic difference in ORR of the triple- group compared to all others. Thus, while controlling the Familywise Error Rate at level $\alpha=0.05$, we find a significant difference in overall response rate (ORR) for all groups when compared to the triple-transduced group ORR. Similarly, according to the multivariate exact logistic regression all comparisons versus the triple combination are statistically significant, as shown in (**Extended Data Table. 2**). Nevertheless, if the Reviewer insists for the revised paper we might do an additional experiment including the single groups to have more animals in these. However, we respectfully submit that we believe that this is not necessary.

In Fig 4h, the authors show that PD-1d is not actually required in its decoy mutant form for efficacy, which somewhat undermines the argument of the paper, that indeed all three 'orthogonal' genetic payloads are required for efficacy, as suggested in Fig 1b. If it is not required, why is the effect of all three genetic alterations studied in the paper, rather than the two functional payloads?

Response: As stated above, the key conclusion of the manuscript is that only the two cytokine modules (IL-2^V and IL-33) are required to induce the novel, synthetic state, and that PD1d is finally not required (**New Fig.5**), as the cells are not TOX⁺ exhausted.

We included the PD1d in our experiments initially as we thought it to be an essential part of the engineering strategy. In fact, the analyses we performed demonstrated that PD1d does not play any functional role, and that it is not required to achieve the orthogonal effect.

The reasons we kept the PD1d approach are twofold: first, we thought that keeping it served well the logical sequence of the narrative in the manuscript, since we gradually unveiled the synthetic states as independent of TOX⁺ canonical exhaustion, and then demonstrated that PD1 blockade is dispensable. Second, we used Pd1d as a beacon for a technical reason: presently there is no antibody available that discriminates between wild-type IL-2 and IL2v allowing to assess expression of IL2v by transduced T cells (which also make IL-2 during CD3/CD28 activation). Since Pd1d was a common module in the two vectors (Pd1d_IL2^V and PD1d_IL-33), we used it to evaluate the transduction efficiency across conditions and to ensure that the same dose of transduced (PD1d⁺) cells was infused in all experiments. In addition, we used it to ensure a 1:1 ratio of the orthogonally engineered cells. In our final experiments, we also tested cells transduced with the bicistronic vector IL2^V_IL-33 carrying the two cytokines without PD1d, where we demonstrated equivalence of the orthogonal engineering (Pd1d_IL2^V plus PD1d_IL-33) with IL2^V_IL-33. In these experiments we used IL-33-transduced cell as the beacon to titrate the cell dose.

Fig 4g is used as evidence that the triple-engineered cells are 'exhaustion resistant' despite expressing PD-1, since anti-PD-L1 treatment is not seen to improve the immunotherapy response. However, a critical positive control for anti-PD-L1 functionality in Fig 4g is not present - does anti-PD-L1 blockade synergise with the transfer of untransduced OT-1 cells? This would be an important control to support the hypothesis.

Response: The Reviewer is right. We used this data (now **Fig.6**) only to support the claim that although OT1 cells expressed PD-1 and TIM-3, pharmacologic blockade of PD-1 alone or PD-1 and TIM3 had no

effect on the therapy. This was further substantiated by the data in Fig. **Fig.6b, c**, where the removal of Pd1d from the engineering module was inconsequential.

We respectfully submit that the experiment requested by the Reviewer, testing the effect of PD-L1 blockade with untransduced OT1 cells, was presented in **Extended data Fig.1e** (cohort #2), where we show that in non-lymphodepleted mice adding anti-PD-L1 antibody to ACT with untransduced (UT) OT1 cells led to transient better tumor control than UT OT1 alone between days 9-15 post ACT (left) and some gain in survival (right), although in both cases did not reach significance ($p=0.43$). Importantly, in this experiment, the hosts had not been previously lymphodepleted, and the OT1 cells did not persist post transfer.

Furthermore, the ability of PD1d alone (PD-L1 blockade) to slightly impact ACT (in the absence of IL-2^V or IL-33) was demonstrated *in vitro* and *in vivo* in lymphodepleted hosts (*no impact on OS*) in (**Extended data Fig. 1a**). Please note that this last experiment was conducted with host lymphodepletion, since we did not use IL-2^V, which enables cell-autonomous cell expansion *in vivo*.

The claim that PD-1, despite being expressed, is not transducing an inhibitory signal is potentially interesting, but almost completely mechanistically unelucidated. Comparing CD8+ T cells in conventional exhausted and synthetic 'exhaustion-resistant' states *in vitro*, can the authors show that the provision of PD-L1 *in vitro* to exhaustion resistant cells does not result in inhibition of signaling and effector functions, compared with control exhausted cells?

Response: We thank the Reviewer for raising this point. We have now added new data to formally demonstrate that PD-1 expression in synthetic CD8 T cells does not translate to functional restrictions.

To directly test this, following the Reviewer's recommendation, we set up an *in vitro* assay of T cell inhibition by plate-bound PD-L1, where T cells are stimulated through the TCR while seeded on recombinant PD-L1, as previously shown by Ogando et al (37). We tested the ability of PD-L1 to suppress the activation of CD45.1⁺ OT1 TILs harvested from responding tumors following 2^V/33-ACT vs. control OT1 T cells harvested from spleens of treatment-naïve mice (**New Fig. 5f**). PD-L1 significantly abrogated the activation (proliferation and upregulation of activation and cytotoxic markers) induced upon TCR-mediated stimulation in naïve OT1 T cells (**New Fig. 5g-i**). However, PD-L1 did not affect significantly the TCR-mediated restimulation of OT1 TILs from 2^V/33-ACT. These results were consistent with the fact that both T_{SP} and T_{SE} cell states were not enriched in the gene signature corresponding to PD-1 mediated inhibition of CD8 T cell activation (37) (**New Fig. 5e and New Fig. 6f**).

Please note, we performed the same *in vitro* assay described above with TOX⁺ canonical exhausted OT1 TILs harvested from B16.OVA tumor-bearing OT1 chimeric mice (see Fig. R3A and **Fig. R5A**). However, cell recovery 48 hours post stimulation was very poor (**Fig. R5B**), with less than 100 cells/well available for analysis, which was two orders of magnitude lower than IL-2^V/33 gene-engineered OT1 TILs, confirming the hypofunctional status of TOX⁺ canonical exhausted TILs. However, in the few recovered cells we could see a clear trend that PD-L1 indeed inhibited the TCR-mediated activation of canonical exhausted TILs (**Fig. R5C**). We did not include this data in the revised manuscript because of the low resolution due to the poor cell recovery.

Fig. R5: A) Schematics summarizing the experimental design. Briefly, 30 OT1 chimeric mice were injected with B16-OVA tumor cell and 12 post inoculation (tumor volume ~ 200 mm³), the tumors were harvested and pooled for CD45⁺ magnetic isolation. B) Absolute number of OT1 T cells recovered post anti-CD3 stimulation in the presence or absence of plate-bound PD-L1. Data summarize two independent experiments (n=2-3 internal replicate/condition) C) Analysis of GzmB and CD25 expression in post stimulated OT1 TILs from the chimeric mice.

Further corroborating that the novel synthetic cell states were rather unaffected by PD-1 blockade, we found that PD1d was not required for tumor control (**New Fig. 5b**) or for generating the polyfunctional GzmC⁺TOX^{neg} C5-like effector state in OT1 TILs (**New Fig. 5c**), and OT1 TILs from 2^V/33-ACT exhibited similar polyfunctional effector phenotype to TILs from PD1d/2^V/33-ACT (**New Fig. 5d**)

Thus, we conclude that although PD-1⁺, synthetic CD8 T cells are not functionally restrained by PD-1. Therefore, PD-1-expression in the new TOX^{neg}GzmC⁺ TIL program is rather a surrogate marker of T-cell activation than a mediator of functional exhaustion.

The authors should also provide a mechanistic basis (and indeed better experimental evidence) for the proposed exhaustion resistance, since it is hard to understand how PD-L1 would be rendered non-functional as a proximal inhibitory signaling molecule despite its expression on these cells.

Response: The breakthrough finding of our work is that orthogonal engineering of T cells with IL-2^V and IL-33 positions the cells in a state that can handle chronic antigen stimulation without upregulating TOX or entering exhaustion. This is the first time such state is made possible, freeing effector T cells from the natural constraints of exhaustion.

So, as the Reviewer asks, why are such chronically stimulated TILs not exhausted?

We conducted several analyses to decipher the state of exhaustion resistance in the synthetic effector (T_{SE}) and synthetic precursor (T_{SP}) cells. First, we attribute the non-exhausted state of these chronically stimulated cells to the full suppression of TOX (Fig. 3c and Fig. 4a), the master transcriptional and epigenetic regulator of exhaustion(13-15). In addition, several TFs (i.e. *Nr4a1*, *Nr4a2*, *Nfatc1*, *Batf*, *Irf1*) that coordinately work with TOX to reinforce the exhausted phenotype (15, 16, 38, 39) are also downregulated or barely expressed in C5- T_{SE} vs. canonical T_{EX} (Fig. 3c and Fig. R6A). Furthermore, in the manuscript we showed that the whole NFAT/Nr4a/TOX-driven canonical exhaustion program is absent in C5- T_{SE} TILs (Fig. 2i and Fig.3i) – and in sharp contrast with canonical TOX+ exhausted T cells, which are dysfunctional (13-15, 40) and (Fig. R6B, top) – C5-TILs, are polyfunctional effector cells, with direct antitumor potential (Fig. 4 and Fig. R6B bottom). Indeed, we hypothesize that the latter is a direct consequence of TOX suppression, since it directly controls the genome accessibility of many effector genes (15). Finally, the experimental evidence presented above that orthogonally engineered OT1 cells are directly resistant to suppression by PD-L1, complements the first mechanistic view of how exhaustion resistance is mediated by the orthogonal cytokines.

Fig. R6: A) Radar plots showing differentially expressed key exhaustion-related TFs between T_{SE} and T_{EX} TILs. B) Analysis of the polyfunctionality between PD-1+ TOX+ canonical OT1 TILs (isolated from OT1 chimeric mice) and TSE-like OT1 TILs harvested upon PD1d/2v/33-ACT. In both cases, isolated OT1 TILs were stimulated with 10 ng/ml of the cognate OVA peptide for 4 hours in the presence of brefeldin A.

Careful analysis of the states reached under different experimental conditions revealed that IL-2v was primarily responsible for promoting downregulation of TOX in both exogenous OT1 T cells (Fig 3 and New Fig. 7) as well in endogenous PD-1+ Tex and Pex CD8+ TILs (Fig. R7).

Fig. R7: Analysis of TOX expression in endogenous Tex TILs recovered at day 12 post ACT (A) or Pex (B) from mice treated with PD1d/2^V/33, PD1d/2^V and PD1d/33. Data collected from 5 independent experiments n= 5-6 mice/group. One-way ANOVA test in combination with a Dunnet Test to correct for multiple comparisons was used * p<0.05, ** p<0.01, *** p<0.001, ****p<0.0001. Naïve CD8 T cells isolated from the spleen of non-tumor bearing mice were (gray) used as negative control for the expression of the studied markers.

We next reasoned that sustained exposure of TILs to IL-2v, due to the continued secretion by the gene engineered cells, could be the reason of TOX suppression and exhaustion resistance. To model this, we used a recently published protocol of sustained antigen stimulation of naïve CD8 T cells, which recapitulates exhaustion *in vitro* (22). Isolated naïve OT1 from spleens were used either directly for the sustained antigen stimulation assay (positive control of TOX+ canonical exhaustion (22)) or pre-exposed for 3, 7 or 12 days to IL-2v upon activation with anti-CD3/anti-Cd39 *in vitro* (Fig. R8A), which induces a central memory-like (T_{CM}) CD62L⁺CD44⁺ TCF1⁺ phenotype since day 3. Following 5 days of repeat peptide stimulation in the absence of cytokines, we interrogated cells on day 6 for expression of PD-1, TOX. In addition, we restimulated them with OVA peptide for 4 hours and measured expression of IFN_γ and TNF_α as surrogate markers of antitumor effector functions.

Fig. R8: A) Experimental Design. *In vitro* expansion protocol. B) Analysis of PD-1, TOX, IFN_γ and TNF_α expression after 5 days of peptide stimulation. For IFN_γ and TNF_α expression the cells were 4-hours re-stimulated with the peptide in the presence of Brefeldin A.

As shown in (Fig. R8B), OT1 T cells pre-exposed *in vitro* for a week or more to IL-2^V and then subjected to repeat TCR stimulation, upregulated PD-1 but not TOX. Conversely, most naïve OT1 cells that had not

been exposed to IL-2v differentiated to PD-1⁺TOX⁺ canonical exhausted cells. Interestingly, TOX expression in OT1 cells at the end of repeat antigen stimulation was inversely proportional to the length of exposure to IL-2v, and gradually decreased from no exposure (naïve) to 12-day exposure. In addition, it was inversely proportional to their polyfunctionality (IFN γ /TNF α co-expression). Thus, from this we conclude that activated CD8 T cells can be driven towards a PD-1⁺TOX^{neg} polyfunctional state upon sustained antigen stimulation (the only driver of TOX expression(14, 15)), by virtue of long-term (>7 days) exposure to IL-2v.

We next used a similar experimental set up, but this time to test whether previously *in vitro* chronically stimulated, PD-1⁺ TOX⁺ dysfunctional CD8 T cells (**Fig. R9A**) could be functionally rescued by exposure to IL-2v. Indeed, as shown in (**Fig. R9B**), a seven-day treatment with IL-2v alone (10 IU/mL) downregulated the expression of both PD-1 and TOX and functionally rescued the cells. Notably, only PD-1, but not TOX, was upregulated again upon a second round of TCR-mediated chronic stimulation (**Fig. R9C**), and these cells conserved a higher level of IFN γ secretion, and a 9-fold more production of TNF α relative to the end of the 1st round of chronic stimulation (**Fig. R9A**), two effector molecules that are epigenetically controlled by TOX (14, 15).

Fig. R9: Analysis of PD-1, TOX, IFN γ and TNF α expression upon a 1st round of 5 days sustained of Ag stimulation (A), IL-2v-mediated homeostatic resting (B) and a 2nd round (5 days) of chronic TCR stimulation (C). For IFN γ and TNF α expression the cells were 4-hours re-stimulated with the peptide in the presence of Brefeldin A. Naïve OT1 T cells were used as negative control for the FACS staining.

How IL-2 signaling directed through the IL-2R $\beta\gamma$ receptor chains achieve suppression of TOX will require additional investigation. However, it appears to be part of a transcriptional and epigenetic reprogramming that provides the base for producing novel cell states which profoundly deviate from canonical exhaustion upon antigen engagement. Notably, IL-2R driving TOX downregulation in exhausted CD8⁺ T cells was just recently reported (3, 17), yet in the revised manuscript we show that both T_{SE} and T_{SP} cell states are still

unique and expressed the lowest level of TOX amongst these novel cell states (**New Fig. 3j and New Fig. 6k**).

The fact that IL-2v is associated with high level of intratumoral persistence of TOX^{neg} CD8⁺ T cells in the context of sustained antigen stimulation (**Fig. 1d, Extended Data Fig. 2b and New Fig. 7**) is also a novel and remarkable finding, since TOX expression is required for persistence under similar conditions (13-16). Interestingly, a new regulon analysis (7), incorporated in the revised manuscript (New Fig. 7 e-h), suggests that IL-2v regulates this process by activating a transcription factor (TF) network motif comprising *Fos11*, *Egr1*, *Myc*, *Irf8* and *Batf3*, which might be involved in the metabolic support of sustained gene expression and protein translation (**New Fig. 7 g-h**), both highly energetically costly cellular processes (9, 10), which are dampened in canonical exhausted CD8 T cells (10). Indeed, the AP-1 transcription factor *Fos11* has been associated with long-term persistence of CD8 T cells in the context of chronic viral infections (11), while *Batf3* has been recently shown to be a critical TF for CD8 T-cell memory formation and survival via negative regulation of the proapoptotic factor BIM (12). Indeed, T cells that lacked *Batf3* succumbed to an aggravated contraction and had a diminished memory response (12). Thus, IL-2v alone contributes to reprogram gene-engineered tumor specific TILs to escape TOX⁺ canonical exhaustion, endows them with superior metabolic transcriptional programs, and positions them in an activated, memory-like state post-ACT, which can persist in the absence of TOX (**New Fig. 7**). We also believe that our findings of key anabolic and mitochondrial pathways being specifically upregulated in the T_{SE} and T_{SP} states (**New Fig. 4f-g, New Fig. 6d-e and New Fig. 7**) are highly relevant in explaining how exhaustion resistance is bestowed, since TOX⁺ exhaustion has been increasingly recognized as a metabolic and mitochondrial failure (41).

Importantly, in the context of our extensive *in vitro* characterization mentioned above, we confirmed using a well-controlled *in vitro* experimental system and with a new scRNAseq dataset (please see **Fig. R1A** in the response to Reviewer 1) that IL-2v at 50 IU/mL induces activation of 24 out of the 28 regulons predicted in (**New Fig. 7**) from our *in vivo* OT1 data (see **Fig. R1B**).

Finally, we would like to share with the reviewer new data that indicate how the above-mentioned IL-2v-induced transcriptional program may contribute to the better cytotoxicity of T_{SE} TILs.

We have recently fully inferred the transcriptional TF-gene regulatory network of T_{SE} and T_{SP} cell states relative to canonical TOX⁺ Pex and Tex by calculating the regulon activity of more than 800 TFs with SCENIC (7), a computational method for gene-regulatory network (GRN) reconstruction (**Fig. R10A**). Notably, after unbiased clustering analysis, we could clearly identify 4 clusters of regulons (i.e. modulons) that were differentially active across canonical and synthetic T cell states (**Fig. R10B**). These cell state-specific modulons also constitute highly connected network motifs in the SCENIC-inferred gene regulatory network (**Fig. R10C**). In addition, by using orthogonal partial least squares discriminant analysis (OPLS-DA) (8), a computational approach to discriminate between two or more groups (classes) using multivariate data, we could identify well-known exhaustion related TFs such *Eomes*, *Maf*, *Irf1*, *Prdm1*(40) within the set of TFs with the highest statistical power to discriminate between T_{EX}-TILs from the other states (**Fig. R10D**) (Please note that the regulon of TOX cannot be calculated using SCENIC, because this TF does not recognize a linear DNA motif, however TOX can be analyzed as a target of other TFs of the network).

Interestingly, we could not identify canonical exhaustion-related TFs within the most discriminant modulon for T_{SE} (**Fig. R10E**), confirming that this cell state has a distinct and unique transcriptional wiring. Notably, 6 of these TFs (*Batf3*, *Irf8*, *Jund*, *Atf3*, *Atf4*, and *Xbp1*) were also identified as part the transcriptional circuit induced by IL-2v, and are interpreted to be involved in the metabolic support to sustained gene expression and protein translation (**New Fig.7**). Since these metabolic/mitochondrial processes are required for sustained killing by cytotoxic CD8 T cells (42), we asked whether they were also involved in the direct regulation of the many effector molecules that compose the specific cytolytic machinery of T_{SE} TILs. Remarkably, most of the upstream regulators of the cytotoxic/effector molecules in the SCENIC-inferred TF-gene regulatory network belong, almost exclusively, to the T_{SE}-specific modulon (**Fig. R10F**), and the above-mentioned IL-2v-related TFs are forming part of this transcriptional regulatory circuit (**Fig. R10F**). Thus, IL-2v could support both the better metabolic/mitochondrial fitness and cytotoxic potential of T_{SE} TILs by regulating a common transcriptional core, that is not highly active in canonical Tex TILs. (**Fig. R10E**). Indeed, we hypothesize that this set of TFs are the key mediators of IL-2v-induced exhaustion resistance.

Fig. R10: A) Schematics summarizing the Regulon/ Modulon Analysis using SCENIC. B) Unsupervised clustering analysis based on the regulon activity across canonical and synthetic T-cell states of 848 TFs. C) SCENIC-predicted gene regulatory network, the size of each TF (ball) reflects the discriminant power calculated using OPLS-DA. D) Top_13 most discriminant TFs for Tex. In red we highlight those that are well-known key TFs E) Top_13 most discriminant TFs for TSE. In red we

highlight those directly regulated by IL-2v F) Distribution of incoming regulatory links (edges in the network) of a set of relevant cytotoxic and effector target molecules.

Collectively, based on these findings we can only speculate that lack of TOX expression induced by IL-2v leads to a major epigenetic reprogramming in chronically stimulated synthetic T cells, since TOX is responsible for the chromatin remodeling necessary for the commitment to canonical exhaustion (14, 15), thus allowing chromatin accessibility in regions that would be otherwise inaccessible during chronic TCR stimulation. This could explain the transcriptional activities described above.

Further dissecting the above regulatory pathways will require in depth biochemical and molecular interrogation, along with validation of hypotheses using knockout methods. As such, we respectfully request that this work be part of a follow up manuscript. However, we discuss the above hypotheses in the Discussion section of the revised manuscript.

The manuscript would benefit from much better mechanistic understanding of the specific contributions of IL-33 and IL-2v to anti-tumor responses.

Response: This question is indeed important. In the revised manuscript we first draw from our *in vivo* observations to infer the specific contribution of each cytokine, and their synergistic effects. Importantly, our data show that IL-2v in the absence of IL-33 dramatically expands cells positioned in a novel synthetic memory-like $\text{TOX}^{\text{neg}}\text{TCF1}^{\text{int}}$ state (C7) in tumors (**New Fig.7**), however these cells are unlikely to transition to the synthetic precursor-like (T_{SP}) $\text{TOX}^{\text{neg}}\text{PD1}^{\text{+}}\text{GzmC}^{\text{+}}\text{TCF1}^{\text{+}}$ or effector-like (T_{SE}) $\text{TOX}^{\text{neg}}\text{PD1}^{\text{+}}\text{GzmC}^{\text{+}}\text{TCF1}^{\text{neg}}$ state in the absence of IL-33 coexpression.

We have conducted additional analyses of the scRNAseq data to infer the IL-2R-related transcriptional pathways involved in the determination of the different synthetic states achieved *in vivo* in the presence of IL-2v alone or IL-2v in combination with IL-33 (**New Fig. 7**). Our findings in the revised manuscript explain how IL-2^v alone drives a profound metabolic reprogramming of the cells sustaining memory and cell persistence. Moreover, we provide an analysis of the IL-2-related TF regulatory networks that underlie the differentiation in the different synthetic states, in an attempt to understand how cell persistence and expansion is achieved in the absence of TOX, how C7 (IL-2v) and C5 or C6 (IL-2v+IL-33) states are regulated, and specifically how alternate TFs with overlapping functions can secure the key metabolic programs ensuring cell stemness and expansion in these two alternate synthetic cell fates (**New Fig. 7**). These findings provide novel mechanistic evidence on the effects of IL-2v

As mentioned above (Response to Reviewer 1), in addition to the data presented here, we should point out that we have now further characterized extensively the effects of IL-2v in mouse T cells at a molecular level *in vitro*. Briefly, we have learned that IL-2v provides a very different level of IL-2R stimulation, and thereby drives unique signaling in CD8 cells, decoupling proliferation from differentiation, thereby allowing cells to activate, proliferate and simultaneously maintain stemness with powerful effector potential. We have dissected the signaling pathways and metabolic underpinnings of IL-2v versus IL-2 or IL-15, and have demonstrated that IL-2v unlike IL-2 maintains oxidative phosphorylation and drives mitochondrial biogenesis and fitness. This data substantiate our *in vivo* observations here and indicate that indeed IL-2v is responsible for driving stemness in CD8⁺ TILs. This has the specific consequences of enabling cell-

autonomous engraftment in the host and expansion in tumors in the absence of lymphodepletion, and specifically expansion of a large pool of TCF1⁺ precursor cells within tumors. Unfortunately, the work characterizing the *in vitro* effects of IL-2v is quite extensive and will need to be part of a follow-on manuscript.

To further understand the effect of IL-33, we verified that neither endogenous nor exogenous OT1 TILs expressed the IL-33 receptor ST2 (**Extended Data Fig. 6b**), suggesting that the contribution of secreted IL-33 to the generation of the C6-P_{SE} or C5-T_{SE} cell states is T-cell extrinsic, most likely due to IL-33's ability to reprogram the TME (20, 25-27). This data is consistent with a body of literature indicating that IL-33 exerts direct activation of the tumor myeloid compartment. Indeed, the anti-tumor effect of IL-33 is dependent on the activation of tumor-associated DCs and its ability to restore their cross-priming potential, thereby activating CD8⁺ TILs (20, 25-27, 33, 34). Accordingly, IL-33 has been successfully used as an adjuvant to enhance antigen-specific tumor immunity (20, 25-27, 33, 34).

Based on this evidence, we speculate that IL-33 drives intratumoral inflammation, specifically by activating myeloid cells, triggering mechanisms which are critical to ensure proper T-cell activation. The most likely mechanism is perhaps by unleashing the cross-priming potential of tumor-associated DCs and inducing high level of intratumoral inflammation (20, 25-27, 33, 34). Additional characterization of the effects of IL-33 on the tumor myeloid compartment, and deciphering of these effects on T-cell signaling, would be important, but we respectfully believe that it is beyond the scope of the present manuscript.

Finally, we should also point out that we did not manage to analyze here by scRNAseq OT1 TILs following ACT with cells transduced only with IL-33, since we could not recover many such cells for analysis in the absence of concomitant IL-2v (**Extended Data Figure 2b**). Indeed, we observed that IL-33 drives differentiation of the few residual OT1 towards TCF1^{neg} effector-like cells, which barely persist in non-lymphodepleted hosts and in the absence of IL-2^v

Endogenous TIL reprogramming seems to be largely IL-33 driven, what is the mechanistic basis for this?

Response: We thank the Reviewer for offering this question. In fact, we observed that both cytokines are critical for full reprogramming of endogenous TILs, similarly to transferred OT1 cells.

Given the space constraints of the revised manuscript, and for the sake of being able to fully develop the characterization of the synthetic states of transferred OT1 cells, we elected to remove the analyses of endogenous T cells from the manuscript. To fully answer this Reviewer, we provide the below explanation.

IL-33 alone – in the absence of IL-2v – led to functional activation of endogenous Tex-like cells (**Fig. R11A**), consistent with the expected properties of IL-33 in a tumor context (20, 25-27, 33, 34), but functional reinvigoration was much better when combined with IL-2v (**Fig. R11B**). However, it is important to note that functionally reinvigorated endogenous Tex CD8⁺ TILs in the context of IL-33 alone still expressed TOX (**Fig. R11C**), indicating that the functional improvement induced by IL-33 is still taking place in the context of canonical TOX⁺ exhaustion. For example, IL-33 driven TIL activation in the absence of IL-2v was associated

with short persistence of transferred IL-33-secreting OT1 (**Extended Data Fig.2b**). It is only when IL-2v is co-expressed that TILs (exogenous and endogenous) can reprogram away from TOX+ exhaustion.

Fig. R11: (A) Heatmap showing the normalized (row scaled) protein expression of TIM3, TNF α , GZMB, PRF1, IFN γ , GZMA and KI67 in GZMC⁺ versus GZMC^{neg} CD44⁺ PD-1⁺ TCF1^{neg} OT1 TILs recovered at day 12. Each column represents an individual mouse (n=34 mice in total). **(B)** Comparison of endogenous Tex CD8⁺ TILs harvested at day 12 post ACT from PD1d/2^v/33 or PD1d/33-treated mice in terms of expression of effector molecules. Data collected from 3 independent experiments n= 3-5 mice/group. A two-tailed Student's t test with Welch's correction was used for comparing expression of each marker, in each time point relative to baseline Tex TILs * p<0.05, ** p<0.01, *** p<0.001, ****p<0.0001. Naïve CD8 T cells isolated from the spleen of non-tumor bearing mice were used as negative control for the expression of the studied markers. **(C)** Analysis of TOX expression in endogenous Tex TILs recovered at day 12 post ACT from mice treated with PD1d/2^v/33, PD1d/2^v and PD1d/33. Data collected from 5 independent experiments n= 5-6 mice/group. One-way ANOVA test in combination with a Dunnet Test to correct for multiple comparisons was used * p<0.05, ** p<0.01, *** p<0.001, ****p<0.0001.

Since we verified that endogenous TILs did not express the IL-33 receptor ST2 in our system (**Extended Data Fig. 6b**), we hypothesize that the contribution of secreted IL-33 to the reprogramming of the endogenous CD8 T-cell compartment is indirect, through an intermediate cell, compatible with the well-established ability of IL-33 to reprogram the myeloid TME (20, 25-27). Indeed, a body of literature characterizing the antitumor effects of soluble IL-33, has demonstrated its role in activating CD8⁺ TILs by unleashing the cross-priming potential of tumor-associated DCs and inducing high level of intratumoral inflammation (20, 25-27, 33, 34). Consistent with intratumoral activation, tumor-resident DCs also upregulate costimulatory molecules which prove critical for the proper polyfunctionality of CD8 TILs (43). Furthermore, a recent direct costimulatory function of IL-33 on effector CD4⁺ T cells was described in the context of allotransplantation (35), although in our experiments neither endogenous CD8⁺ TILs nor OT1

cells expressed the IL-33 receptor, ST2. However, as discussed above, we elected to remove the analyses of endogenous T cells from the manuscript due to space constraints, in order to fully explain the states of transferred engineered cells.

Do the endogenous TIL directly respond to IL-33, do they express ST-2, or is there an intermediate cell type involved?

Response: We verified that endogenous TILs did not express the IL-33 receptor ST2 in our system (**Extended Data Fig. 6b**), suggesting that the contribution of secreted IL-33 to the reprogramming of the endogenous CD8 T-cell compartment is indirect, through an intermediate cell, compatible with the well-established ability of IL-33 to reprogram the myeloid TME (20, 25-27). However, as discussed above, we elected to remove the analyses of endogenous T cells from the manuscript, and we respectfully submit that defining the precise cellular and molecular pathways triggered by IL-33 within the myeloid compartment, and how these translate to T cell signaling imparting the T_{SE} state, albeit important, are questions beyond the scope of the present manuscript.

Do the transferred cells become ST-2 positive?

Response: No, we confirmed that exogenous OT1 TILs did not express the IL-33 receptor ST2 *in vivo* in our system (**Extended Data Fig. 6b**).

Does ablation of ST-2 on either transferred cell populations or host cells ablate the response?

Response: Since we verified that neither endogenous nor exogenous OT1 TILs express the IL-33 receptor ST2 in our system (**Extended Data Fig. 6b**), we respectfully believe that there is no need to explore ablation of ST2 in these cell populations.

What proportion of the effects are cell intrinsic vs cell extrinsic - ie can the phenotypes of cells expressing A) the PD1-decoy with the Il-2 variant, and B) the PD1-decoy with IL-33 in the mixed transfer be distinguished and compared by flow cytometry?

Response: We have taken advantage of the congenic tracking system that allowed us to specifically follow *in vivo* OT1 T cells gene-engineered with either PD1d/2^V (CD45.1⁺ CD45.2^{neg}) or PD1d/33 (CD45.1⁺ CD45.2⁺, see **Fig.R13A**). Using this experimental system, we found that both PD1d/33 and PD1d/2^V-transduced OT-1 T cells converge to and equally contribute to the C5 OT1 cell pool during tumor regression, as they acquire similar cell states *in vivo* during the response phase (see below **Fig.R13B**), which demonstrates that there is no important bias. We conclude that the synergistic effect of the two cytokines is exerted in a paracrine manner on both cell populations to produce the synthetic state C5. Specifically, IL-2v can presumably act both in an autocrine and paracrine fashion, in cells that produce it, and paracrine fashion in TILs that do not, while IL-33 acts through indirect paracrine mechanisms to activate TILs, presumably through bystander myeloid cells. Ultimately, all transferred cells converge to the same state.

Fig.R12: *In vivo* tracking of OT1 PD1d/2V and PD1d/33. A) Left: Experimental design. Briefly, congenic CD45.1⁺CD45.2^{neg} OT1 T cells were gene-engineered with to express PD1d/2V and CD45.1⁺CD45.2⁺ were gene-engineered with PD1d/33. At day 20 (day 8 post ACT) each TIL population was independently purified by FACS sorting and and single cell sequenced using 10X Genomics. B) Right top: UMAP plot showing a low-dimensional representation cell heterogeneity and unsupervised clustering results. Right low: Contribution of each experimental condition to the formation of each cluster.

Figure 3c is confusing, since the synthetic state of exogenous orthogonally engineered TIL is distinct from that of the endogenous TIL (the latter which overlapped with exhausted and effector memory cells) - yet figure 6 shows reprogramming of endogenous TIL towards the TOXlow Gzmchigh state following ACT with orthologous engineered IL-33 IL2v orthologous cells.

Response: We thank the Reviewer for helping us clarify this point. We show in **Fig. 3c** that the C5-T_{SE} cell state is only acquired by the transferred OT1 cells, while endogenous CD8⁺ T cells are also reprogrammed, transitioning to a state that resembles but is not quite C5. Thus, although reprogrammed away from canonical exhaustion and into a superior effector state, endogenous tumor-specific CD8⁺ TILs did not fully acquire the exact effector state of transferred cells.

Due to space considerations and for the sake of fully explaining the states of exogenous cells, we elected to focus the current revised manuscript on the new biology of the engineered transferred OT1 TIL compartment and leave the characterization of endogenous CD8⁺ TIL compartment for a follow-on manuscript. We hope that this Reviewer will agree with this approach.

For the sake of clarity, and to fully answer this Reviewer, we provided the section below, where it can be seen that that orthogonal ACT reprograms endogenous Tex TIL towards a two cell state that resembles C5, because it is significantly enriched in the C5-specific gene signature, notably lower TOX, but it is not

the C5 state proper. For instance, endogenous C5-like Tex TILs during tumor response express lower levels of GzmC than C5-OT1 TILs (**Fig. R13**).

Fig. R13: Representative dot plots showing expression analysis of Gzmc and TCF1 in CD44⁺PD-1⁺ OT1 TILs harvested from responding mice upon orthogonal ACT (left part) or in CD44⁺PD-1⁺ endogenous Tex TILs from the same condition in the presence (middle) or absence (right) of PD1d.

Orthogonal engineering ACT reprograms endogenous, tumor specific CD8⁺ TILs to new states away from PD-1⁺TOX⁺ canonical exhaustion

Interestingly, at baseline as well as following orthogonal ACT in the wild-type host, endogenous C4 Tex-like cells featured a high proportion of clonally expanded cells (**Fig. R12A**), indicating engagement against tumor. To learn whether these endogenous cells also deviated from canonical exhausted states under the influence of orthogonal ACT, we interrogated them longitudinally. Analysis of the exhausted compartment across many time points revealed four distinct states, one ascribable to TCF1⁺ Pex and the other three to Tex (**Fig. R12B, top part**). At baseline and upon tumor escape (from day 19), endogenous CD8⁺ TILs were mostly in Tex1, characterized by *Tox*, *Nr4a2*, *Tigit*, *Gzmk*, *Pdcd1*, *Ccl3* and *Ccl4* (**Fig. R12B, top part, bottom part**), consistent with the canonical *Tox*^{high}*Nr4a2*^{high} exhaustion program (14, 38). Following orthogonal ACT, endogenous Tex TILs departed transcriptionally from this state and acquired Tex2 (days 5 and 8) and later Tex3 (day 12). Neither subset upregulated *Cx3cr1* or *Tbx21* (**Fig. R12B**), distinguishing them from transitory effector-like exhausted cells (44). Tex2 and Tex3 exhibited significant enrichment in T_{SE}-related genes relative to Tex1 (**Fig. R12C**). Relative to Tex1, Tex2 downregulated canonical Tex TFs *Tox* and *Nr4a2* and Tex markers *Gzmk*, *Ccl3*, *Ccl4* and *Ccl5* (**Fig. R12D**) while Tex3 TILs also upregulated *Gzmb*, *Prf1*, *Plac8*, *Serpib9*, *Tnfrsf9* (4-1BB) and *Ly6c2* (**Fig. R12E**), suggesting better cytolytic and metabolic functions, also seen by pathway analysis (**Fig. R12F**). Thus, under the influence of orthogonal ACT, endogenous CD8⁺ TILs acquired an “infectious” activated-Tex state, transcriptionally closer to T_{SE},

with low *Tox* and upregulated effector machinery. Importantly, even though highly expanded clonotypes could distribute across more than one Tex subtype (**Fig. R12G**), they mostly acquired Tex2 and Tex3 during tumor regression.

Additional work will also be required to fully decipher the underpinning signaling and epigenetic mechanisms in the endogenous T cells, but we believe these are beyond the scope of the present manuscript.

Fig. R14 (A) Top fraction of endogenous CD8 TILs by state that belong to expanded clonotypes (medium, large and hyperexpanded). Bottom: number of cells by state assigned into specific frequency ranges of clonal expansion in each experimental condition. (B) Top Left: UMAP plot showing a low-dimensional representation of cell heterogeneity and

unsupervised clustering results of endogenous exhausted CD8 TILs (Pex and Tex) expanded by orthogonal ACT across all studied time points (from baseline to day 26 post ACT). Top Right: cluster composition/time point. Bottom part: dot plot showing clusters-specific markers. (C) Enrichment score of C5 gene signature (genes upregulated in the comparison showed in Extended Data Fig. 2i) in endogenous Tex subtypes (the enrichment score was calculated with AUCell). A non-parametric Kruskal Wallis test was used for multiple comparisons * $p < 0.05$, ** $p < 0.01$, *** $p < 0.001$, **** $p < 0.0001$. (D) Volcano plot showing differentially expressed genes between Tex2 and Tex1 subtypes. (E) Volcano plot showing differentially expressed genes between Tex3 and Tex1 subtypes. (F) GEO terms enrichment analysis of each exhausted subtypes. (G) Distribution of the top 4 expanded exhausted clonotypes between Tex1, Tex2 and Tex3 subtypes in baseline tumors, or during the response or escape phases following orthogonal ACT. Contour plots depict the subtypes covered by each clonotype.

(Figure 5a) GzmC is not detected in the effector memory cluster and only in the 'synthetic state' cluster - is this expected?

Response: Yes, this is expected. GzmC is specifically upregulated with the acquisition of the effector state, which requires the presence of both IL-2^V and IL-33 in the tumor microenvironment. Indeed, EM-like cells do not express Gzmc (**New Fig.7c**)

To what extent is this effector molecule co-regulated with GzmB?

Response: This is an important observation. They are indeed co-regulated, since cells in the new synthetic state upregulated both GzmC and GzmB, and there appears to be a correlation in the level of expression of both proteins. Remarkably, upon orthogonal ACT a proportion of transferred OT1 cells expressed high levels of GzmC during tumor regression, and these cells also expressed high GzmB. Furthermore, among OT1 cells we distinguished a second population of GzmC^{Lo/int} which coexpressed GzmB.

These GzmC^{high}/GzmB^{high} cells were not observed among the endogenous TIL upon orthogonal ACT. Endogenous cells upregulated nevertheless GzmC and became GzmC^{Lo/int}/GzmB⁺ cells (**Fig. R15**). Remarkably, this is only a unique characteristic of TILs upon orthogonal ACT, since canonical exhausted tumor specific CD8⁺ TILs at the steady state did not express Gzmc (**New Fig. 7b, and see Fig. R3b**).

Fig.R15: Co-expression of Granzyme B and Granzyme C in PD-1+ TILs during the response phase

PD-1 is not only a marker of exhausted cells, but an activation marker, upregulated on recently stimulated non-exhausted cells. Given the observed 'synthetic' state is TOX negative, to what extent is this merely activation-driven PD-1 expression? In which case, does this rather represent activation-driven co-inhibitory receptor expression rather than exhaustion-resistance? Evidence should be provided that this is a distinct state from that merely reflecting activation.

Response: We agree with the Reviewer that PD-1-expression likely reflects T-cell activation in the synthetic state. However, under physiologic conditions PD-1 functionally restrains activated T cells, as we (40) and many others have demonstrated. Importantly, cells in the C5-synthetic state do not resemble early activated CD8⁺ TILs (45). Indeed, in (Fig. 2e) we provide evidence that this state is not merely reflecting early activation, since none of the C5-annotated cells are projected on this state.

Our claim that C5-T_{SE} PD-1⁺ cells are functionally non-exhausted and indeed resistant to PD-1 suppression and exhaustion stems from several observations: first, we attribute the non-exhausted state of these chronically stimulated cells to the full suppression of TOX (Fig. 3c and Fig. 4a), the master transcriptional and epigenetic regulator of exhaustion (13-15). In addition, several TFs (i.e. *Nr4a1*, *Nr4a2*, *Nfatc1*, *Batf*, *Irf1*) that coordinately work with TOX to reinforce the exhausted phenotype (15, 16, 38, 39) are also downregulated or barely expressed in C5-T_{SE} vs canonical Tex (Fig. 3c and Fig. R6A). Furthermore, in the manuscript we show that whole NFAT/Nr4a/TOX-driven canonical exhaustion program is absent in C5-T_{SE} TILs (Fig. 2i and Fig.3i), and in sharply contrast with canonical TOX⁺ exhausted T cells, which are dysfunctional (13-15, 40) and (Fig. R6B, top), C5-TILs, are polyfunctional effector cells, with direct antitumor potential (Fig. 4 and Fig. R6B bottom). Indeed, we hypothesize that the latter is a direct consequence of TOX suppression, since it directly controls the genome accessibility of many effector genes (15). Moreover, the experimental evidence presented above that orthogonally engineered OT1 cells are directly resistant to suppression by PD-L1, complements the first mechanistic view of how exhaustion resistance is mediated by the orthogonal cytokines. Finally, while it is very well accepted that canonical Tex undergo metabolic exhaustion (41), which is associated with loss of mitochondrial fitness, we provide new data in the revised manuscript that suggest better mitochondrial fitness in C5-T_{SE} like cells relative to canonical Tex. Indeed, many genes upregulated in T_{SE} are involved in pathways associated with better mitochondrial metabolism (New Fig. 4c-e). Thus, our findings altogether point to a different cell state, distant from TOX⁺ canonical exhaustion.

Does exogenous (eg intraperitoneal) administration of IL-2v and IL-33 cause a similar anti-tumour response to orthogonal gene-engineered T cell transfer?

Response: Indeed, orthogonal ACT is superior to systemic therapy with recombinant IL-2^V and IL-33 and this was added to the revised manuscript.

To address this, we treated a first group of mice with recombinant mouse (rm)IL-33, using the lowest dose with reported antitumor efficacy (20) (0.5 μg/mouse once a day) and recombinant (r)IL2^V (20 μg/mouse twice a day) (21), starting on day 0 (Extended Data Fig.5a). A second group of mice received the same systemic cytokine treatment in combination with untransduced (UT) OT1, and a third group was treated with orthogonal engineered PD1d/2v and PD1d/33 OT1 cells. The treatment had to be stopped on day 8

due to severe adverse side effects in the groups receiving recombinant cytokines. Indeed, signs of toxicity (ruffled fur, reduced animal mobility) were apparent after each morning's injection with the cytokine cocktail. In sharp contrast, no systemic toxicity was observed when IL-33 and IL-2^V were delivered by transferred OT1.

Relative to PBS-treated control mice, systemic treatment with rIL-33/rIL-2V, did significantly delay tumor growth (**Extended Data Fig.5a**). Adoptively transferred UT OT1 did not add any therapeutic benefit relative to the cytokines alone. However, neither group performed as well as did the cohort treated with IL-2v/33-secreting OT1 T cells. The difference in therapeutic efficacy of rIL-2v+rIL-33 was already evident 8 days post ACT, when systemic treatment was discontinued, proving the therapeutic superiority of orthogonal ACT over systemic cytokine treatment. FACS analysis of TILs on day 9 post ACT showed that endogenous CD44⁺PD-1⁺ CD8⁺ TILs recovered from mice treated with rIL-33/rIL-2v upregulated GzmC in both the TCF1⁺ and the TCF1^{neg} compartments (**Extended Data Fig.5b**), thus suggesting that the PD-1⁺GzmC⁺ phenotype on endogenous CD8 TILs can be successfully induced in tumor-specific TILs by the two-cytokine combination *in vivo*.

Thus, even though systemic administration of IL-33/IL-2v was significantly associated with antitumor activity, and induced similar cell states, IL-33/2^V-secreting OT-1 T cells are needed for safe and potent tumor control. Finally, the fact that addition of UT OT1 to systemic cytokines did not mimic the antitumor effect of orthogonal engineered was surprising. FACS analysis, revealed that persistence of total UT OT1 was markedly shorter (**Extended Data Fig.5c**), while the fraction of GzmC⁺ cells was markedly smaller than what we observed following orthogonal engineering (**Extended Data Fig.5d**). Together, these results suggest that orthogonal engineered T cell therapy is superior to the therapy with the separated components, although the combination of IL-33 and IL-2v can mediate a C5-like state in endogenous TILs.

To what extent do IL-2v and IL-33 need to be expressed on different co-transferred cells, as is presented here? If they are co-expressed in a bicistronic vector, does this also result in potent tumor immunity. PD-1d may not be required if the cells are indeed 'exhaustion-resistant'

Response: We thank the Reviewer for this observation. Indeed, in the manuscript we demonstrate that PD1d is dispensable. Therefore, as the Reviewer suggests, we built and tested a new bicistronic vector combining IL-2v with IL-33. Both molecules were effectively expressed after T-cell transduction (data not shown) and OT1 T cells transduced with this bicistronic vector controlled tumors in a similar way that the initial approach of 1:1 mixed PD1d/2^V and PD1d/33 cells, indicating similar potent activation of tumor immunity (**New Extended Data Fig.8 b,c**).

In the revised manuscript we show that tumor progression was not associated with evolution toward T-cell exhaustion or other kind of CD8 T-cell hyporesponsiveness states, but rather with extinction of the optimal effector state adopted by TILs (**New Extended Data Fig.8**). This was associated with emergence of tumor escape variants that downregulated the surface expression of MHC-I, a common escape mechanism of melanoma tumors cells to immunotherapy(46). Notably, it is important to highlight that the baseline expression of MHC-class I in ex-vivo analyzed CD45^{neg} B16-OVA cells was below 50%, which validates the poor immunogenic nature of the tumor model we used for assessing the antitumor efficacy of orthogonal

ACT. Furthermore, we also attribute the therapeutic failure partly to the activation of counteracting CD4⁺ T-cell mediated regulatory mechanisms, most likely Treg, in the tumor microenvironment. Therefore, we plan to combine this therapy without proper lymphodepletion and with low-dose cyclophosphamide preconditioning regimens in the clinic, a pharmacological approach that preferentially depletes CD4 regulatory T cells (1)

The Methods state, 'Finally, engineered OT-1 T cells were adjusted based on the PD-1. IgG4 decoy expression prior to cell transfer.' What does this mean? Clarity is needed on exactly how transduced cells were prepared and adjusted before transfer

Response: As explained, above, since Pd1d was a common module, we used it to evaluate the transduction efficiency across conditions and ensured that the same dose of PD1d⁺ cells was infused in all experiments. In addition, we used it to ensure a 1:1 ratio of the orthogonally engineered cells. We added untransduced OT1 cells to the cell mixture infused to mice in order to also get the same number of total cells in all the groups. This has been clarified this in the revised manuscript.

The authors state: On day 12 and 15 mice were treated with i.v transfer of 5x10⁶ gene-engineered CD44+ CD62L+ TCF1+ OT-1 T cells or control nontransduced OT-1. It is not possible for live cells to be separated and transferred on the basis of intracellular TCF1 positivity - a more accurate description of how cells were isolated before transfer, and a measurement of their expression of CD44, CD62L, and TCF1 upon transduction of each of the constructs prior to transfer, would enable a more thorough assessment.

Response: We thank the Reviewer for this observation. As it shown in **Extended data Fig. 1d**, our *in vitro* expansion protocol is highly efficient in inducing CD44⁺TCF1⁺CD62⁺ central memory-like cells. There was no need to sort OT1 cell prior to ACT, since virtually all cells exhibited this phenotype.

Minor point: Typo on Line 81 - (100 m3) tumors

Response: We thank the Reviewer for this observation. This was corrected.

References

1. Herrera FG, Ronet C, Ochoa de Olza M, Barras D, Crespo I, Andreatta M, et al. Low-Dose Radiotherapy Reverses Tumor Immune Desertification and Resistance to Immunotherapy. *Cancer Discovery*. 2022;12(1):108-33.
2. Carmona SJ, Siddiqui I, Bilous M, Held W, Gfeller D. Deciphering the transcriptomic landscape of tumor-infiltrating CD8 lymphocytes in B16 melanoma tumors with single-cell RNA-Seq. *bioRxiv*. 2019:800847.
3. Hashimoto M, Araki K, Cardenas MA, Li P, Jadhav RR, Kissick HT, et al. PD-1 combination therapy with IL-2 modifies CD8+ T cell exhaustion program. *Nature*. 2022.
4. Eberhardt CS, Kissick HT, Patel MR, Cardenas MA, Prokhnevskaya N, Obeng RC, et al. Functional HPV-specific PD-1+ stem-like CD8 T cells in head and neck cancer. *Nature*. 2021;597(7875):279-84.
5. Rojas G, Carmenate T, Santo-Tomás JF, Valiente PA, Becker M, Pérez-Riverón A, et al. Directed evolution of super-secreted variants from phage-displayed human Interleukin-2. *Scientific Reports*. 2019;9(1):800.
6. Browaeys R, Saelens W, Saeys Y. NicheNet: modeling intercellular communication by linking ligands to target genes. *Nature Methods*. 2020;17(2):159-62.
7. Aibar S, González-Blas CB, Moerman T, Huynh-Thu VA, Imrichova H, Hulselmans G, et al. SCENIC: single-cell regulatory network inference and clustering. *Nature Methods*. 2017;14(11):1083-6.
8. Boccard J, Rutledge DN. A consensus orthogonal partial least squares discriminant analysis (OPLS-DA) strategy for multiblock Omics data fusion. *Anal Chim Acta*. 2013;769:30-9.
9. Lane N, Martin W. The energetics of genome complexity. *Nature*. 2010;467(7318):929-34.
10. Giles JR, Ngiow SF, Manne S, Baxter AE, Khan O, Wang P, et al. Longitudinal single cell transcriptional and epigenetic mapping of effector, memory, and exhausted CD8 T cells reveals shared biological circuits across distinct cell fates. *bioRxiv*. 2022:2022.03.27.485974.
11. Stelekati E, Chen Z, Manne S, Kurachi M, Ali M-A, Lewy K, et al. Long-Term Persistence of Exhausted CD8 T Cells in Chronic Infection Is Regulated by MicroRNA-155. *Cell reports*. 2018;23(7):2142-56.
12. Ataide MA, Komander K, Knöpper K, Peters AE, Wu H, Eickhoff S, et al. BATF3 programs CD8+ T cell memory. *Nature immunology*. 2020;21(11):1397-407.
13. Alfei F, Kanev K, Hofmann M, Wu M, Ghoneim HE, Roelli P, et al. TOX reinforces the phenotype and longevity of exhausted T cells in chronic viral infection. *Nature*. 2019;571(7764):265-9.
14. Scott AC, Dündar F, Zumbo P, Chandran SS, Klebanoff CA, Shakiba M, et al. TOX is a critical regulator of tumour-specific T cell differentiation. *Nature*. 2019;571(7764):270-4.
15. Khan O, Giles JR, McDonald S, Manne S, Ngiow SF, Patel KP, et al. TOX transcriptionally and epigenetically programs CD8+ T cell exhaustion. *Nature*. 2019;571(7764):211-8.
16. Seo H, Chen J, González-Avalos E, Samaniego-Castruita D, Das A, Wang YH, et al. TOX and TOX2 transcription factors cooperate with NR4A transcription factors to impose CD8⁺ T cell exhaustion. *Proceedings of the National Academy of Sciences*. 2019;116(25):12410-5.
17. Codarri Deak L, Nicolini V, Hashimoto M, Karagianni M, Schwalie PC, Lauener L, et al. PD-1-cis IL-2R agonism yields better effectors from stem-like CD8+ T cells. *Nature*. 2022.
18. Sockolovsky JT, Trotta E, Parisi G, Picton L, Su LL, Le AC, et al. Selective targeting of engineered T cells using orthogonal IL-2 cytokine-receptor complexes. *Science*. 2018;359(6379):1037-42.
19. Kalbasi A, Siurala M, Su LL, Tariveranmohabadi M, Picton LK, Ravikumar P, et al. Potentiating adoptive cell therapy using synthetic IL-9 receptors. *Nature*. 2022;607(7918):360-5.
20. Dominguez D, Ye C, Geng Z, Chen S, Fan J, Qin L, et al. Exogenous IL-33 Restores Dendritic Cell Activation and Maturation in Established Cancer. *The Journal of Immunology*. 2017;198(3):1365-75.
21. Carmenate T, Pacios A, Enamorado M, Moreno E, Garcia-Martinez K, Fuente D, et al. Human IL-2 mutein with higher antitumor efficacy than wild type IL-2. *Journal of immunology*. 2013;190(12):6230-8.

22. Zhao M, Kiernan CH, Stairiker CJ, Hope JL, Leon LG, van Meurs M, et al. Rapid in vitro generation of bona fide exhausted CD8⁺ T cells is accompanied by Tcf7 promoter methylation. *PLoS pathogens*. 2020;16(6):e1008555.
23. Vardhana SA, Hwee MA, Berisa M, Wells DK, Yost KE, King B, et al. Impaired mitochondrial oxidative phosphorylation limits the self-renewal of T cells exposed to persistent antigen. *Nature immunology*. 2020;21(9):1022-33.
24. Belk JA, Yao W, Ly N, Freitas KA, Chen YT, Shi Q, et al. Genome-wide CRISPR screens of T cell exhaustion identify chromatin remodeling factors that limit T cell persistence. *Cancer cell*. 2022;40(7):768-86.e7.
25. Villarreal DO, Wise MC, Walters JN, Reuschel EL, Choi MJ, Obeng-Adjei N, et al. Alarmin IL-33 Acts as an Immunoadjuvant to Enhance Antigen-Specific Tumor Immunity. *Cancer Research*. 2014;74(6):1789-800.
26. Kallert SM, Darbre S, Bonilla WV, Kreutzfeldt M, Page N, Müller P, et al. Replicating viral vector platform exploits alarmin signals for potent CD8⁺ T cell-mediated tumour immunotherapy. *Nature Communications*. 2017;8:15327.
27. Gao K, Li X, Zhang L, Bai L, Dong W, Gao K, et al. Transgenic expression of IL-33 activates CD8⁺ T cells and NK cells and inhibits tumor growth and metastasis in mice. *Cancer Letters*. 2013;335(2):463-71.
28. Peine M, Marek RM, Löhning M. IL-33 in T Cell Differentiation, Function, and Immune Homeostasis. *Trends in immunology*. 2016;37(5):321-33.
29. Bonilla WV, Fröhlich A, Senn K, Kallert S, Fernandez M, Johnson S, et al. The Alarmin Interleukin-33 Drives Protective Antiviral CD8⁺ T Cell Responses. *Science*. 2012;335(6071):984-9.
30. Yang Q, Li G, Zhu Y, Liu L, Chen E, Turnquist H, et al. IL-33 synergizes with TCR and IL-12 signaling to promote the effector function of CD8⁺ T cells. *European journal of immunology*. 2011;41(11):3351-60.
31. Baumann C, Bonilla WV, Fröhlich A, Helmstetter C, Peine M, Hegazy AN, et al. T-bet- and STAT4-dependent IL-33 receptor expression directly promotes antiviral Th1 cell responses. *Proceedings of the National Academy of Sciences*. 2015;112(13):4056-61.
32. Schiering C, Krausgruber T, Chomka A, Fröhlich A, Adelman K, Wohlfert EA, et al. The alarmin IL-33 promotes regulatory T-cell function in the intestine. *Nature*. 2014;513:564.
33. Gao X, Wang X, Yang Q, Zhao X, Wen W, Li G, et al. Tumoral Expression of IL-33 Inhibits Tumor Growth and Modifies the Tumor Microenvironment through CD8⁺ T and NK Cells. *The Journal of Immunology*. 2015;194(1):438-45.
34. Villarreal DO, Weiner DB. Interleukin 33: a switch-hitting cytokine. *Current opinion in immunology*. 2014;28:102-6.
35. Dwyer GK, Mathews LR, Villegas JA, Lucas A, Gonzalez de Peredo A, Blazar BR, et al. IL-33 acts as a costimulatory signal to generate alloreactive Th1 cells in graft-versus-host disease. *The Journal of clinical investigation*. 2022.
36. Hochberg Y. A Sharper Bonferroni Procedure for Multiple Tests of Significance. *Biometrika*. 1988;75(4):800-2.
37. Ogando J, Sáez ME, Santos J, Nuevo-Tapióles C, Gut M, Esteve-Codina A, et al. PD-1 signaling affects cristae morphology and leads to mitochondrial dysfunction in human CD8(+) T lymphocytes. *Journal for immunotherapy of cancer*. 2019;7(1):151.
38. Chen J, López-Moyado IF, Seo H, Lio C-WJ, Hempleman LJ, Sekiya T, et al. NR4A transcription factors limit CAR T cell function in solid tumours. *Nature*. 2019;567(7749):530-4.
39. Chen Y, Zander RA, Wu X, Schauder DM, Kasmani MY, Shen J, et al. BATF regulates progenitor to cytolytic effector CD8⁺ T cell transition during chronic viral infection. *Nature immunology*. 2021.

40. McLane LM, Abdel-Hakeem MS, Wherry EJ. CD8 T Cell Exhaustion During Chronic Viral Infection and Cancer. *Annual Review of Immunology*. 2019;37(1):457-95.
41. Gabriel SS, Tsui C, Chisanga D, Weber F, Llano-León M, Gubser PM, et al. Transforming growth factor- β -regulated mTOR activity preserves cellular metabolism to maintain long-term T cell responses in chronic infection. *Immunity*. 2021;54(8):1698-714.e5.
42. Lisci M, Barton PR, Randzavola LO, Ma CY, Marchingo JM, Cantrell DA, et al. Mitochondrial translation is required for sustained killing by cytotoxic T cells. *Science*. 374(6565):eabe9977.
43. Duraiswamy J, Turrini R, Minasyan A, Barras D, Crespo I, Grimm AJ, et al. Myeloid antigen-presenting cell niches sustain antitumor T cells and license PD-1 blockade via CD28 costimulation. *Cancer cell*. 2021.
44. Beltra J-C, Manne S, Abdel-Hakeem MS, Kurachi M, Giles JR, Chen Z, et al. Developmental Relationships of Four Exhausted CD8+ T Cell Subsets Reveals Underlying Transcriptional and Epigenetic Landscape Control Mechanisms. *Immunity*. 2020;52(5):825-41.e8.
45. Andreatta M, Corria-Osorio J, Müller S, Cubas R, Coukos G, Carmona SJ. Interpretation of T cell states from single-cell transcriptomics data using reference atlases. *Nature Communications*. 2021;12(1):2965.
46. Lee JH, Shklovskaya E, Lim SY, Carlino MS, Menzies AM, Stewart A, et al. Transcriptional downregulation of MHC class I and melanoma de-differentiation in resistance to PD-1 inhibition. *Nature Communications*. 2020;11(1):1897.

Decision Letter, first revision:

12th Dec 2022

Dear George,

We have received back the referee comments of your revised manuscript entitled, "Orthogonal Cytokine Engineering Enables Novel Synthetic Effector States in Tumor-rejecting CD8+ T Cells Escaping Canonical Exhaustion", which was seen by 2 of the original referees & one new referee to replace the original referee who declined to re-review. All three referees are endorsing publication of the study in *Nature Immunology*, pending toning down some statements as suggested by referee #4.

Please also note that the study should be considered as a Technical Report {although the format of such content is broadly similar to a Research Article}; when you are ready to resubmit the manuscript please select the "Technical Report" content type in the manuscript menu.

We therefore invite you to revise your manuscript taking into account all reviewer and editor comments. Please highlight all changes in the manuscript text file in Microsoft Word format. We will then commence our pre-accept editing process on that revised version.

* Include a "Response to referees" document detailing, point-by-point, how you addressed each

referee comment. If no action was taken to address a point, you must provide a compelling argument. This response will be sent back to the referees along with the revised manuscript.

* If you have not done so already please begin to revise your manuscript so that it conforms to our Article format instructions at <http://www.nature.com/ni/authors/index.html>. Refer also to any guidelines provided in this letter.

* Please include a revised version of any required reporting checklist. It will be available to referees to aid in their evaluation of the manuscript goes back for peer review. They are available here:

Reporting summary:

When submitting the revised version of your manuscript, please pay close attention to our [href="https://www.nature.com/nature-portfolio/editorial-policies/image-integrity">Digital Image Integrity Guidelines. and to the following points below:](https://www.nature.com/nature-portfolio/editorial-policies/image-integrity)

[REDACTED]

We hope to receive your revised manuscript in January after the New Year's holiday. If you cannot send it within this time, please let us know. We will be happy to consider your revision so long as nothing similar has been accepted for publication at Nature Immunology or published elsewhere.

Nature Immunology is committed to improving transparency in authorship. As part of our efforts in this direction, we are now requesting that all authors identified as 'corresponding author' on published papers create and link their Open Researcher and Contributor Identifier (ORCID) with their account on the Manuscript Tracking System (MTS), prior to acceptance. ORCID helps the scientific community

achieve unambiguous attribution of all scholarly contributions. You can create and link your ORCID from the home page of the MTS by clicking on 'Modify my Springer Nature account'. For more information please visit www.springernature.com/orcid.

Kind regards & Happy Holidays,

Laurie

Laurie A. Dempsey, Ph.D.
Senior Editor
Nature Immunology
l.dempsey@us.nature.com
ORCID: 0000-0002-3304-796X

Reviewers' Comments:

Reviewer #1:

Remarks to the Author:

The prior comments were adequately considered and addressed. I suggest keeping the data concerning reprogramming of endogenous CD8 TILs in the manuscript as I think it makes for a stronger study.

Reviewer #3:

Remarks to the Author:

The authors have gone to great lengths to address my previous concerns. In particular, I am now satisfied that they have clarified the specific functional requirement of the three orthogonal payloads, IL-2v, IL-33 and PD-1d, in the observed immunotherapy efficacy of their orthogonal gene engineering approach - finding that PD-1d is indeed not required for the observed effect, and that co-expression of IL-2v and IL-33 in a single bi-cistronic vector is alone sufficient to drive the observed tumour clearance.

The new data presented in Fig 5g-i, and the new transcriptional analyses provided, are compelling and strongly supportive of their conclusion that IL-33/IL-2v co-transduction drives a synthetic T cell state resistant to the co-inhibitory effects of PD-1 receptor signalling.

Together, I believe these findings are of significance to cellular immunotherapy field, describing exhaustion-resistant synthetic cell states programmed by orthogonal co-expression of distinct genetic payloads for optimal therapeutic efficacy.

[Rahul Roychoudhuri]

Reviewer #4:

Remarks to the Author:

Corria-Osorio et al. report that forced expression of mutant IL-2 plus IL-33 in adoptively transferred cells robustly enhances anti-tumor activity and gives rise to a phenotypically distinct population of T cells that lack typical signs of T cell exhaustion. The authors then make use of extensive and sophisticated bioinformatics analyses to argue that a significant proportion of IL-2 plus IL-33 stimulated T cells develop a unique phenotype that they consider a synthetic state. While I find the overall clinical observation very impressive, I am, as discussed in more detail below, less convinced by the conclusion that a unique phenotype was induced.

Major points:

While I agree with the authors that IL-2 plus IL-33 produces a specific phenotype and that some of the engineered cells are different from the cell types seen in non-engineered cells or in cells that have received a different type of treatment, I am not completely convinced that this is indeed a unique differentiation state. Very few analysis have been done to infer the uniqueness of these populations, for example the study lacks comparison to T cells found in acute infections.

In this sense, the C5 cluster in Figure 2d looks very similar to the C4 cluster, which raises the question of whether they are different phenotypes or whether they may not simply reflect the same cell type but in a different activation state because of exposure to different cytokines.

Furthermore, the authors claim to have found a unique progenitor population after IL-2v/IL-33 exposure. How does this conclusion fits with the data shown in Figure 2d, where the progenitors of PD-1d/IL-2v/IL-33-treated mice are in the same cluster as cells from other treatments. Ext. Fig. 8a raises similar questions. The scRNA data in this figure are difficult to read, but it looks like the population is lost over time and converts into normal memory cells. This also argues against a stable and unique population having been formed. Furthermore, to call this population a unique progenitor population would require functional assays and adoptive transfers, which have not been performed.

So, all in all, I am not completely convinced that this is a unique population, but in any case, the observations made by the authors are very interesting and the data are clear and new. So I would recommend that the authors back off somewhat on the conclusion that it is a unique population.

The bioinformatics data are very extensive and sometimes appear somewhat redundant. Some shortening would streamline the paper and make it more accessible to readers.

The graphic quality of the figures needs improvement. Often panels and labels are too small to be read and the resolution is too low. For example, the population in ExtData Fig. 8A is not visible.

I am not quite sure why the authors included the FTY720 treatment experiment, but the result is very surprising. I almost get the impression that the treatment did not work, which sometimes happens. So far, the authors have not provided evidence that the FTY720 treatment worked, i.e., that the cells in the circulation were reduced following the treatment. Such evidence is essential, otherwise the data should be removed.

The authors refer to baseline Gzmb expression in OT-1 before treatment as shown in Figure 4a, but there is no baseline data of OT-1 in Figure 4a. The same is true for Tox.

It is unclear what kind of data is shown in Figure 4d. It says protein levels, but it looks more like a measurement of RNA expression. The authors should clarify what is shown in the legend.

2/3 of the manuscript and the core conclusion are derived from the extensive bioinformatics analysis. Given this extent, I am surprised that all three people who are listed in the authors contribution section to have performed the bioinformatics analysis are listed only as middle authors.

The authors say that IL-2v/IL-33-induced progenitors remain Tox-negative and refer to Figure 8a of Ext, which shows no Tox expression data.

Minor points:

In the legend of Figure 1, $\geq n5$ is indicated, but in panel B, the IL-2v alone group has only 4 mice.

The term "orthogonal" should be explained, what is the difference compared to autocrine or paracrine?

Extended data 1F, the colors in the legend in the figure and the lines of a group do not match.

Extended Data 1F, while tumor growth rates look very different between early and late ACT, survival rates look much more similar, why is that?

The use of the term "bulk" in the following lines is confusing and should be removed: "we analyzed bulk CD8+ TILs from different ACT settings using single-cell RNA."

Author Rebuttal, first revision:

Reviewers' Comments:

Reviewer #1:

Remarks to the Author:

The prior comments were adequately considered and addressed. I suggest keeping the data concerning reprogramming of endogenous CD8 TILs in the manuscript as I think it makes for a stronger study.

Response: We thank the Reviewer for this kind evaluation of our work and the suggestion. [REDACTED]

Reviewer #3:

Remarks to the Author:

The authors have gone to great lengths to address my previous concerns. In particular, I am now satisfied that they have clarified the specific functional requirement of the three orthogonal payloads, IL-2v, IL-33 and PD-1d, in the observed immunotherapy efficacy of their orthogonal gene engineering approach - finding that PD-1d is indeed not required for the observed effect, and that co-expression of IL-2v and IL-33 in a single bi-cistronic vector is alone sufficient to drive the observed tumour clearance.

The new data presented in Fig 5g-i, and the new transcriptional analyses provided, are compelling and strongly supportive of their conclusion that IL-33/IL-2v co-transduction drives a synthetic T cell state

resistant to the co-inhibitory effects of PD-1 receptor signalling.

Together, I believe these findings are of significance to cellular immunotherapy field, describing exhaustion-resistant synthetic cell states programmed by orthogonal co-expression of distinct genetic payloads for optimal therapeutic efficacy.

(Rahul Roychoudhuri)

Response: We thank the Reviewer for this kind evaluation of our work. We are advancing these concepts to the clinic in our center.

Reviewer #4:

Remarks to the Author:

Corria-Osorio et al. report that forced expression of mutant IL-2 plus IL-33 in adoptively transferred cells robustly enhances anti-tumor activity and gives rise to a phenotypically distinct population of T cells that lack typical signs of T cell exhaustion. The authors then make use of extensive and sophisticated bioinformatics analyses to argue that a significant proportion of IL-2 plus IL-33 stimulated T cells develop a unique phenotype that they consider a synthetic state. While I find the overall clinical observation very impressive, I am, as discussed in more detail below, less convinced by the conclusion that a unique phenotype was induced.

Major points:

R4-Q1a While I agree with the authors that IL-2 plus IL-33 produces a specific phenotype and that some of the engineered cells are different from the cell types seen in non-engineered cells or in cells that have received a different type of treatment, I am not completely convinced that this is indeed a unique differentiation state. Very few analysis have been done to infer the uniqueness of these populations, for example the study lacks comparison to T cells found in acute infections.

Response: We thank the Reviewer for the revision of our work.

We respectfully submit that our exhaustive computational and experimental characterization does support the uniqueness of the C5-T_{SE} effector-like state. Indeed, these cells were compared phenotypically, transcriptionally, and functionally with canonical effector and exhausted TILs in our models, and computationally with all available mouse and human data, as detailed below:

i) We conducted several comparative analyses to demonstrate that the synthetic effector (T_{SE}) is transcriptionally and functionally different from canonical, terminal exhausted TILs (T_{EX}) (**Fig. 2, Fig. 3 and Fig.4**). Notably, this novel T_{SE} cell state does not express TOX, the master transcriptional and epigenetic regulator of exhaustion[1-3]. This piece of evidence alone could be enough to define T_{SE} as novel and unique PD-1⁺ effector-like TIL state because canonical PD-1⁺ T_{EX} TILs cannot be generated in the absence of TOX [1-3]. In addition, several transcription factors (TFs) (i.e. *Nr4a1*, *Nr4a2*, *Nfatc1*, *Batf*, *Irf1*) that coordinately work with TOX to reinforce the exhausted phenotype [3-6] are also downregulated or barely expressed in C5-T_{SE} vs. canonical T_{EX} (**Fig. 3c and Fig. R1A**). Furthermore, in the manuscript we showed that the whole NFAT/Nr4a/TOX-driven canonical exhaustion program is absent in C5-T_{SE} TILs (**Fig. 2i and Fig.3i**).

Moreover, in sharp contrast with canonical TOX⁺ exhausted T cells, which are dysfunctional [1-3, 7] and (see **Fig. R1B, top**), C5-TILs, are polyfunctional effector cells, with direct antitumor potential (**Fig. 4 and Fig. R1B bottom**). Indeed, we hypothesize that the latter is a direct consequence of TOX suppression,

since it directly controls the genome accessibility of many effector genes [3]. Furthermore, the experimental evidence that C5-T_{SE} TILs are resistant to suppression by the PD-1/PD-L1 axis (Fig.5) is another major difference with canonical T_{EX} cells, which are functionally restricted by this inhibitory checkpoint [7].

In addition, it is very well accepted that canonical T_{EX} undergo metabolic exhaustion [8], which is associated with loss of mitochondrial fitness. The computational analysis depicted in (Fig. 4f-g) clearly suggests better mitochondrial fitness in C5-T_{SE} like cells relative to canonical T_{EX}. Indeed, many genes upregulated in T_{SE} relative to T_{EX} are involved in pathways associated with better mitochondrial metabolism.

Fig. R1: A) Radar plots showing differentially expressed key exhaustion-related TFs between T_{SE} and T_{EX} TILs. B) Analysis of the polyfunctionality between PD-1⁺ TOX⁺ canonical OT1 TILs (isolated from OT1 chimeric mice) and TSE-like OT1 TILs harvested upon PD1d/2v/33-ACT. In both cases, isolated OT1 TILs were stimulated with 10 ng/ml of the cognate OVA peptide for 4 hours in the presence of brefeldin A.

ii) We also showed that the C5-T_{SE} state was distinct not only from canonical T_{EX} TILs (Fig. 3c), but also from effector-like states observed in the context of acute and chronic viral infections (Extended Data Fig. 3e,f). Indeed, we demonstrated that C5-T_{SE} TILs are different from both short-lived effector CD8⁺ T cells (SLEC, effector cells observed at day 8 post-acute viral infection) and PD-1⁺ CX₃CR1⁺ exhausted intermediate effector cells, recently described by several groups in the context of chronic viral infection [9-12]. Indeed, these cells do not upregulate *Gzmc*, but expresses *Cx3cr1* and *Klrg1*, two markers not expressed by the TIM3⁺PD-1⁺Gzmc⁺ T_{SE} state induced by orthogonal engineering (Extended Data Fig. 4a). In addition, while CD4⁺ T-cell help is required for the formation of the CX₃CR1⁺ effector state [12], CD4⁺ cells appeared to be dispensable or even deleterious in the context of our approach. Indeed, depletion of CD4⁺ cells led to more impactful results following IL2v/33 ACT, with more mouse cures, in the absence of general lymphodepletion (Extended Data Fig. 8f). Notably, C5-T_{SE} displayed the lowest expression of *Tox* and the highest expression of multiple granzymes relative to these naturally occurring effector-like cell states.

iii) We extended our comparisons to publicly reported human T cells. We could not identify C5-T_{SE} cells in human steady-state CD8⁺ TILs in three tumor type datasets in immunotherapy-naïve patients (Fig. 3f) or in TILs from patients during response to anti-PD-1 (Fig. 3g). Furthermore, we could not identify C5-like cells among circulating CD8⁺ CD19/4-1BBz-CAR-T cells during peak expansion *in vivo* in four patients with durable CAR-T persistence [13], where the majority of CAR-T cells corresponded to T_{EX}-like cells (Fig. 3h).

iv) We compared T_{SE} TILs with mouse CD8⁺ “better effectors” (T_{BE}) reported recently following systemic administration of recombinant anti-PD-1_IL2v bi-specific immunocytokine [14]. Although a subset of PD-1_IL2v-induced T_{BE} TILs reached a non-canonical state expressing low levels of *Tox* (Extended Data Fig. 3g) and projecting on a C5-like space by ProjecTILs (Fig.3j), these T_{BE} cells were still distinct from C5-T_{SE} TILs induced by orthogonal cytokine cell engineering; C5-T_{SE} TILs expressed no *Tox*, exhibited lower *Tbx21*

and *Cx3cr1*, and most notably they exhibited markedly higher granzyme levels than T_{BE} TILs (**Fig.3j**, **Extended Data Fig. 3g**)

v) Finally, we showed that the T-cell state induced upon systemic administration of anti-PD-1 in combination with IL-2 in the context of chronic viral infection [15] is distinct from our $C5-T_{SE}$ (**Fig. 6k**). Although the former cells downregulated *Tox* [15], they were mostly classified as effector-memory (EM)-like cells upon unbiased projection in our ACT reference map (**Fig. 6k**).

Altogether, we hope that this reviewer will agree that this data indicate that $C5-T_{SE}$ is a novel and unique, synthetic state endowed with powerful effector properties, seemingly never seen naturally to date at the steady state or in the context of conventional immunotherapies in mice or humans, but massively acquired *in vivo* by exogenous T cells following orthogonal-engineering ACT with IL-2v and IL-33.

R4-Q1b In this sense, the C5 cluster in Figure 2d looks very similar to the C4 cluster, which raises the question of whether they are different phenotypes or whether they may not simply reflect the same cell type but in a different activation state because of exposure to different cytokines.

Response: We thank the Reviewer for raising this point.

Since the $C5-T_{SE}$ is a $PD-1^+ TCF1^{neg}$ effector like cell state, its similarity with cluster C4 (T_{EX} -like cells) is expected. Furthermore, its projection on the T_{EX} space of the TIL reference map is also not surprising, as this is the closest/more similar cell state of that UMAP space, and this was the initial version of our TIL Reference map [16], which had not been trained with the new data of the synthetic states (the reference only included prior public data). This has been also clarified in the revised manuscript (lines 134-136). The same happens, for instance, if the so-called $PD-1^+ TCF1^{neg}$ “better effectors TILs” (T_{BE}), reported recently following systemic administration of recombinant anti-PD-1_IL2v bi-specific immunocytokine, which are also distinct of canonical T_{EX} [14] are projected onto the initial version of our TIL Reference map [16] (**Fig. R2**).

Fig. R2: Projection of better effectors” (T_{BE}) onto the reference TIL map using ProjectTILs.

The observation that supports our claim about a different PD-1+ effector program rather than an activated T_{EX} cell state is that $C5-T_{SE}$ does not express *TOX*, the master transcriptional and epigenetic regulator of exhaustion [1-3]. Indeed, as commented above, we respectfully believe that this piece of evidence is enough to support this conclusion, since T_{EX} cells can be neither generated nor persist in the absence of *TOX* [1-3]. In addition, canonical T_{EX} TILs do not downregulate *Tox* upon activation, as it is depicted in (**Fig. R3A**) in the context of PD-1 blockade, and in (**Fig. R3B**) upon ablation of the negative regulator *Zc3h12a/REGNASE-1* in adoptively transferred $CD8^+$ T cells [17]; here *Tox* is either upregulated or does not change, respectively, in the $CD69^{high}$ T_{EX} TILs.

Fig. R3: A) Projection of mouse CD8 TILs harvested upon systemic treatment with anti PD-1 (Codarri Deka, L. et al. 2022)[14] onto the TIL reference map using ProjecTILs[16]. Contour plots depict the clusters covered by each cell dataset and bar plots are showing the cluster composition. The radar plot shows the relative expression of important gene markers between TILs projected on T_{EX} from the anti PD-1 treatment and the reference T_{EX} cell state from the TIL map. B) Similar to A, data was retrieved from (Wei, J. et al. 2019) [17].

Finally, further evidence that C5 cells are profoundly different than C4 Tex cells comes from the analysis of the TF regulatory networks, as discussed above (please see also additional data in **Fig. R8** and discussion on pages 8 and 9 here), which revealed an entire TF network established in the C5 state, which is absent in C4 cells. Finally, the identification of C6 cells (**Fig. 3a-b**), which appear as the most likely precursors of C5, further lends support to the claim that these cells follow a distinct commitment program.

R4-Q1c Furthermore, the authors claim to have found a unique progenitor population after IL-2v/IL-33 exposure. How does this conclusion fits with the data shown in Figure 2d, where the progenitors of PD-1d/IL-2v/IL-33-treated mice are in the same cluster as cells from other treatments.

Response: We thank the Reviewer for raising this point.

We think that this population was not detected in our first scRNAseq experiment (**Fig. 2**) because we had not reached enough numbers of high-quality sequenced cells to identify it as a discrete cluster since this analysis was done on ~ 1,200 CD8 TILs obtained from all conditions. However, this population was detected in our second scRNAseq experiment as a discrete cluster (**cluster C6 in Fig. 3a-b**), where we collected sufficient CD8 T cells (more than 40,000) to analyze endogenous CD8⁺ and exogenous OT1 TILs separately.

R4-Q1d Ext. Fig. 8a raises similar questions. The scRNA data in this figure are difficult to read, but it looks like the population is lost over time and converts into normal memory cells. This also argues against a stable and unique population having been formed.

Response: Indeed, this figure illustrates the dynamics of OT1 T-cell states over time. We have improved its resolution in the revised manuscript as well as the legend description to make this point clear.

The Reviewer is right in pointing out that the C5-T_{SE} cell state is lost over time, which correlated with the escape phase. However, we hypothesize that this is not due to the instability of this cell state per se, but rather, as discussed in the manuscript (lines 330-342), a direct consequence of lack of differentiation of precursor-like OT1 TILs (most likely effector memory cells, see **Fig. 5g**) to the C5-T_{SE} state, caused by lack

of antigen recognition, since tumor escape variants downregulated the surface expression of MHC-I as early as 12 days post ACT (**Extended Data Fig. 8d**). A further escape mechanism may have been mediated by the activation of counteracting CD4⁺ T-cell mediated regulatory mechanisms, most likely Treg, which, as it is very well known, can suppress effector CD8⁺ T-cell differentiation [18-20]. Indeed, depletion of CD4⁺ cells improved the therapeutic efficacy (**Extended Data Fig. 8e,f**).

We also respectfully submit that the uniqueness and/or stability of terminally differentiated effector CD8⁺ T-cells, like the C5-T_{SE} state, cannot be defined by their life span. For instance, the short-lived effector cells (SLEC), a unique and stable phenotype that dominates the antiviral CD8 T cell response by day 8 upon acute viral infections, die by programmed apoptosis within approximately 1 week after its generation [7, 21-23].

R4-Q1e Furthermore, to call this population a unique progenitor population would require functional assays and adoptive transfers, which have not been performed.

Response: The Reviewer is right. For that reason, we have softened this claim by calling them precursor-like cells, since they express several well-known precursor markers like *Tcf7*, *Ccr7* and *IL7R* (**Fig 5b and Fig 7a-d**)

So, all in all, I am not completely convinced that this is a unique population, but in any case, the observations made by the authors are very interesting and the data are clear and new. So I would recommend that the authors back off somewhat on the conclusion that it is a unique population.

Response: We thank the Reviewer for this remark. However, as discussed above, we respectfully submit that we provide enough evidence in the manuscript to sustain that the breakthrough finding of our work is that orthogonal engineering of T cells with IL-2v and IL-33 positions the cells in a unique T-cell effector program that can handle chronic antigen stimulation without upregulating TOX and entering to canonical exhaustion. This is the first time that such PD-1⁺TOX^{neg} state is made possible, freeing effector T cells from the natural constraints of exhaustion in the context of chronic antigen stimulation.

Notably, the uniqueness of this novel differentiation program is strictly dependent on the presence of the novel IL-2 variant which does not engage CD25. Indeed, careful analysis of the states reached under different experimental conditions revealed that this factor was primarily responsible for promoting downregulation of TOX, which we consider as one of the founding pillars of the new synthetic state. This was true in both exogenous OT1 T cells (**Fig 3 and Fig. 7**) as well in endogenous PD-1⁺ Tex and Pex CD8⁺ TILs (**Fig. R4** not included in the current manuscript due to lack of space).

Fig. R4: Analysis of TOX expression in endogenous Tex TILs recovered at day 12 post ACT (A) or Pex (B) from mice treated with PD1d/2^V/33, PD1d/2^V and PD1d/33. Data collected from 5 independent experiments n= 5-6 mice/group. One-way ANOVA test in combination with a Dunnet Test to correct for multiple comparisons was used * p<0.05, ** p<0.01, *** p<0.001, ****p<0.0001. Naïve CD8 T cells isolated from the spleen of non-tumor bearing mice were (gray) used as negative control for the expression of the studied markers.

Thanks to constitutive secretion by the gene engineered cells, TILs had sustained exposure to IL-2v, which we hypothesize is the reason of sustained TOX suppression and exhaustion resistance. To model this, we used a recently published protocol of sustained antigen stimulation of naïve CD8 T cells, which recapitulates exhaustion *in vitro* [24]. Isolated naïve OT1 from spleens were used either directly for the sustained antigen stimulation assay (positive control of TOX+ canonical exhaustion [24]) or pre-exposed for 3, 7 or 12 days to IL-2v upon activation with anti-CD3/anti-CD28 *in vitro* (Fig. R5A), which induces a central memory-like (T_{CM}) CD62L⁺CD44⁺ TCF1⁺ phenotype from day 3. Following 5 days of repeat peptide stimulation in the absence of cytokines, we interrogated cells on day 6 for expression of PD-1 and TOX. In addition, we restimulated them with OVA peptide for 4 hours and measured expression of IFN_γ and TNF_α as surrogate markers of antitumor effector functions.

Fig. R5: A) Experimental Design. *In vitro* expansion protocol. B) Analysis of PD-1, TOX, IFN_γ and TNF_α expression after 5 days of peptide stimulation. For IFN_γ and TNF_α expression the cells were 4-hours re-stimulated with the peptide in the presence of Brefeldin A.

As shown in (Fig. R5B), OT1 T cells pre-exposed *in vitro* for a week or more to IL-2 v and then subjected to repeat TCR stimulation, upregulated PD-1 but not TOX. Conversely, most naïve OT1 cells that had not been exposed to IL-2v differentiated to PD-1⁺TOX⁺ canonical exhausted cells. Interestingly, TOX expression in OT1 cells at the end of repeat antigen stimulation was inversely proportional to the length of exposure to IL-2v, and gradually decreased from no exposure (naïve) to 12-day exposure. In addition, it was inversely proportional to the cell polyfunctionality (IFN γ /TNF α co-expression). Thus, from this we conclude that although under normal conditions activated CD8⁺ T cells upregulate TOX upon sustained antigen stimulation (the only driver of TOX expression [2, 3]) and enter an exhaustion differentiation program, they can be driven towards a PD-1⁺TOX^{neg} polyfunctional state, by virtue of long-term (>7 days) exposure to IL-2v, which commits them to a non-exhaustion effector program.

We next used a similar experimental set up, but this time to test whether previously *in vitro* chronically stimulated, PD-1⁺ TOX⁺ dysfunctional CD8 T cells (Fig. R6A) could be functionally rescued by exposure to IL-2v. Indeed, as shown in (Fig. R6B), a seven-day treatment with IL-2v alone (10 IU/mL) downregulated the expression of both PD-1 and TOX and functionally rescued the cells. Notably, only PD-1, but not TOX, was upregulated again upon a second round of TCR-mediated chronic stimulation (Fig. R6C), and these cells conserved a higher level of IFN γ secretion, and a 9-fold more production of TNF α relative to the end of the 1st round of chronic stimulation (Fig. R6A), two effector molecules that are epigenetically controlled by TOX [2, 3].

Fig. R6: Analysis of PD-1, TOX, IFN γ and TNF α expression upon a 1st round of 5 days sustained of Ag stimulation (A), IL-2V-mediated homeostatic resting (B) and a 2nd round (5 days) of chronic TCR stimulation. For IFN γ and TNF α expression the cells were 4-hours re-stimulated with the peptide in the presence of Brefeldin A. Naïve OT1 T cells were used as negative control for the FACS staining.

How IL-2 signaling directed through the IL-2R $\beta\gamma$ receptor chains achieve suppression of TOX will require additional investigation. However, it appears to be part of a transcriptional and epigenetic reprogramming that provides the basis for producing novel cell states which profoundly deviate from canonical exhaustion upon antigen engagement. Notably, IL-2R driving TOX downregulation in exhausted CD8⁺ T cells was just recently reported [14, 15], yet in the revised manuscript we show that both T_{SE} and T_{SP} cell states are still unique and expressed the lowest level of TOX amongst these novel cell states (**Fig. 3j** and **Fig. 6k**).

The fact that IL-2v is associated with high level of intratumoral persistence of TOX^{neg} CD8⁺ T cells in the context of sustained antigen stimulation (**Fig. 1d**, **Extended Data Fig. 2b** and **Fig. 7**) is also a novel and remarkable finding, since TOX expression is required for persistence under similar conditions [1-4]. Interestingly, a new regulon analysis [25], incorporated in the revised manuscript (**Fig. 7 e-h**), suggests that IL-2v regulates this process by activating a transcription factor (TF) network motif comprising *Fos/1*, *Egr1*, *Myc*, *Irf8* and *Batf3*, which might be involved in the metabolic support of sustained gene expression and protein translation (**Fig. 7 g-h**), both highly energetically costly cellular processes [26, 27], which are dampened in canonical exhausted CD8 T cells [27]. Indeed, the AP-1 transcription factor *Fos/1* has been associated with long-term persistence of CD8 T cells in the context of chronic viral infections [28], while *Batf3* has been recently shown to be a critical TF for CD8 T-cell memory formation and survival via negative regulation of the proapoptotic factor BIM [29]. Indeed, T cells that lacked *Batf3* succumbed to an aggravated contraction and had a diminished memory response [29]. Thus, IL-2v alone contributes to reprogram gene-engineered tumor specific TILs to escape TOX⁺ canonical exhaustion, endows them with superior metabolic transcriptional programs, and positions them in an activated, memory-like state post-ACT, which can persist in the absence of TOX (**Fig. 7**). We also believe that our findings of key anabolic and mitochondrial pathways being specifically upregulated in the T_{SE} and T_{SP} states (**Fig. 4f-g**, **Fig. 6d-e** and **Fig. 7**) are highly relevant in explaining how exhaustion resistance is bestowed, since TOX⁺ exhaustion has been increasingly recognized as a metabolic and mitochondrial failure [8].

[REDACTED]

We have recently fully inferred the transcriptional TF-gene regulatory network of T_{SE} and T_{SP} cell states relative to canonical TOX⁺ Pex and Tex by calculating the regulon activity of more than 800 TFs with SCENIC [25], a computational method for gene-regulatory network (GRN) reconstruction (**Fig. R8A**). Notably, after unbiased clustering analysis, we could clearly identify 4 clusters of regulons (i.e. modulons) that were differentially active across canonical and synthetic T cells states (**Fig. R8B**). These cell state-specific modulons also constitute highly connected network motifs in the SCENIC-inferred gene regulatory network (**Fig. R8C**). In addition, by using orthogonal partial least squares discriminant analysis (OPLS-DA) [30], a computational approach to discriminate between two or more groups (classes) using multivariate data, we could identify well-known exhaustion related TFs such *Eomes*, *Maf*, *Irf1*, *Prdm1*[7] within the set of TFs with the highest statistical power to discriminate between T_{EX}-TILs from the other states (**Fig. R8D**) (Please note that the regulon of TOX cannot be calculated using SCENIC, because this TF does not recognize a linear DNA motif, however TOX can be analyzed as a target of other TFs of the network).

Fig. R8: A) Schematics summarizing the Regulon/Modulon Analysis using SCENIC. B) Unsupervised clustering analysis based on the regulon activity across canonical and synthetic T-cell states of 848 TFs. C) SCENIC-predicted gene regulatory network, the size of each TF (ball) reflects the discriminant power calculated using OPLS-DA. D) Top_13 most discriminant TFs for Tex. In red we highlight those that are well-known key TFs E) Top_13 most discriminant TFs for TSE. In red we highlight those directly regulated by IL-2V F) Distribution of incoming regulatory links (edges in the network) of a set of relevant cytotoxic and effector target molecules.

Interestingly, we could not identify canonical exhaustion-related TFs within the most discriminant modulon for T_{SE} (Fig. R8E), confirming that this cell state has a distinct and unique transcriptional wiring. Notably, 6 of these TFs (*Batf3*, *Irf8*, *Jund*, *Atf3*, *Atf4*, and *Xbp1*) were also identified as part the transcriptional circuit induced by IL-2v, and are interpreted to be involved in the metabolic support to sustained gene expression and protein translation (Fig.7). Since these metabolic/mitochondrial processes are required for sustained killing by cytotoxic CD8 T cells [31], we asked whether they were also involved in the direct regulation of the many effector molecules that compose the specific cytolytic machinery of T_{SE} TILs. Remarkably, most of the upstream regulators of the cytotoxic/effector molecules in the SCENIC-inferred TF-gene regulatory network belong, almost exclusively, to the T_{SE}-specific modulon (Fig. R8F), and the above-mentioned IL-2v-related TFs are forming part of this transcriptional regulatory circuit (Fig. R8F). Thus, IL-2v could support both the better metabolic/mitochondrial fitness and cytotoxic potential of T_{SE} TILs by regulating a common

transcriptional core, that is not highly active in canonical Tex TILs. (**Fig. R8E**). Indeed, we hypothesize that this set of TFs are the key mediators of IL-2v-induced exhaustion resistance.

Collectively, based on these findings we can speculate that lack of TOX expression induced by IL-2v leads to a major epigenetic reprogramming in chronically stimulated synthetic T cells, since TOX is responsible for the chromatin remodeling necessary for the commitment to canonical exhaustion [2, 3], thus allowing chromatin accessibility in regions that would be otherwise inaccessible during chronic TCR stimulation. This could explain the transcriptional activities described above.

Further dissecting the above regulatory pathways will require in depth biochemical and molecular interrogation, along with validation of hypotheses using knockout methods. As such, we respectfully request that this work be part of a follow up manuscript. However, we discuss the above hypotheses in the Discussion section of the revised manuscript.

R4-Q2 The bioinformatics data are very extensive and sometimes appear somewhat redundant. Some shortening would streamline the paper and make it more accessible to readers.

Response: The Reviewer is right in pointing out the extensive computational analysis of this work. However, we respectfully submit that it was necessary to clearly demonstrate the novelty and uniqueness of the described T-cell states. We respectfully propose to keep it like that, especially this will be submitted as a Technical Report manuscript.

Regarding the redundancy, we respectfully submit that is minimal and specifically between **fig.2** and **fig.3**. Indeed, as commented above, due to the low-resolution of the 1st scRNAseq experiment, that was performed on total CD8⁺ TILs (**Fig.2**), we asked whether the novel C5 effector state was acquired both by transferred and endogenous CD8⁺ TILs. To this end, we repeated the scRNAseq experiment, interrogating separately FACS-sorted OT1 and endogenous CD44⁺CD8⁺ TILs (**Fig.3a**). Therefore, we had to go through the annotation process of the new clusters again (**Fig. 3b,c, Extended Data Fig. 3a-c**), which, to our opinion, is the only redundant analysis of the manuscript.

For the sake of the clarity, we list the scientific aims that guided each bioinformatic analysis showed in the manuscript.

- i) **Fig.2.** To illustrate the discovery and first characterization of the novel **effector CD8⁺ T-cell state** in the context of the low-resolution first scRNAseq experiment
- ii) **Fig. 3a-d, Extended Data Fig. 3a-d.** To demonstrate that during tumor regression, only adoptively transferred, orthogonally engineered cells differentiate to the novel C5-T_{SE} effector-like state.
- iii) **Extended Data Fig. 3e,f.** to show that C5-T_{SE} was different from previously described effector states such as short-lived effector CD8⁺ T cells or CX₃CR1⁺ exhausted intermediate effector cells.
- iv) **Fig. 3f-i.** To find out whether this novel effector state may be naturally present but may have gone unnoticed in human steady-state CD8⁺ TILs in three tumor type datasets in immunotherapy-naïve patients (**Fig. 3f**) or in TILs from patients during response to anti-PD-1 (**Fig. 3g**) or among circulating CD8⁺ CD19/4-1BBz-CAR-T cells during peak expansion *in vivo* in four patients (**Fig. 3h**) or upon combining adoptive transfer with PD-1 blockade (**Fig. 3i**).

v) **Fig.3j, Extended Data Fig. 3g.** To demonstrate that C5-T_{SE} TILs were still different from CD8⁺ “better effectors” (T_{BE}) reported recently following systemic administration of recombinant anti-PD-1_IL2v bi-specific immunocytokine

Together, this data demonstrate that C5-T_{SE} is a novel and unique T-cell state.

vi) **Fig. 4f,g, Extended Data Fig. 4c,d.** To demonstrate that, relative to T_{EX} TILs, the C5-T_{SE} cell state was enriched in biological process involved in energy production, effector molecule synthesis and cell killing (**Fig. 4f**), as well as in genes associated with better mitochondrial fitness such as cristae formation and mitochondrial translation (**Fig. 4g**).

vii) **Fig. 5e.** To show that T_{SE} cells were not enriched in CD8⁺ T-cell gene signatures of PD-1-mediated inhibition.

viii) **Fig. 6b-f.** To demonstrate that the novel precursor-like C6-T_{SP} was transcriptional different from canonical P_{EX}.

ix) **Fig. 6g.** Trajectory and pseudotime inference analysis of OT1 TILs to show that C6-T_{SP} cell state could be the precursor of the C5-T_{SE}.

x) **Fig. 6k, Extended Data Fig. 6d.** To demonstrate that C6-T_{SP} TILs were still different from atypical precursor-like CD8⁺ T cells reported recently following systemic administration of anti-PD-1 in combination with IL-2.

xi) **Fig. 7.** To characterize how IL-2v engineering contribute to key programs to the T_{SE} state

xii) **Extended Data Fig. 8a.** To show that tumor progression is not associated with exhaustion of the transferred OT1 TILs.

R4-Q3 The graphic quality of the figures needs improvement. Often panels and labels are too small to be read and the resolution is too low. For example, the population in ExtData Fig. 8A is not visible.

Response: We thank the Reviewer for this observation.

The resolution of the figures is 300 ppi, which is the requested resolution by the journal. We have increased the font size in each panel as well as the quality of the **ExtData Fig. 8A**.

R4-Q4 I am not quite sure why the authors included the FTY720 treatment experiment, but the result is very surprising. I almost get the impression that the treatment did not work, which sometimes happens. So far, the authors have not provided evidence that the FTY720 treatment worked, i.e., that the cells in the circulation were reduced following the treatment. Such evidence is essential, otherwise the data should be removed.

Response: We included this experiment to show that TDLNs did not contribute to the intratumoral CD8 T cell response induced by orthogonal ACT, which was uniquely associated with marked cell-autonomous expansion of adoptively transferred T cells in tumors and tumor regression. This result is not surprising for us since it has been previously demonstrated that intratumoral Tcf1⁺PD-1⁺CD8⁺ T cells are enough to mediate extended tumor control in response to therapeutic vaccination independently of TDLN [32].

Finally, the Reviewer is right that we did not provide direct evidence that the FTY720 worked. However, we know that it was the case because those experiments were done in parallel with the ones reported in our previous work [33], where we showed that antitumor efficacy of low-dose radiotherapy is dependent on the TDLN. However, as suggested by this Reviewer, we removed these data from the manuscript since we agree that such evidence (reduction of the cells in the circulation upon FTY720 treatment) is indeed essential.

R4-Q5 The authors refer to baseline GzmB expression in OT-1 before treatment as shown in Figure 4a, but there is no baseline data of OT-1 in Figure 4a. The same is true for Tox.

Response: We thank the reviewer for pointing to the lack of clarity in this point in our manuscript. We should clarify that in Figure 4a we refer to Gzmc, not Gzmb. Furthermore, we did not refer to “baseline expression data in OT1”, but rather we referred to antigen-experienced (CD44^{high}PD-1⁺) **endogenous** CD8⁺ TILs in untreated mice at the steady state (12 days post tumor inoculation, which is what we considered as baseline). These were, as expected, TOX⁺Gzmc^{low/neg}, consistent with canonical Tex. We have clarified this point in the revised manuscript (lines 193-200).

R4-Q6 It is unclear what kind of data is shown in Figure 4d. It says protein levels, but it looks more like a measurement of RNA expression. The authors should clarify what is shown in the legend.

Response: We confirm that this is protein level. This analysis was described in the methods section under the subheading “**Heatmaps**” (as shown below), and we have clarified this point in the figure’s legend (line 712).

“**Heatmaps:** The percentage of OT1 TILs expressing effector molecules (TNF α , IFN γ , GzmB, GzmA, Prf1, Ki67) was determined by FACS as explained in the methods section (Preparation of single tumor-cell suspensions, antibodies for flow cytometry and ex-vivo re-stimulation for cytokine production). These values were row scaled (normalization) to the mean. Heatmaps and clustering trees were generated in R v4.1.1 using the heatmap function from the *namesake* package. Euclidean distance was used for hierarchical clustering using the “ward” method.”

R4-Q6 2/3 of the manuscript and the core conclusion are derived from the extensive bioinformatics analysis. Given this extent, I am surprised that all three people who are listed in the authors contribution section to have performed the bioinformatics analysis are listed only as middle authors.

Response: In the author contribution section are listed four researchers. Dr. Jesus Corria Osorio who did 70% of the bioinformatics work (**From Fig.3 to Fig. 7, including extended Data Figures**). Drs. Santiago Carmona and Massimo Andreatta who did 25 % of the work (**Fig.2**) and finally Dr. Isaac Crespo who did 5% of the work (**Fig. 7e, f, h extended Data Figure. 7b,c**).

R4-Q7 The authors say that IL-2v/IL-33-induced progenitors remain Tox-negative and refer to Figure 8a of Ext, which shows no Tox expression data.

Response: The Reviewer is right, we did not show TOX expression data in (**Extended Data Fig. 8a**). Here, we were talking about the cluster C7, indeed we found that in progressing tumors, OT1 cells were not exhausted, but rather remained in this cluster, that we demonstrated in (**Fig. 7a-d**), is TOX-negative. We have also clarified this point in the revised manuscript (line 334).

Minor points:

In the legend of Figure 1, $\geq n5$ is indicated, but in panel B, the IL-2v alone group has only 4 mice.

Response: We thank the Reviewer for this observation. This was corrected (line 633)

The term "orthogonal" should be explained, what is the difference compared to autocrine or paracrine?

Response: We thank the Reviewer for this suggestion. A brief description of this term was added to the revised manuscript (line 49).

Finally, this term is not related with the terms autocrine and paracrine. "Orthogonal" is a mathematical term often used to illustrate variables that are statistically independent or in Euclidean geometry, lines that are perpendicular at their point of intersection. Similarly, two vectors are considered orthogonal if they form a 90-degree angle.

In this case we borrow it to illustrate perturbations that activate non-overlapped/complementary immune functional axes like T-cell stemness (IL2v) and innate tumor inflammation (IL-33). Indeed, the first cytokine is acting in an autocrine way (**Fig. 7**) on the gene engineered T cells and in a paracrine way on the endogenous CD8⁺ TILs (**Fig. R4B**, data not shown in the manuscript due to lack of space). On the other hand, IL-33 is only acting in a paracrine way since its receptor is not expressed in CD8⁺ TILs (**Extended Data Fig. 6b**), driving intratumoral inflammation by activating myeloid cells, and unleashing the cross-priming potential of tumor-associated DCs [34-39].

Extended data 1F, the colors in the legend in the figure and the lines of a group do not match.

Response: We thank the Reviewer for this observation. The colors are right, we just realized that the description of the group that corresponds to the treatment with UT OT1 plus anti-PD-1 (purple) is missing in the figure legend. This was fixed in the revised manuscript.

Extended Data 1F, while tumor growth rates look very different between early and late ACT, survival rates look much more similar, why is that?

Response: We thank the Reviewer for this observation. The tumor growth kinetic shown in Extended Data 1F left, is not related with the survival curves shown in Extended Data 1F right. For the sake of the clarity, we now split this figure in two panels.

The use of the term "bulk" in the following lines is confusing and should be removed: "we analyzed bulk CD8⁺ TILs from different ACT settings using single-cell RNA."

Response: The Reviewer is right. We are replacing it by "total CD8 TILs" (lines 118 and 153): "we analyzed total CD8⁺ TILs from different ACT settings using single-cell RNA-sequencing (scRNAseq)"

References

1. Alfei, F., et al., *TOX reinforces the phenotype and longevity of exhausted T cells in chronic viral infection*. Nature, 2019. **571**(7764): p. 265-269.
2. Scott, A.C., et al., *TOX is a critical regulator of tumour-specific T cell differentiation*. Nature, 2019. **571**(7764): p. 270-274.
3. Khan, O., et al., *TOX transcriptionally and epigenetically programs CD8+ T cell exhaustion*. Nature, 2019. **571**(7764): p. 211-218.
4. Seo, H., et al., *TOX and TOX2 transcription factors cooperate with NR4A transcription factors to impose CD8+ T cell exhaustion*. Proceedings of the National Academy of Sciences, 2019. **116**(25): p. 12410-12415.
5. Chen, J., et al., *NR4A transcription factors limit CAR T cell function in solid tumours*. Nature, 2019. **567**(7749): p. 530-534.
6. Chen, Y., et al., *BATF regulates progenitor to cytolytic effector CD8+ T cell transition during chronic viral infection*. Nature Immunology, 2021.
7. McLane, L.M., M.S. Abdel-Hakeem, and E.J. Wherry, *CD8 T Cell Exhaustion During Chronic Viral Infection and Cancer*. Annual Review of Immunology, 2019. **37**(1): p. 457-495.
8. Gabriel, S.S., et al., *Transforming growth factor- β -regulated mTOR activity preserves cellular metabolism to maintain long-term T cell responses in chronic infection*. Immunity, 2021. **54**(8): p. 1698-1714.e5.
9. Beltra, J.-C., et al., *Developmental Relationships of Four Exhausted CD8+ T Cell Subsets Reveals Underlying Transcriptional and Epigenetic Landscape Control Mechanisms*. Immunity, 2020. **52**(5): p. 825-841.e8.
10. Chu, T. and D. Zehn, *Charting the Roadmap of T Cell Exhaustion*. Immunity, 2020. **52**(5): p. 724-726.
11. Hudson, W.H., et al., *Proliferating Transitory T Cells with an Effector-like Transcriptional Signature Emerge from PD-1+ Stem-like CD8+ T Cells during Chronic Infection*. Immunity, 2019. **51**(6): p. 1043-1058.e4.
12. Zander, R., et al., *CD4+ T Cell Help Is Required for the Formation of a Cytolytic CD8+ T Cell Subset that Protects against Chronic Infection and Cancer*. Immunity, 2019. **51**(6): p. 1028-1042.e4.
13. Sheih, A., et al., *Clonal kinetics and single-cell transcriptional profiling of CAR-T cells in patients undergoing CD19 CAR-T immunotherapy*. Nature Communications, 2020. **11**(1): p. 219.
14. Codarri Deak, L., et al., *PD-1-cis IL-2R agonism yields better effectors from stem-like CD8+ T cells*. Nature, 2022.
15. Hashimoto, M., et al., *PD-1 combination therapy with IL-2 modifies CD8+ T cell exhaustion program*. Nature, 2022.
16. Andreatta, M., et al., *Interpretation of T cell states from single-cell transcriptomics data using reference atlases*. Nature Communications, 2021. **12**(1): p. 2965.
17. Wei, J., et al., *Targeting REGNASE-1 programs long-lived effector T cells for cancer therapy*. Nature, 2019. **576**(7787): p. 471-476.
18. McNally, A., et al., *CD4+CD25+ regulatory T cells control CD8+ T-cell effector differentiation by modulating IL-2 homeostasis*. Proc Natl Acad Sci U S A, 2011. **108**(18): p. 7529-34.
19. Chappert, P., et al., *Antigen-specific Treg impair CD8(+) T-cell priming by blocking early T-cell expansion*. Eur J Immunol, 2010. **40**(2): p. 339-50.
20. Sojka, D.K., Y.H. Huang, and D.J. Fowell, *Mechanisms of regulatory T-cell suppression - a diverse arsenal for a moving target*. Immunology, 2008. **124**(1): p. 13-22.
21. Hildeman, D., et al., *Apoptosis and the homeostatic control of immune responses*. Curr Opin Immunol, 2007. **19**(5): p. 516-21.

22. Welsh, R.M., K. Bahl, and X.Z. Wang, *Apoptosis and loss of virus-specific CD8+ T-cell memory*. *Curr Opin Immunol*, 2004. **16**(3): p. 271-6.
23. Badovinac, V.P., B.B. Porter, and J.T. Harty, *Programmed contraction of CD8(+) T cells after infection*. *Nat Immunol*, 2002. **3**(7): p. 619-26.
24. Zhao, M., et al., *Rapid in vitro generation of bona fide exhausted CD8+ T cells is accompanied by Tcf7 promotor methylation*. *PLoS Pathog*, 2020. **16**(6): p. e1008555.
25. Aibar, S., et al., *SCENIC: single-cell regulatory network inference and clustering*. *Nature Methods*, 2017. **14**(11): p. 1083-1086.
26. Lane, N. and W. Martin, *The energetics of genome complexity*. *Nature*, 2010. **467**(7318): p. 929-34.
27. Giles, J.R., et al., *Longitudinal single cell transcriptional and epigenetic mapping of effector, memory, and exhausted CD8 T cells reveals shared biological circuits across distinct cell fates*. *bioRxiv*, 2022: p. 2022.03.27.485974.
28. Stelekati, E., et al., *Long-Term Persistence of Exhausted CD8 T Cells in Chronic Infection Is Regulated by MicroRNA-155*. *Cell Reports*, 2018. **23**(7): p. 2142-2156.
29. Ataide, M.A., et al., *BATF3 programs CD8+ T cell memory*. *Nature Immunology*, 2020. **21**(11): p. 1397-1407.
30. Boccard, J. and D.N. Rutledge, *A consensus orthogonal partial least squares discriminant analysis (OPLS-DA) strategy for multiblock Omics data fusion*. *Anal Chim Acta*, 2013. **769**: p. 30-9.
31. Lisci, M., et al., *Mitochondrial translation is required for sustained killing by cytotoxic T cells*. *Science*. **374**(6565): p. eabe9977.
32. Siddiqui, I., et al., *Intratumoral Tcf1+PD-1+CD8+ T Cells with Stem-like Properties Promote Tumor Control in Response to Vaccination and Checkpoint Blockade Immunotherapy*. *Immunity*, 2019. **50**(1): p. 195-211.e10.
33. Herrera, F.G., et al., *Low-Dose Radiotherapy Reverses Tumor Immune Desertification and Resistance to Immunotherapy*. *Cancer Discovery*, 2022. **12**(1): p. 108-133.
34. Villarreal, D.O., et al., *Alarmin IL-33 Acts as an Immunoadjuvant to Enhance Antigen-Specific Tumor Immunity*. *Cancer Research*, 2014. **74**(6): p. 1789-1800.
35. Kallert, S.M., et al., *Replicating viral vector platform exploits alarmin signals for potent CD8+ T cell-mediated tumour immunotherapy*. *Nature Communications*, 2017. **8**: p. 15327.
36. Gao, K., et al., *Transgenic expression of IL-33 activates CD8+ T cells and NK cells and inhibits tumor growth and metastasis in mice*. *Cancer Letters*, 2013. **335**(2): p. 463-471.
37. Dominguez, D., et al., *Exogenous IL-33 Restores Dendritic Cell Activation and Maturation in Established Cancer*. *The Journal of Immunology*, 2017. **198**(3): p. 1365-1375.
38. Gao, X., et al., *Tumoral Expression of IL-33 Inhibits Tumor Growth and Modifies the Tumor Microenvironment through CD8⁺ T and NK Cells*. *The Journal of Immunology*, 2015. **194**(1): p. 438-445.
39. Villarreal, D.O. and D.B. Weiner, *Interleukin 33: a switch-hitting cytokine*. *Current Opinion in Immunology*, 2014. **28**: p. 102-106.

Decision Letter, second revision:
--

13th Jan 2023

Dear Dr. Coukos,

Thank you for submitting your revised manuscript "Orthogonal Cytokine Engineering Enables Novel Synthetic Effector States Escaping Canonical Exhaustion in Tumor-rejecting CD8+ T Cells" (NI-TR33866B). I have discussed your revisions with the other editors, and as noted previously we can commence the editorial check process. Hence, we'll be happy in principle to publish it in Nature Immunology, pending minor revisions to comply with our editorial and formatting guidelines.

We will now perform detailed checks on your paper and will send you a checklist detailing our editorial and formatting requirements in about a week. Please do not upload the final materials and make any revisions until you receive this additional information from us.

If you had not uploaded a Word file for the current version of the manuscript, we will need one before beginning the editing process; please email that to immunology@us.nature.com at your earliest convenience.

Thank you again for your interest in Nature Immunology Please do not hesitate to contact me if you have any questions.

Kind regards,

Laurie

Laurie A. Dempsey, Ph.D.
Senior Editor
Nature Immunology
l.dempsey@us.nature.com
ORCID: 0000-0002-3304-796X

Final Decision Letter:

Dear George,

I am delighted to accept your manuscript entitled "Orthogonal Cytokine Engineering Enables Novel Synthetic Effector States Escaping Canonical Exhaustion in Tumor-rejecting CD8+ T Cells" for publication in an upcoming issue of Nature Immunology.

Over the next few weeks, your paper will be copyedited to ensure that it conforms to Nature Immunology style. Once your paper is typeset, you will receive an email with a link to choose the appropriate publishing options for your paper and our Author Services team will be in touch regarding any additional information that may be required.

Due to the importance of these deadlines, we ask that you please let us know now whether you will be

difficult to contact over the next month. If this is the case, we ask you provide us with the contact information (email, phone and fax) of someone who will be able to check the proofs on your behalf, and who will be available to address any last-minute problems.

Please note that *Nature Immunology* is a Transformative Journal (TJ). Authors may publish their research with us through the traditional subscription access route or make their paper immediately open access through payment of an article-processing charge (APC). Authors will not be required to make a final decision about access to their article until it has been accepted. [Find out more about Transformative Journals](https://www.springernature.com/gp/open-research/transformative-journals).

Your paper will be published online soon after we receive your corrections and will appear in print in the next available issue. Content is published online weekly on Mondays and Thursdays, and the embargo is set at 16:00 London time (GMT)/11:00 am US Eastern time (EST) on the day of publication. Now is the time to inform your Public Relations or Press Office about your paper, as they might be interested in promoting its publication. This will allow them time to prepare an accurate and satisfactory press release. Include your manuscript tracking number (NI-TR33866C) and the name of the journal, which they will need when they contact our office.

About one week before your paper is published online, we shall be distributing a press release to news organizations worldwide, which may very well include details of your work. We are happy for your institution or funding agency to prepare its own press release, but it must mention the embargo date and *Nature Immunology*. Our Press Office will contact you closer to the time of publication, but if you or your Press Office have any enquiries in the meantime, please contact press@nature.com.

Also, if you have any spectacular or outstanding figures or graphics associated with your manuscript - though not necessarily included with your submission - we'd be delighted to consider them as candidates for our cover. Simply send an electronic version (accompanied by a hard copy) to us with a

possible cover caption enclosed.

If you have not already done so, we strongly recommend that you upload the step-by-step protocols used in this manuscript to the Protocol Exchange. Protocol Exchange is an open online resource that allows researchers to share their detailed experimental know-how. All uploaded protocols are made freely available, assigned DOIs for ease of citation and fully searchable through nature.com. Protocols can be linked to any publications in which they are used and will be linked to from your article. You can also establish a dedicated page to collect all your lab Protocols. By uploading your Protocols to Protocol Exchange, you are enabling researchers to more readily reproduce or adapt the methodology you use, as well as increasing the visibility of your protocols and papers. Upload your Protocols at www.nature.com/protocolexchange/. Further information can be found at www.nature.com/protocolexchange/about .

Please note that we encourage the authors to self-archive their manuscript (the accepted version before copy editing) in their institutional repository, and in their funders' archives, six months after publication. Nature Portfolio recognizes the efforts of funding bodies to increase access of the research they fund, and strongly encourages authors to participate in such efforts. For information about our editorial policy, including license agreement and author copyright, please visit www.nature.com/ni/about/ed_policies/index.html

Kind regards,

Laurie

Laurie A. Dempsey, Ph.D.
Senior Editor
Nature Immunology
l.dempsey@us.nature.com
ORCID: 0000-0002-3304-796X